# An investigation of the modulatory effects of empathic and autistic traits on emotional and facial motor responses during live social interactions

**Chun-Ting Hsu** [1]*, **Wataru Sato** [1]*, **Sakiko Yoshikawa** [2]

1 Psychological Process Research Team, Guardian Robot Project, RIKEN, Soraku-gun, Kyoto, Japan,
2 Institute of Philosophy and Human Values, Kyoto University of the Arts, Kyoto, Kyoto, Japan

* hsuchunting@gmail.com (CH); wataru.sato.ya@riken.jp (WS)

**Data Availability Statement:** Data tables of ratings and pre-processed EMG data (excluding trials with artifacts but including influential trials), and the R

## Abstract

A close relationship between emotional contagion and spontaneous facial mimicry has been theoretically proposed and is supported by empirical data. Facial expressions are essential in terms of both emotional and motor synchrony. Previous studies have demonstrated that trait emotional empathy enhanced spontaneous facial mimicry, but the relationship between autistic traits and spontaneous mimicry remained controversial. Moreover, previous studies presented faces that were static or videotaped, which may lack the "liveliness" of real-life social interactions. We addressed this limitation by using an image relay system to present live performances and pre-recorded videos of smiling or frowning dynamic facial expressions to 94 healthy female participants. We assessed their subjective experiential valence and arousal ratings to infer the amplitude of emotional contagion. We measured the electromyographic activities of the zygomaticus major and corrugator supercilii muscles to estimate spontaneous facial mimicry. Individual differences measures included trait emotional empathy (empathic concern) and the autism-spectrum quotient. We did not find that live performances enhanced the modulatory effect of trait differences on emotional contagion or spontaneous facial mimicry. However, we found that a high trait empathic concern was associated with stronger emotional contagion and corrugator mimicry. We found no two-way interaction between the autism spectrum quotient and emotional condition, suggesting that autistic traits did not modulate emotional contagion or spontaneous facial mimicry. Our findings imply that previous findings regarding the relationship between emotional empathy and emotional contagion/spontaneous facial mimicry using videos and photos could be generalized to real-life interactions.

## Introduction

Facial expressions of emotions serve as essential communication instruments during social interactions [1–3]. During such interactions, facial mimicry and emotional contagion mutually influence each other to facilitate social alignment [4]. Trait emotional empathy enhances

markdown files for the analysis are attached as Supporting information files.

**Funding:** WS and SY received funding from the 'Core Research for Evolutional Science and Technology' (CREST, https://www.jst.go.jp/kisoken/crest/en/about/index.html) of the 'Japan Science and Technology Agency' (JST), grant number JPMJCR17A5. WS received funding from the 'Mirai' Program (https://www.jst.go.jp/mirai/en/index.html) of the JST, grant number JPMJMI20D7. The funders had no role in study design, data collection and analysis, decision to publish, or preparation of the manuscript.

**Competing interests:** The authors have declared that no competing interests exist.

spontaneous facial mimicry and emotional contagion, but in previous studies, the relationship between autistic traits and spontaneous mimicry was inconsistent [5–12]. Moreover, most studies of facial expressions have used static photographs or pre-recorded video clips, compromising the ecological validity and generalizability of the results. We used a live image relay system that rendered live interactions possible, and we presented real-time dynamic facial expressions. This way, we previously reported that live social interactions evoke more emotional contagion and facial mimicry, compared with prerecorded photographs and videos [13, 14]. In this follow-up study, we investigated whether live social interactions enhanced the modulatory effects of trait emotional empathy and autistic traits on spontaneous facial mimicry and emotional contagion. We describe the theoretical constructs and research motivations below.

## Spontaneous facial mimicry, emotional contagion, emotional empathy, and autistic traits

Several definitions of empathy have been proposed [15], and some researchers regard empathy as a multidimensional concept [16, 17]. For example, Davis suggested separate measurements of the cognitive component (individual cognitive, perspective-taking capabilities or tendencies) and the emotional/affective component (individual emotional reactivity in response to the emotions of others). In the present study, we adopted Davis' broader definition of emotional empathy and the trait emotional empathy measurement of the Interpersonal Reactivity Index (IRI, see Methods for details). This approach allows consistent coupling between the theoretical construct and the measurements.

Spontaneous mimicry is a phenomenon whereby simply observing the non-verbal behavior of another individual implicitly elicits a corresponding action in the observer (i.e., without explicit instruction or intention to elicit such an action) [18]. Spontaneous facial mimicry refers to the spontaneous mimicry of facial expressions while observing a face; it typically emerges within 1 s after stimulus onset [19, 20]. In an early demonstration, Dimberg showed participants static grayscale pictures of happy or angry faces, while using electromyography (EMG) to record muscle-activation patterns similar to the presented photos. When participants viewed happy faces, the zygomaticus major (ZM) contracted (lip-corner pulling), and the corrugator supercilii (CS) relaxed; when they viewed angry faces, the CS contracted (frowning) [19].

Emotional contagion involves a collection of cognitive, psychophysiological, behavioral, and social processes by which a person or group influences the behavior of another person or group via conscious or unconscious induction of emotional states and behavioral attitudes [21, 22]. Facial mimicry is integral to primitive emotional contagion. Mechanistically, the "facial feedback hypothesis" [23] suggested that proprioceptive feedback of mimicry acts as an embodied sensorimotor simulation [18, 24, 25] that affects subjective emotional experiences, leads to emotional contagion, and facilitates inference of the emotional state of an individual's counterpart [24] as well as emotional empathy [26–28]. Other theoretical proposals, such as the social regulator view [29], suggest that emotional contagion (a feeling state) is independent of motor mimicry (a behavior) [30]. Mimicry was not a necessary precondition for emotional contagion—although they frequently occur concurrently—spontaneous facial mimicry and emotional contagion do not *necessarily* occur concurrently [30]. Affiliative and antagonistic social contexts can modulate spontaneous mimicry [30, 31]. Notably, the link between spontaneous facial mimicry and emotional contagion is not uniformly supported by empirical evidence [32–35]. Furthermore, the social regulator perspective also proposed that empathy and mimicry occur concurrently when pre-existing social bonds or desires to affiliate are involved [36]. Despite conflicting theoretical proposals, spontaneous mimicry involves multilevel

appraisal and response generation mechanisms [37] that emotional contagion cannot entirely explain. Therefore, in the present study, we regarded spontaneous facial mimicry and emotional contagion (as reflected in self-experiential ratings) as separate processes that often interact and occur concurrently; we separately investigated the modulatory effects of trait differences (trait emotional empathy and autistic traits) and live interactions on spontaneous facial mimicry and emotional contagion.

Spontaneous mimicry has been considered a primary motor feature of emotional empathy [38, 39]. Previous studies have revealed that the trait emotional empathy predicts the extent of spontaneous facial mimicry and emotional contagion [5–7, 40, 41]. Sonnby-Borgström analyzed EMG activities in response to static grayscale pictures of happy or angry faces by referencing high- versus low-trait emotional empathy. High trait emotional empathy participants exhibited more significant mimicking behavior and a stronger correlation between facial mimicry and self-reported feelings, compared with low trait emotional empathy participants [5]. Dimberg et al. reported that a high-emotional-empathy group rated happy and angry expressions as significantly happier and angrier, compared with the ratings by a low-emotional-empathy group [7]. A recent meta-analysis investigated the correlations among spontaneous facial mimicry, empathy, and emotion recognition in 28 studies. The moderator test revealed a weak but significant positive correlation between facial mimicry and trait emotional empathy ($r = 0.13$, $p = 0.001$) [42].

Individuals with an autism spectrum disorder (ASD) diagnosis have traditionally been associated with atypical social interactions and non-verbal communication, compared with individuals without an ASD diagnosis [43]. Spontaneous facial mimicry was reportedly reduced and delayed in individuals with an ASD diagnosis, compared with individuals without an ASD diagnosis [8–11]. According to Press et al. [12], this difference is presumably because task designs using simple facial action observations confound with the finding that individuals with an ASD diagnosis allocate less attention to social or communication cues, compared with individuals without an ASD diagnosis [44, 45]. When using a stimulus-response compatibility paradigm, Press et al. reported spontaneous mimicry and automatic imitation to be intact in individuals with an ASD diagnosis. The empathy profile of individuals with an ASD diagnosis has been continuously investigated and discussed [46]. Recent evidence indicates no difference in emotional empathy or emotional contagion between individuals with an ASD diagnosis and individuals without an ASD diagnosis [47–49].

It has been proposed that the autism spectrum is a continuum that covers the entire population (with and without ASD diagnosis). Baron-Cohen developed the Autism-Spectrum Quotient (AQ), a self-administered scale measuring the extent of autistic traits. Five elements are measured: social skill, attention-switching, attention to detail, communication, and imagination. Individuals with high autistic traits exhibit lower performance in all aspects, except attention to detail [50]. Few studies have investigated the relationship between autistic traits and spontaneous mimicry in participants without ASD diagnosis. Hermans et al. described reduced spontaneous facial mimicry of the CS response to static angry faces in women with high AQ, compared with women with lower AQ [51]. Sims et al. reported weaker ZM responses to dynamic smiling faces among individuals with high AQ (in the upper AQ quartile of their cohort) pre-conditioned to high monetary rewards, compared with individuals with low AQ (in the lower AQ quartile) [52]. Neufeld et al. also found a negative correlation between AQ score and ZM response to a virtual agent's dynamic, happy expressions [53]. As mentioned, these studies also involved task designs using simple facial action observations. In summary, empirical evidence concerning the effects of autistic traits or ASD diagnosis on spontaneous facial mimicry is inconsistent. No correlation of autistic traits and ASD diagnosis has been observed with respect to emotional contagion.

## Live interaction design, audience effect, and second-person neuroscience

Most studies regarding facial expression processing have used static photographs or pre-recorded video clips—such stimuli may lack "liveliness." The models delivering the stimuli generally do not interact with participants, who became detached observers who do not actively engage in social interaction. Considering the participants' lack of background knowledge about the models, contextual information that would aid social inference-making is limited and disorganized, creating a "spectatorial gap" between the inferred and the actual psychological state of the model [54, 55]. Such designs may compromise the ecological validity and generalizability of the results to social interactions in daily life [55–58].

Several theoretical proposals could explain the live effect. The audience effect refers to a change in behavior caused by the belief that an individual is under observation [59]. The watching eyes effect refers to the attention capture, self-referential processing, and prosocial behaviors elicited by the perception of a direct gaze [60]. These behavioral effects have been associated with metacognition for reputation management during social interactions [61–63]. Furthermore, Zajonc used drive theory to explain how the presence of conspecifics increases individuals' arousal and influences their performance of tasks [64].

Natural and contextual designs have been advocated for psychological studies regarding interpersonal interactions in the communication sciences, to improve the validity of research findings [58, 65, 66]. The theoretical evolution of ASD or autistic traits also requires a more naturalistic study design. Similar to the Milton "double empathy theory" [67], Bolis et al. developed the concept of "dialectical misattunement." Differences in predictive processing and active inferences between pairs of agents are considered critical in terms of the outcomes of social interactions [68]. Friendship quality is negatively correlated with the dyadic mismatch (operationalized as the absolute value of the AQ difference) between pairs of friends [69]. Crompton et al. showed that the effectiveness of information transfer along a chain of individuals with an ASD diagnosis was comparable with the effectiveness of information transfer along a chain of non-ASD individuals. Both chains were more effective than a mixed chain [70]. Face-to-face live interactions are necessary to investigate the effects of dialectical misattunement.

To our knowledge, no previous study has investigated the modulatory effects of emotional empathic and autistic traits on spontaneous mimicry and emotional contagion in response to live facial expressions. Several studies have used a "simulated live design" that lacked live interactions; the authors of these studies claimed that knowledge and beliefs about the potential for social interaction alone could change participant behaviors, such as eye gaze and autonomic and brain responses to social tasks [71–75]. In a recent study, we developed a live image-relay system that presented real-time dynamic facial expressions; we found that live social interactions evoked more emotional contagion and facial mimicry, compared with prerecorded photographs and videos [13]. This system and paradigm have been adapted for magnetic resonance imaging experiments, which revealed enhanced activity and connectivity in the right mirror neuron system under live conditions [14].

In the present study, we used the live image-relay system (Fig 1) established in our previous study [13]. Participants viewed live and videotaped dynamic facial expressions of positive (smiling) and negative (frowning) emotions (Fig 2), with simultaneous EMG recording of the ZM and CS. Subjective valence and arousal ratings [76] were also obtained (Fig 3). We assessed trait emotional empathy and autistic traits using the EC and PD subscales of the IRI [16, 17] and AQ [50], respectively. We used subjective experiential valence and arousal ratings to estimate emotional contagion, and we used EMG data to detect spontaneous facial mimicry. We tried to answer the following research questions:

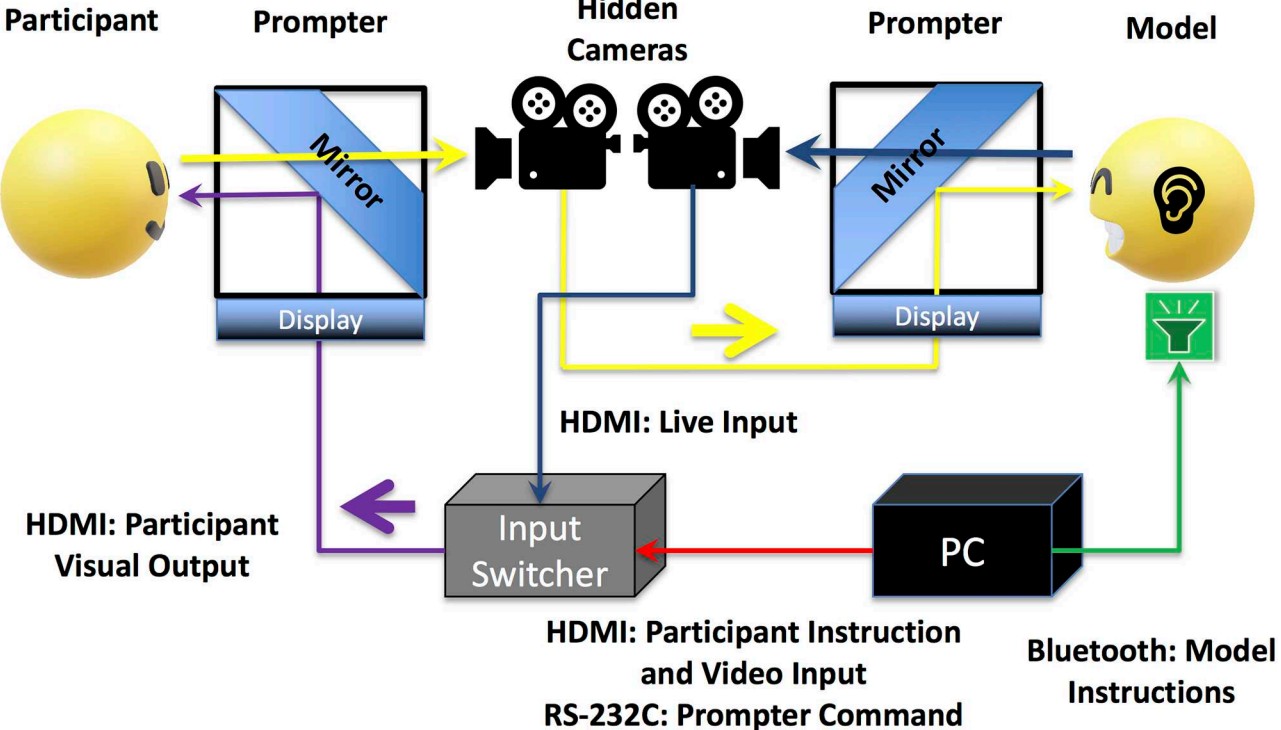

**Fig 1. The live image-relay system.** The participant and model both faced a prompter similar to that used by professional news broadcasters. The prompter featured a horizontal display and an oblique transparent mirror. The participant could view visual information through the reflection in the mirror; the participant's image was video-recorded and relayed live by a camera hidden behind the mirror. The participant's live image was sent to the prompter of the model (the yellow route). The live image of the model (captured by a hidden camera) was sent to an input switcher (the blue route) as a visual output option. The PC running the paradigm sent commands (the red route) via serial communication to the input switcher; this determined the visual output viewed by the participant (the purple route). During the video trials, the PC sent instructions and prerecorded video stimuli (red) to the input switcher. The switcher sent only the input from the PC to the participant's prompter (purple). During live trials, the switcher first relayed the instructions from the PC to the participant (red). The PC then sent an audio signal to the model via Bluetooth earphones to instruct the model to produce dynamic facial expressions (the green route) while also sending commands (red) to the input switcher to switch to the model's live image input (blue) and then switch back after the presentation of the stimulus (thus after 3 s).

1. Could live conditions further enhance the effects of high trait emotional empathy on emotional contagion?

2. Could live conditions further enhance the effects of high trait emotional empathy on spontaneous facial mimicry?

   Based on previous findings, we expected that dynamic live expressions would enhance the modulatory effects of trait emotional empathy through the audience effect in the affiliative context. Specifically, we expected that the enhancing effects of high trait emotional empathy on emotional contagion would be further enhanced under live conditions. We also expected that the enhancing effect of trait emotional empathy on spontaneous facial mimicry would be further enhanced under live conditions [5–7, 40].

3. Could live conditions modulate the relationship between autistic traits and emotional contagion?

4. Could live conditions modulate the relationship between autistic traits and spontaneous facial mimicry?

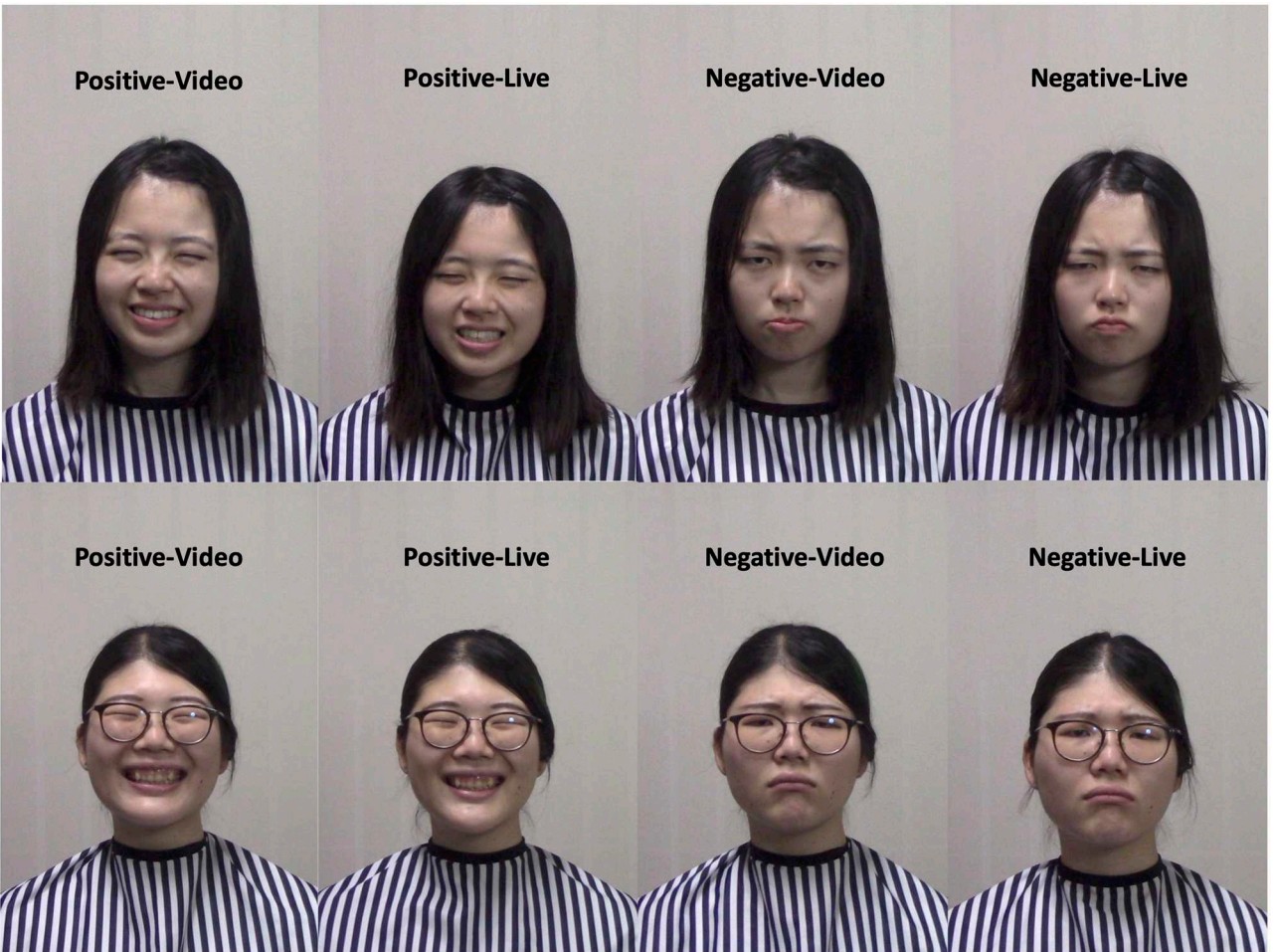

**Fig 2. Examples of positive (smiling) and negative (frowning) expressions.** The upper and lower panels show the prerecorded and live expressions of the two models. The models were instructed to display happy, smiling faces for positive expressions. They were asked to display angry, frowning faces with mouth protrusions for negative expressions.

Because previous studies did not describe a clear relationship between autistic traits and spontaneous facial mimicry, we explored the possibility that live conditions could evoke a modulatory pattern different from those observed under video conditions for spontaneous facial mimicry. We did not expect the autistic traits to modulate emotional contagion under live or video conditions.

## Materials and methods

### Participants

We recruited only female participants to control for any effect of gender differences on facial mimicry responses [77, 78]. Ninety-four women from the Japanese cities of Kyoto and Osaka were recruited to this study (mean ± standard deviation [SD] age = 22.851 ± 2.396 years; range: 18–29 years). We previously published the data of 23 participants collected between August and October 2019 [13]. Another 27 participants were evaluated between August and September 2020. The remaining 44 participants participated in a published neuroimaging study of the same design between February and March 2021 [14]. EMG recordings failed for

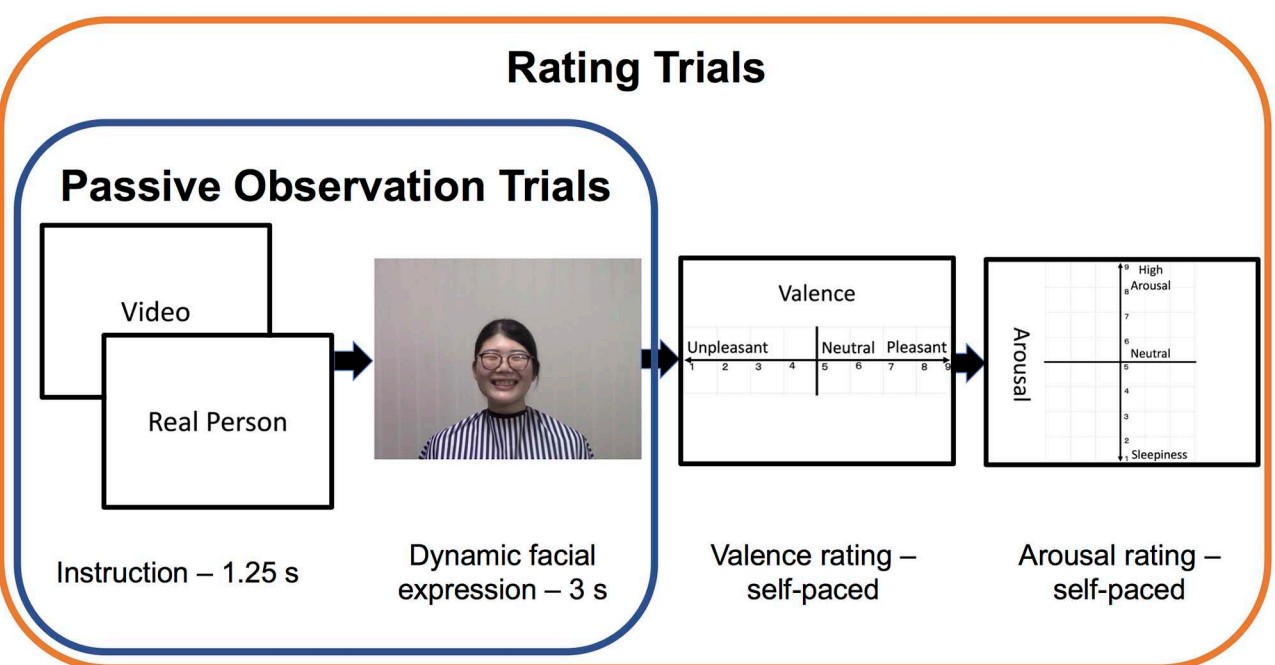

**Fig 3. The experimental paradigm.** During the passive viewing trials, each participant first viewed an instruction of either "Video" or "Real Person," followed by prerecorded and live dynamic facial expressions, respectively, 3-s in duration, with a fixation cross that remained for the duration of the jittered inter-stimulus interval. During the subsequent rating trials, the participants viewed the instruction followed by the stimulus as in the passive viewing trials; they were asked to rate subjective experiential valence and arousal using a keyboard placed in front of them.

one of the 44 neuroimaging participants due to an equipment setup error. This study was approved by the ethics committee of the Unit for Advanced Studies of the Human Mind, Kyoto University, and performed following the ethical standards of the committee. Written informed consent was obtained from all participants prior to their participation in the study. All participants consented to the publication of their data and to being recorded on video. All participants were rewarded monetarily.

The sensitivity analysis for two-tailed t-tests of multiple regression coefficients with 12 predictors and a sample size 94 showed that the present study has 80% of power for a minimum detectable effect size of $f^2 = 0.0855$, $r^2 = 0.0788$. Furthermore, for an effect size as small as $r^2 = 0.01$, a sample size of 61 is required to achieve the point of stability with 80% confidence for the effect size estimation [79, 80]. Therefore, the sample size of the present study should be more than sufficient to accurately estimate small effect sizes and make inferences based on that.

While there might be concerns that the study type (behavioral vs. neuroimaging) might influence the participants' responses, we also included robust estimation results of 50 participants from the behavioral study in the supplementary tables. A sensitivity analysis for two-tailed t-tests of multiple regression coefficients with 12 predictors and a sample size of 50 showed that the present study has 80% of power for a minimum detectable effect size of $f^2 = 0.1655$, $r^2 = 0.1420$, which is close to the medium effect size of $f^2 = 0.15$, $r^2 = 0.13$ as described by Cohen [81].

## Materials

**Trait emotional empathy: IRI.** Davis developed the Interpersonal Reactivity Index (IRI), which comprises four 7-item subscales that estimate perspective-taking (PT), fantasy (FS),

empathic concern (EC), and personal distress (PD). The EC subscale assesses an individual's warmth, compassion, and concern for others. The PD subscale estimates an individual's discomfort and anxiety when witnessing others' negative experiences. The PT subscale reflects an individual's tendency (or ability) to adopt another individual's perspective (or point of view), which corresponds to cognitive empathy [16]. The IRI (particularly the EC subscale) remains the most prevalent measure of empathy [15]; overall, the IRI is a multidimensional measure of empathy [16, 17]. Davis presumed that the EC and PD subscales estimated the emotional/affective component of empathy. However, the PD subscale was significantly correlated with the EC subscale ($r = 0.380$, $p = .0002$) and the AQ ($r = 0.392$, $p < .0001$) in our samples. The statistical models including the IRIPD, IRIEC, and AQ were easily overfitted, preventing the modeling of random slopes of the emotional and presentation condition and negatively affecting the generalizability of our statistical results [82]. Therefore, we included only the EC subscale in our statistical analysis.

**Autistic traits: AQ.** Assuming that the autism spectrum is a continuum that covers the entire population (with and without an ASD diagnosis), the AQ measures individual autistic traits. The 50 questions assess five areas: social skills, attention-switching, attention to detail, communication, and imagination [50].

**Subjective experiential ratings: The Russel "affect grid".** Subjective experiential ratings are used to estimate the extent of emotional contagion in the present study. We instructed the participants about the ratings based on the Russel "affect grid" [76]. The instructions read: "When viewing the images, please rate the emotion you feel. Please rate the emotion in two dimensions: (1) Pleasure-displeasure, which is about the quality of the emotion, and (2) arousal, which is about the energy of the emotion. Please input the number you find most suitable (1–9) in the sequence of 'pleasure-displeasure' and 'arousal' using the typewriter keys". Further explanation of valence and arousal using the affect grid was then provided: "The center of the square represents the neutral emotion, which is neither positive nor negative; it is the ordinary emotion you experience. The upper right corner represents the emotion of excitement and happiness. The lower left corner represents the emotion of depression, sadness, or worry. The upper left corner represents stress and nervousness. The lower right corner represents tranquility and relaxation."

**Facilities.** For the 50 participants of the behavioral experiment, each model and participant faced a prompter. A concealed VIXIA HF R800 camera (Canon Inc., Tokyo, Japan) was placed behind the mirror of each prompter (Fig 1). The models viewed "live-relay images" of the participants. Presentation software (Neurobehavioral Systems, Inc., Berkeley, CA, USA) was used to run the experiments on a Precision T3500 computer (Dell Inc., Round Rock, TX, USA) with the Windows 7 Professional operating system (Microsoft Corp., Redmond, WA, USA). Serial signals were sent to an SL-41C switcher (Imagenics Co., Ltd., Tokyo, Japan) to switch the visual input of the participant's prompter between the computer and the model's camera. Presentation software transmitted sounds to the models using a Pasonomi TWS-X9 Bluetooth earphone (Shenzhen City Meilianfa Technology Co., Ltd., Shenzhen, China), instructing them to produce dynamic facial expressions. The switcher maintained the resolution (1,280 × 960) and the height-to-width ratio of the video output of the presentation software and live-relay images. A BrainAmp ExG MR amplifier (Brain Products, Munich, Germany) and BrainVision Recorder software (Brain Products) were used to record EMG from the ZM and CS of participants. The setting for the 44 participants of the neuroimaging experiment is identical, except that the participants lay in the scanner viewing the image relayed by the switcher.

**Paradigm and procedures.** As described previously [13], a 2 × 2 study design was used, with factors of presentation (video vs. live) and emotion (positive vs. negative), resulting in

four conditions (Fig 3). After obtaining written informed consent and the EMG electrodes were attached, the participant and model engaged in conversation for 3 minutes through the prompter. The conversation demonstrated to the participants that the prompter system transmitted live images of the model. In the behavioral experiment (50 participants), the conversation was followed by eight practice passive-viewing trials, 60 passive-viewing trials, four practice rating trials, and 16 rating trials. In the neuroimaging experiment (44 participants), there were 200 passive-viewing trials, four practice rating trials, and 16 rating trials. The practice trials gave participants an understanding of what to expect and allowed them to ask any questions before data acquisition began. The relative proportions of passive viewing (EMG) trials and rating trials aligned with our prior assumption about the effect size and signal-to-noise ratio of spontaneous facial mimicry and ratings. The EMG would have a smaller effect size and lower signal-to-noise ratio than ratings, thus requiring more repetitions per participant. However, participants might experience a loss of attention or interest if the passive viewing period persists for too long. The total testing time per session for the study's behavioral and neuroimaging versions was less than 1 hour.

During the passive-viewing component, the participants fixated on a cross (behavioral experiment: mean inter-trial interval = 2,604 ms; range: 2,000–3,750 ms; neuroimaging experiment: mean inter-trial interval = 4,475 ms; range: 2,950–10,150 ms) until they were shown the announcement of either "Video" or "Real Person" for 1 s (1.25 s in the neuroimaging experiment). The instruction enabled a clear audience effect [59, 60, 63]. Participants did not need to de-ambiguate or infer a live vs. pre-recorded situation by relying on subtle cues such as microexpressions or environmental inconsistencies, which do not guarantee successful detection/ judgment of live performances. Also, otherwise, it would have been difficult to ensure whether participants were really in mental states appropriate for live interactions. Without instructions, it would be more difficult to claim whether an effect (if any) was attributable to low-level processing rather than social cognition. During the video trials, one pre-recorded video clip was presented immediately after the instructions. During the live trials, the models performed dynamic facial expressions according to the instructions in their earphones. The screen displayed the fixation cross again after the facial expressions had been shown. In the behavioral experiment, after eight practice trials, 15 passive-viewing trials were completed per condition for a total of 60 trials, with a break after 32 trials (8 trials per condition). In the neuroimaging experiment, after eight practice trials, there were five runs of 40 trials (10 trials per condition) for a total of 200 trials. To avoid movement-associated EMG artifacts, and to detect spontaneous, automatic facial mimicry without top-down emotional processing, as in previous studies [83, 84], no rating was requested during the passive-viewing trials. For each participant, the sequence of conditions during the trial was pseudo-randomized, and the presentation sequence of the pre-recorded videos per condition was randomized. EMG data were collected during the passive-viewing trials.

After the passive-viewing trials, participants received instructions about and explanations of the subjective experiential ratings. During the subjective rating trials [76], each trial began with the described passive-viewing component; valence (pleasure-displeasure) and arousal ratings were completed sequentially after the participant had viewed dynamic facial expressions (Fig 3). Four practice rating trials were followed by 16 test-rating trials, during which no EMG data were acquired. For video conditions, the clips used in the passive viewing trials (15 per emotional condition) and rating trials (4 per emotional condition) were not repeated. After the ratings had been completed, the electrodes were removed; participants then completed the AQ and IRI questionnaires. Seventy-two participants (all participants tested in 2020 and 2021 and one participant in 2019) also completed the State-Trait Anxiety Inventory [85] and the Interaction and Audience Anxiousness Scale [86] (the resulting data were not

analyzed in the present study). The participants were remunerated for their participation and then dismissed.

**Pre-recorded videos.**  As described previously [13, 14], four female models each recorded more than 20 video clips of positive and negative dynamic facial expressions (Fig 2). The models were instructed to display happy, smiling faces for positive expressions. They were instructed to display angry, frowning faces with a protruding mouth for negative expressions. The models' appearances were consistent between the video clips and live performances. An apron covered their clothes, and hairpins were used to maintain their hairstyles. The clips lasted 3 s; they featured a neutral expression, gradual dynamic changes, and maximal emotional facial expression, all sustained for 1 s. Two smiling and two frowning clips were used for the positive-video and negative-video conditions in practice passive-viewing trials; 15 clips per video condition (50 clips per video condition in the neuroimaging study) were used in the actual passive-viewing trials; one clip per video condition for the practice ratings, and four clips per video condition were used for the actual ratings. During passive-viewing trials, no clips were repeated within a single condition. The positive-live and negative-live conditions did not use pre-recorded clips; the models' live performances were relayed. Two models in the behavioral study gave written informed consent (as outlined in the PLOS consent form) to publish their images (Figs 2 and 3).

**Stimuli validations.**  We recorded and visually inspected the live performances of the models during the passive-viewing trials to ensure valid performances of dynamic facial expressions. In the behavioral study, one positive trial for two participants and one negative trial for two other participants were excluded from the analysis because they either were not performed correctly or were performed after a noticeable delay. Video recordings of the live performances in the 2019 sessions were validated by naïve participants, whose valence and arousal ratings were not different from the ratings for prerecorded stimuli. The naïve participants could not differentiate between the prerecorded stimuli and live-performance recordings [13]. Video recordings of the live performances and pre-recorded videos in the 2021 neuroimaging study were validated using OpenFace [87, 88], showing that the action unit 12 (AU12, lip corner puller, corresponding to the ZM) dynamic change was significantly weaker in the positive-live than in the positive-video trials. AU4 (brow lowerer, corresponding to the CS) dynamic change was significantly weaker in the negative-live than in the negative-video trials [14]. This increased our certainty that the observed interaction between emotion and presentation condition was not due to enhanced emotional expression during the live performances.

Because some people might suspect that the models' negative expressions did not clearly distinguish between angry (as instructed) and sad, we performed another post hoc rating study on sadness and anger using a Likert scale (range: 1–5). Sixteen negative video clips (four per presentation condition [prerecorded or live]) were presented to 28 naïve female participants (mean ± SD age = 27.64 ± 3.07 years; range: 20–30 years). The negative stimuli were highly rated in terms of both anger (mean ± SD = 3.36 ± 0.61), and sadness (mean ± SD = 3.01 ± 0.85), but a paired t-test of the participant-wise mean rating values showed that participants generally perceived more anger than sadness in the negative stimuli (t = 2.55, degrees of freedom [df] = 27, p = 0.017). The distribution of the anger and sadness ratings did not violate the assumption of normality (anger ratings: Shapiro-Wilk W = 0.96, p = 0.39; sadness ratings: Shapiro-Wilk W = 0.96, p = 0.30).

**EMG data pre-processing.**  For the neuroimaging experiment, we first inspected video recordings of participants inside the scanner, and excluded runs in which the participant closed her eyes for more than half of the time for EMG analysis (see [14] for details). Raw EMG data from the neuroimaging study were first pre-processed using the BrainVision Analyzer 2 software. MR gradient artifacts were removed based on the pulse trigger, and data were

down-sampled to 500 Hz. Raw EMG data were pre-processed using the EEGLAB MATLAB toolbox (version 2019.1; Swartz Center for Computational Neuroscience, San Diego, CA, USA). A notch filter was applied at around 60 Hz and multiples of 60 Hz; a high-pass filter was applied at 20 Hz, and a low-pass filter was applied at 500 Hz. The data were then screened for movement artifacts. For each trial, the signal was detrended and baseline-corrected for low-frequency drift. The baseline for this preprocessing step was defined as the mean value from 3 s before to 1 s after stimulus onset, which was the end of the model's neutral expression immediately before the dynamic facial expression change. All data points in each trial were shifted by the per-trial baseline mean value. The data points were then rectified to retrieve the amplitudes of oscillation. The absolute or rectified values (oscillation amplitudes) plus one were natural-log-transformed to correct for the right-skewness of the raw data distribution. Trial-wise, the mean differences in the EMG natural log-transformed oscillation amplitudes between the neutral (0–1 s after stimulus onset) and maximal phases (2.5–3.5 s after stimulus onset) of the dynamic facial expression served as dependent variables (facial muscular responses) during statistical analysis.

**Statistical analyses.** Linear mixed effect (LME) models, model comparisons, and influence diagnostics were performed using R software (version 4.3.1; R Core Team; Vienna, Austria) and the packages lme4 1.1–34, lmerTest 3.1–3, HLMdiag 0.5.0, and emmeans 1.8.7, along with the optimizer BOBYQA. The dependent variables included valence ratings, arousal ratings, and the ZM and CS responses. For each LME model, the emotional condition, presentation condition, mean-centered IRIEC, AQ, and their interactions were fixed effects, whereas "subject" and "Type" (behavioral vs. neuroimaging study) were random factors. The reference level for the emotional condition was "Negative," and the reference level for the presentation condition was "Video." Model complexity, in terms of the number of random effects (n(n+1)/2 model parameters to be estimated), was increased in a stepwise manner; each model was compared with the previous (i.e., less complicated) model using F-tests [lmerTest::anova()] until the more complex model was no longer significantly superior to the simpler one. The model iteration process was considered complete when the model estimation resulted in a singular fit, indicating overfitting or failure to converge. The most complex model showing convergence was chosen. Initially, random intercepts for participants were included; the inclusion of by-subject random slopes for the effects of the emotional condition, presentation condition, their interaction term, and random intercepts for the study type depended on the results of the model comparison process. The random intercept for the study type was included in the model whenever it did not cause a singular fit to account for the potential effect of the study type.

The homogeneity of residual variance is an a priori assumption when seeking valid inferences from LME models [89, 90]. We first performed the robust LME (robustlmm 3.2) to check the extent of bias caused by highly influential data points in the model without diagnostics. In the robust approach [91], each item's robustness weight is estimated; highly influential data points are given a low robustness weight. This approach could limit the bias caused by highly influential data points, while retaining all data points in the model. However, information criteria (e.g., Akaike's information criterion and tests based on log-likelihood statistics), Satterthwaite and Kenward-Roger approximation of the degrees of freedom, and estimation of effect sizes and their confidence intervals are, for the time being, unavailable in the robust approach. Therefore, we used the Satterthwaite approximation of the degrees of freedom of the non-robust estimation of the complete data set to calculate p-values for the robust estimation [92]. When the robust and non-robust estimations showed compatible significant effects, we report and discuss the results based on the non-robust estimation of the complete data set. Results of the robust estimation are presented in the supplementary tables S1, S3, S5, and S7

Tables. To show whether participants of the neuroimaging study responded differently, we also included results of the robust estimation of the 50 participants in the behavioral study in the supplementary tables S2, S4, S6, and S8 Tables.

When the robust and non-robust estimations showed incompatible significant effects (as in CS responses), we performed upward residual and influence analyses using the R package HLMdiag [93]. Trials were excluded if they had an absolute standardized residual larger than 3, or a Cook's distance greater than $1.5 \times$ the interquartile range above the third quartile ($Q_3 + 1.5 \times IQR$ as implemented in the HLMdiag package). After the reduced data had been refitted, the second-level Cook's distance was checked, as suggested in the HLMdiag package [93], and highly influential participants were excluded. We present and discuss the results of the CS responses for which model diagnostics were performed.

The Satterthwaite approximation was used to evaluate the df values. Effect size estimation of the fixed effects' semi-partial R-squared (variance-explained) was conducted using the R package r2glmm 0.1.2 and the Nakagawa and Schielzeth approach [94–96]. According to Cohen [81], the effect sizes for partial $r^2$ values of regression models are considered small at 0.0196 ($f^2 = 0.02$), medium at 0.13 ($f^2 = 0.15$), and large at 0.26 ($f^2 = 0.35$), but see also [97]. Interactions between emotional conditions (positive vs. negative) and presentation conditions (live vs. video) were evaluated using a simple-effects analysis via the emmeans function. Simple-slope analyses were performed to investigate significant interactions among individual differences, emotional conditions, and presentation conditions using the emtrends function in the R package emmeans. The "pick-a-point" approach [98] was used for two-way interactions involving one continuous and one categorical independent variable, as follows. Simple-effect contrasts were estimated using the emmeans function at mean-centered continuous variable levels from -3 SD to 3 SD in steps of 1 SD, with comparisons between two categorical variable levels; the Šidák method was used to correct for multiple comparisons. The emmeans package uses a reference grid approach to estimate/predict marginal means based on a fitted model [99], rather than based directly on the data. The purpose of this package was to provide an interpretation of the fitted model (please see the Basics vignette for the emmeans package). Because a significant main effect involved in a significant interaction could be biased, we refrained from reporting such main effects in the Results section. Readers can examine the simple-effect/simple-slope results to determine whether the significant interaction biased the main effect. Marginal effects of interaction terms involving individual differences (continuous variables) were visualized using the ggemmeans function in the ggeffects 1.1.2 R package. Two-tailed p-values are reported throughout the text.

## Results

### Questionnaire scores

The mean IRIEC ± SD score was 17.745 ± 5.168 (range: 2–28), and the standardized IRIEC Cronbach's α was 0.788. The AQ's mean ± SD score was 21.830 ± 6.920 (range: 4–41), and the standardized Cronbach's α was 0.764.

### Subjective ratings

**Valence ratings.** The final LME model of valence ratings included the emotional conditions, the presentation conditions, the mean-centered IRIEC and AQ scores, and their interactions with the emotion and presentation conditions as fixed effects. It also included the random intercept and random slopes for the emotional and presentation conditions and their interactions by the random factor "subject" (Table 1), as well as the random intercept by the random factor "Type." Adding the random intercept over the factor "Type" significantly

**Table 1. Statistical summary of valence ratings.**

**Fixed Effects**

| Effect | Beta | 95% CI | SE | df | t-value | Pr(>|t|) | Rsq | Rsq CI |
|--------|------|--------|-----|------|---------|----------|------|--------|
| Intercept | 5.304 | (4.705, 5.903) | 0.249 | 0.990 | 21.312 | 0.031* | | |
| Emotion | 2.261 | (2.065, 2.458) | 0.101 | 91.000 | 22.444 | < 0.001* | 0.661 | (0.637, 0.683) |
| Presentation | 0.127 | (0.047, 0.206) | 0.041 | 91.000 | 3.110 | 0.003* | 0.006 | (0.001, 0.016) |
| E * P | 0.339 | (0.216, 0.462) | 0.063 | 91.000 | 5.378 | < 0.001* | 0.021 | (0.009, 0.038) |
| IRIEC | 0.004 | (-0.014, 0.023) | 0.009 | 90.651 | 0.395 | 0.694 | <0.001 | (<0.001, 0.005) |
| IRIEC * E | 0.087 | (0.049, 0.126) | 0.020 | 91.000 | 4.384 | < 0.001* | 0.069 | (0.047, 0.095) |
| IRIEC * P | 0.0009 | (-0.015, 0.017) | 0.008 | 91.000 | 0.110 | 0.913 | <0.001 | (<0.001, 0.003) |
| IRIEC * E * P | -0.016 | (-0.040, 0.008) | 0.012 | 91.000 | -1.297 | 0.198 | 0.001 | (<0.001, 0.007) |
| AQ | 0.010 | (-0.003, 0.024) | 0.007 | 90.398 | 1.486 | 0.141 | 0.004 | (<0.001, 0.012) |
| AQ * E | 0.025 | (-0.004, 0.054) | 0.015 | 91.000 | 1.656 | 0.101 | 0.010 | (0.003, 0.023) |
| AQ * P | -0.009 | (-0.021, 0.002) | 0.006 | 91.000 | -1.545 | 0.126 | 0.002 | (<0.001, 0.008) |
| AQ * E * P | -0.017 | (-0.035, 0.001) | 0.009 | 91.000 | -1.843 | 0.069 | 0.003 | (<0.001, 0.010) |

**Random Effects**

| Group | Effect | Variance | SD | 95% CI | Corr. I. | 95% CI | Corr. E. | 95% CI | Corr. P. | 95% CI |
|-------|--------|----------|-----|--------|----------|--------|----------|--------|----------|--------|
| Subject | Intercept | 0.180 | 0.424 | (0.355, 0.497) | | | | | | |
| | E | 0.898 | 0.948 | (0.804, 1.091) | 0.13 | (-0.095, 0.342) | | | | |
| | P | 0.100 | 0.317 | (0.242, 0.385) | -0.15 | (-0.423, 0.128) | -0.01 | (-0.252, 0.264) | | |
| | E * P | 0.261 | 0.511 | (0.403, 0.613) | 0.19 | (-0.100, 0.442) | 0.14 | (-0.107, 0.379) | 0.64 | (0.389, 0.858) |
| Type | Intercept | 0.120 | 0.346 | (0.095, 1.037) | | | | | | |
| Residual | | 0.451 | 0.672 | (0.645, 0.700) | | | | | | |

Formula: Valence ~ 1 + emotional_condition * presentation_condition * IRIEC + emotional_condition * presentation_condition * AQ + (1 + emotional_condition * presentation_condition | subject) + (1 | Type). Number of observations: 1504. Number of subjects: 94. Abbreviations: AQ: autism spectrum quotient; CI: 95% confidence interval; Corr. I.: correlation with the random effect of the intercept; Corr. E.: correlation with the random effect of the emotional conditions; df: Satterthwaite approximations of the degrees of freedom; IRIEC: empathic concern subscale of the Interpersonal Reactivity Index; Rsq: effect size of the semi-partial R-squared; SD: standard deviation; SE: standard error; * $p < 0.05$

improved the model fitting (model comparison $p$ = 8.992e-05). We noted a significant interaction between the emotional conditions and presentation conditions. Simple-effect analysis according to the emotional condition showed that the interaction was attributable to a significant effect of positive-live (mean = 7.16, standard error [SE] = 0.267, 95% confidence interval [CI] = [5.21, 9.11]) being rated as more positive than positive-video (mean = 6.64, SE = 0.264, 95% CI = [4.58, 8.17]; difference = 0.519, SE = 0.102, df = 91, t = 5.076, $p$ < 0.0001*) conditions; while negative-live (mean = 3.63, SE = 0.259, 95% CI = [1.29, 5.96]) was rated as more negative than negative-video (mean = 3.79, SE = 0.260, 95% CI = [1.50, 6.07]; difference = -0.160, SE = 0.065, df = 91, t = -2.469, $p$ = 0.0154*) conditions. Simple-effect analysis according to the presentation condition showed that the differences between both the positive-live and negative-live conditions (difference = 3.54, SE = 0.162, df = 91, t = 21.787, $p$ < 0.0001*) and the positive-video and negative-video conditions (estimate = 2.86, SE = 0.149, df = 91, t = 19.187, $p$ < 0.0001*) were significant.

We found a two-way interaction between the IRIEC and the emotional conditions. The 95% CI of the IRIEC slope was above zero under the positive condition (slope = 0.066, SE = 0.018, df = 91.6, 95% CI = [0.030, 0.101]) but below zero under the negative condition (slope = -0.058, SE = 0.016, df = 91.6, 95% CI = [-0.090, -0.026]; slope difference = 0.124, SE = 0.028, df = 91, t = -4.384, $p$ < 0.0001*). We performed a further simple-effect contrast

**Table 2. Simple-effect contrasts of the IRIEC-emotional condition interactions in terms of the valence ratings.**

| IRIEC | Mean Positive | Mean Negative | Difference | SE | df | t-value | Pr(>\|t\|) |
|---|---|---|---|---|---|---|---|
| -3SD | 5.89 | 4.61 | 1.28 | 0.459 | 91 | 2.793 | 0.0064* |
| -2SD | 6.23 | 4.31 | 1.92 | 0.324 | 91 | 5.927 | < .0001* |
| -SD | 6.57 | 4.01 | 2.56 | 0.204 | 91 | 12.565 | < .0001* |
| 0 | 6.90 | 3.71 | 3.20 | 0.142 | 91 | 22.444 | < .0001* |
| +SD | 7.24 | 3.41 | 3.84 | 0.204 | 91 | 18.832 | < .0001* |
| +2SD | 7.58 | 3.11 | 4.47 | 0.324 | 91 | 13.803 | < .0001* |
| +3SD | 7.92 | 2.81 | 5.11 | 0.459 | 91 | 11.129 | < .0001* |

Abbreviations: df: The Satterthwaite approximations of the degrees of freedom. See Table 1 footnote.

study comparing the positive and negative conditions at mean-centered IRIEC levels from -3 SD to 3 SD in steps of 1 SD, using the Šidák method to correct for multiple comparisons. Individuals with a middle-to-high trait emotional empathy (IRIEC levels of the mean minus two SD and above) tended to feel more positive when observing positive stimuli and feel more negative when watching negative stimuli (Table 2, Fig 4A); individuals with low trait empathic concern tended not to differ in terms of valence when observing stimuli of different emotional conditions.

Regarding data from the 50 participants of the behavioral study, robust estimation also showed a significant interaction between the IRIEC and the emotional conditions, but no interaction between emotional and presentation conditions (S2 Table). Robust estimation showed a significant interaction between the AQ and the presentation conditions, which was

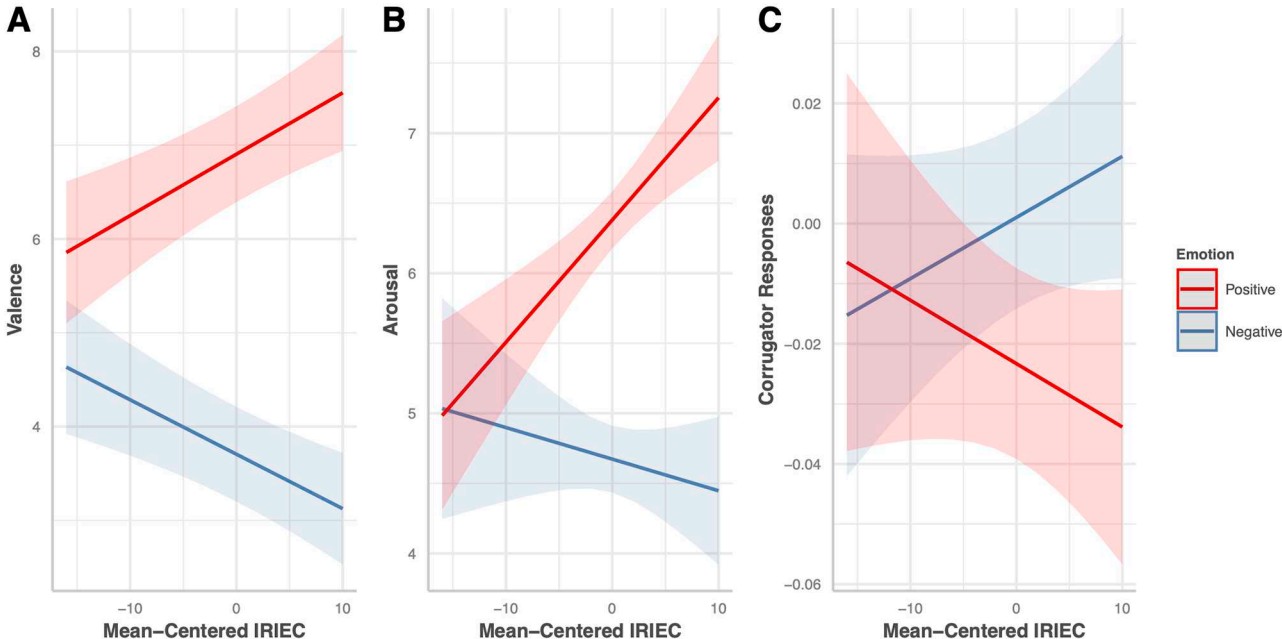

**Fig 4. Two-way interactions between empathic concern and emotional conditions on valence, arousal, and corrugator supercilii reactions.** The lines indicate the values predicted according to the trait scores. The semi-transparent band indicates the 95% confidence interval. A: The two-way interaction between the score on the empathic concern subscale of the Interpersonal Reactivity Index (IRIEC) and emotional conditions, as revealed by the valence ratings. B: The two-way interaction between the IRIEC and the emotional conditions, as revealed by the arousal ratings. C: The two-way interaction effects between the IRIEC and emotional conditions in terms of the CS responses.

not present in the non-robust estimation. Still, the non-robust estimation after model diagnostics showed no such effect ($p = 0.116$), and the lower bound of the effect size confidence interval included 0.

**Arousal ratings.** The final LME model for arousal ratings exhibited the same fixed effects as the valence rating model (Table 3). The random effects included the random intercept and random slopes for the emotional and presentation conditions and their interactions by the random factor "subject." The random factor "Type" could not be included because it invariantly caused singular fits. We noted a significant interaction between the emotional conditions and presentation conditions. Simple-effect analysis according to the emotional condition showed that the interaction was attributable to a significant difference between the positive-live (mean = 6.66, SE = 0.115, 95% confidence interval [CI] = [6.44, 6.89]) and positive-video (mean = 6.10, SE = 0.122, 95% CI = [5.85, 6.34]; difference = 0.569, SE = 0.114, df = 91, t = 4.987, $p < 0.0001^*$) conditions; there was a smaller difference between the negative-live (mean = 4.80, SE = 0.137, 95% CI = [4.53, 5.07]) and negative-video (mean = 4.54, SE = 0.122, 95% CI = [4.30, 4.79]; difference = 0.261, SE = 0.092, df = 91, t = 2.833, $p = 0.0057^*$) conditions. Simple-effect analysis according to the presentation condition showed that the differences between both the positive-live and negative-live conditions (difference = 1.86, SE = 0.193, df = 91, t = 9.656, $p < 0.0001$) and the positive-video and negative-video conditions (estimate = 1.55, SE = 0.178, df = 91, t = 8.729, $p < 0.0001$) were significant.

A significant two-way interaction was apparent between IRIEC scores and emotional conditions. Simple-slope analysis revealed that the 95% CI of the IRIEC slope was above zero under the positive condition (slope = 0.087, SE = 0.021, df = 91, 95% CI = [0.047, 0.128]) but included zero under the negative (slope = -0.023, SE = 0.024, df = 91, 95% CI = [-0.070, 0.025];

**Table 3. Statistical summary of the arousal ratings.**

**Fixed Effects**

| Effect | Beta | 95% CI | SE | df | t-value | Pr(>\|t\|) | Rsq | Rsq CI |
|---|---|---|---|---|---|---|---|---|
| Intercept | 5.527 | (5.390, 5.664) | 0.070 | 91.000 | 78.576 | < 0.001* | | |
| Emotion | 1.207 | (0.965, 1.450) | 0.125 | 91.000 | 9.693 | < 0.001* | 0.271 | (0.235, 0.307) |
| Presentation | 0.293 | (0.175, 0.412) | 0.061 | 91.000 | 4.837 | < 0.001* | 0.021 | (0.009, 0.038) |
| E * P | 0.154 | (0.041, 0.268) | 0.058 | 91.000 | 2.651 | 0.009* | 0.003 | (<0.001, 0.011) |
| IRIEC | 0.032 | (0.005, 0.059) | 0.014 | 91.000 | 2.327 | 0.022* | 0.013 | (0.004, 0.027) |
| IRIEC *E | 0.078 | (0.030, 0.126) | 0.025 | 91.000 | 3.153 | 0.002* | 0.038 | (0.021, 0.059) |
| IRIEC * P | 0.004 | (-0.020, 0.027) | 0.012 | 91.000 | 0.306 | 0.760 | <0.001 | (<0.001, 0.004) |
| IRIEC * E * P | 0.003 | (-0.020, 0.025) | 0.012 | 91.000 | 0.238 | 0.812 | <0.001 | (<0.001, 0.003) |
| AQ | 0.015 | (-0.006, 0.035) | 0.010 | 91.000 | 1.400 | 0.165 | 0.005 | (<0.001, 0.014) |
| AQ * E | 0.013 | (-0.023, 0.048) | 0.018 | 91.000 | 0.648 | 0.496 | 0.002 | (<0.001, 0.009) |
| AQ * P | 0.003 | (-0.015, 0.020) | 0.009 | 91.000 | 0.292 | 0.771 | <0.001 | (<0.001, 0.004) |
| AQ * E * P | -0.004 | (-0.020, 0.013) | 0.009 | 91.000 | -0.422 | 0.674 | <0.001 | (<0.001, 0.004) |

**Random Effects**

| Group | Effect | Variance | SD | 95% CI | Corr. I. | 95% CI | Corr. E. | 95% CI | Corr. P. | 95% CI |
|---|---|---|---|---|---|---|---|---|---|---|
| Subject | Intercept | 0.422 | 0.650 | (0.547, 0.751) | | | | | | |
| | E | 1.372 | 1.171 | (0.994, 1.349) | -0.17 | (-0.377, 0.045) | | | | |
| | P | 0.259 | 0.509 | (0.410, 0.605) | -0.04 | (-0.276, 0.209) | -0.23 | (-0.449, 0.007) | | |
| | E * P | 0.145 | 0.381 | (0.245, 0.494) | 0.05 | (-0.262, 0.365) | 0.21 | (-0.105, 0.505) | 0.39 | (0.052, 0.709) |
| Residual | | 0.692 | 0.832 | (0.798, 0.867) | | | | | | |

Formula: Arousal ~ 1 + emotional_condition * presentation_condition * IRIEC + emotional_condition * presentation_condition * AQ + (1 + emotional_condition * presentation_condition | subject). Number of observations: 1504. Number of subjects: 94. Abbreviations: See Table 1 footnotes.

**Table 4. Simple effect contrasts of IRIEC-emotional condition interaction in arousal ratings.**

| IRIEC | Mean Positive | Mean Negative | Difference | SE | df | t-value | Pr(>|t|) |
|---|---|---|---|---|---|---|---|
| -3SD | 5.03 | 5.02 | 0.005 | 0.568 | 91 | 0.008 | 0.9933 |
| -2SD | 5.48 | 4.91 | 0.572 | 0.401 | 91 | 1.428 | 0.1567 |
| -SD | 5.93 | 4.79 | 1.140 | 0.252 | 91 | 4.526 | < .0001* |
| 0 | 6.38 | 4.67 | 1.707 | 0.176 | 91 | 9.693 | < .0001* |
| +SD | 6.83 | 4.56 | 2.275 | 0.252 | 91 | 9.033 | < .0001* |
| +2SD | 7.28 | 4.44 | 2.843 | 0.401 | 91 | 7.093 | < .0001* |
| +3SD | 7.73 | 4.32 | 3.410 | 0.568 | 91 | 6.004 | < .0001* |

Abbreviations: df: Satterthwaite approximations to the degrees of freedom; Other abbreviations: see Table 1 footnotes.

slope difference = 0.11, SE = 0.035, df = 91, t = 3.153, p = 0.0022*) condition. Simple-effect contrast analysis was performed comparing the positive and negative conditions at mean-centered IRIEC levels from -3 SD to 3 SD in steps of 1 SD, using the Šidák method to correct for multiple comparisons. Individuals with a middle-to-high trait emotional empathy (IRIEC levels of the mean minus one SD and above) tended to feel more aroused when observing positive stimuli than when observing negative stimuli (Table 4, Fig 4B); individuals with low trait empathic concern tended not to differ in terms of arousal when observing stimuli of different emotional conditions. Regarding data from the 50 participants of the behavioral study, robust estimation also showed a significant interaction between the IRIEC and the emotional conditions, but no interaction between emotional and presentation conditions (S4 Table).

## Facial EMG

**Zygomaticus major.** The final LME model of the ZM responses exhibited the same fixed effects as the rating models. The random effects included the random intercept and random slopes for emotional and presentation conditions over the random factor "subject," as well as the random intercept over the random factor "Type" (Table 5). Adding the random intercept over the factor "Type" did not improve the model fitting (model comparison p = 0.9242), but we included it to account for possible influence by the study type. We found significant interactions between the emotional conditions and presentation conditions. Simple-effect analysis according to the emotional condition showed that ZM reactions under the positive-live condition (mean = 0.070, SE = 0.022, df = 28.26, 95% CI = [0.025, 0.115]) were stronger than ZM reactions under the positive-video condition (mean = 0.053, SE = 0.020, df = 21.87, 95% CI = [0.011, 0.095]); mean difference = 0.018, SE = 0.007, df = 185, t = 2.581, p = 0.0106*); ZM reactions under the negative-live condition (mean = -0.004, SE = 0.009, df = 1.10, 95% CI = [-0.096, 0.088]) was no different from ZM reactions under the negative-video condition (mean = 0.006, SE = 0.009, df = 1.42, 95% CI = [-0.055, 0.068]; mean difference = 0.011 SE = 0.007, df = 190, t = 1.568, p = 0.1186) conditions. Simple-effect analysis according to presentation condition showed that the ZM reaction under the positive-live condition was stronger than the ZM reaction under the negative-live condition (mean difference = 0.074, SE = 0.020, df = 93.1, t = 3.698, p = 0.0004*); the ZM reaction under the positive-video condition was stronger than the ZM reaction under the negative-video condition (mean difference = 0.047, SE = 0.020, df = 93.0, t = 2.312, p = 0.0230*). Regarding data from the 50 participants of the behavioral study, robust estimation also showed a significant interaction between the emotional and presentation conditions (S6 Table).

**Corrugator supercilia.** The final LME model of the CS responses exhibited the same fixed effects as the rating model; the random effects included the random intercept and random

**Table 5. Statistical summary of zygomaticus major (ZM) responses.**

**Fixed Effects**

| Effect | Beta | 95% CI | SE | df | t-value | Pr(>\|t\|) | Rsq | Rsq CI |
|---|---|---|---|---|---|---|---|---|
| Intercept | 3.195e-02 | (0.008, 0.056) | 1.247e-02 | 4.047 | 2.563 | 0.062 | | |
| Emotion | 4.242e-02 | (0.015, 0.069) | 1.383e-02 | 85.49 | 3.067 | 0.003* | 0.013 | (0.010, 0.018) |
| Presentation | 2.294e-03 | (-0.005, 0.010) | 3.743e-03 | 74.10 | 0.613 | 0.542 | <0.001 | (<0.001, 0.001) |
| E * P | 1.380e-02 | (0.006, 0.022) | 4.166e-03 | 10540 | 3.312 | <0.001* | 0.001 | (<0.001, 0.002) |
| IRIEC | 6.399e-04 | (-0.004, 0.005) | 2.100e-03 | 85.01 | 0.305 | 0.761 | <0.001 | (<0.001, 0.001) |
| IRIEC * E | 6.131e-04 | (-0.005, 0.006) | 2.728e-03 | 85.59 | 0.225 | 0.823 | <0.001 | (<0.001, 0.001) |
| IRIEC * P | 6.346e-05 | (-0.001, 0.002) | 7.518e-04 | 80.81 | 0.084 | 0.933 | <0.001 | (<0.001, <0.001) |
| IRIEC * E * P | -3.264e-04 | (-0.002, 0.001) | 8.485e-04 | 10540 | -0.385 | 0.700 | <0.001 | (<0.001, 0.001) |
| AQ | 1.166e-03 | (-0.002, 0.004) | 1.571e-03 | 84.79 | 0.742 | 0.460 | 0.001 | (<0.001, 0.002) |
| AQ * E | -1.511e-03 | (-0.005, 0.002) | 2.043e-03 | 85.28 | -0.740 | 0.462 | 0.001 | (<0.001, 0.002) |
| AQ * P | -3.320e-04 | (-0.001, 0.0008) | 5.562e-04 | 76.41 | -0.597 | 0.552 | <0.001 | (<0.001, 0.001) |
| AQ * E * P | -9.896e-06 | (-0.001, 0.001) | 6.226e-04 | 10540 | -0.016 | 0.987 | <0.001 | (<0.001, <0.001) |

**Random Effects**

| Group | Effect | Variance | SD | 95% CI | Corr. I. | 95% CI | Corr. E. | 95% CI |
|---|---|---|---|---|---|---|---|---|
| Subject | Intercept | 0.010 | 0.100 | (0.0974, 0.0977) | | | | |
| | Emotion | 0.017 | 0.129 | (0.1272, 0.1276) | 0.91 | (0.882, 0.946) | | |
| | Presentation | 4.490e-04 | 0.021 | (0.020, 0.030) | 0.55 | (0.556, 0.558) | 0.76 | (0.427, 0.896) |
| Type | Intercept | 8.541e-05 | 0.009 | (0.00251, 0.00253) | | | | |
| Residual | | 0.046 | 0.215 | (0.2151, 0.2154) | | | | |

Formula: ZM ~ 1 + emotional_condition * presentation_condition * IRIEC + emotional_condition * presentation_condition * AQ + (1 + emotional_condition + presentation_condition | subject) + (1 | Type). Number of observations: 10,787. Number of subjects: 93. Abbreviations: See 1 Table footnotes.

slopes for emotional and presentation conditions over the factor "subject," as well as the random intercept over the factor "Type" (Table 6). Adding the random intercept over the factor "Type" did not improve the model fitting (model comparison $p$ = 1), but we included it to account for possible influence by the study type. After model diagnostics, 8,862 data points from 84 participants remained in the model. We found significant interactions between the emotional conditions and the presentation conditions. Simple-effect analysis according to the emotional condition showed that although the CS was more relaxed under positive-live conditions (mean = -0.029, SE = 0.008, df = 1.55, 95% CI = [-0.078, 0.019]) than under positive-video conditions (mean = -0.017, SE = 0.008, df = 1.34, 95% CI = [-0.075, 0.041]; difference = -0.013, SE = 0.004, df = 181, t = -3.420, $p$ = 0.0008*), there was no difference between the negative-live (mean = 0.0029, SE = 0.008, df = 1.27, 95% CI = [-0.060, 0.065]) and negative-video (mean = -0.001, SE = 0.008, df = 1.13, 95% CI = [-0.078, 0.075]; difference = 0.0039, SE = 0.004, df = 193, t = 1.077, $p$ = 0.2828) conditions. Simple-effect analysis according to the presentation condition revealed stronger CS contraction under the negative-live condition than under the positive-live condition (mean difference = 0.0321, SE = 0.005, df = 103, t = 6.523, $p$ < 0.0001*), and under the negative-video condition than under the positive-video condition (mean difference = 0.0162, SE = 0.005, df = 104, t = 3.279, $p$ = 0.0014*).

A significant two-way interaction was apparent between IRIEC scores and emotional conditions. Simple-slope analysis revealed that the 95% CI of the IRIEC slope was more negative under the positive condition (slope = -0.0011, SE = 0.0009, df = 78.3, 95% CI = [-0.003, 0.0006]) than under the negative (slope = 0.0010, SE = 0.0007, df = 87.0, 95% CI = [-0.0004, 0.002]; slope difference = 0.0021, SE = 0.0009, df = 76, t = 2.196, $p$ = 0.0311*) condition.

**Table 6. Statistical summary of corrugator responses.**

**Fixed Effects**

| Effect | Beta | 95% CI | SE | df | t-value | Pr(>\|t\|) | Rsq | Rsq CI |
|---|---|---|---|---|---|---|---|---|
| Intercept | -1.102e-02 | (-0.029, 0.007) | 7.596e-03 | 1.016 | -1.451 | 0.382 | | |
| Emotion | -1.718e-02 | (-0.023, -0.011) | 3.125e-03 | 67.31 | -5.496 | <0.001* | 0.013 | (0.008, 0.018) |
| Presentation | -2.984e-03 | (-0.007, 0.0008) | 1.988e-03 | 73.67 | -1.501 | 0.138 | <0.001 | (<0.001, 0.002) |
| E * P | -7.882e-03 | (-0.012, -0.004) | 2.179e-03 | 8641 | -3.617 | <0.001* | 0.001 | (<0.001, 0.003) |
| IRIEC | -1.841e-05 | (-0.001, 0.001) | 6.217e-04 | 83.77 | -0.030 | 0.976 | <0.001 | (<0.001, 0.001) |
| IRIEC * E | -1.465e-03 | (-0.003, -0.0002) | 6.671e-04 | 76.00 | -2.196 | 0.031* | 0.002 | (<0.001, 0.004) |
| IRIEC * P | -2.334e-05 | (-0.0009, 0.0008) | 4.417e-04 | 98.05 | -0.053 | 0.958 | <0.001 | (<0.001, 0.001) |
| IRIEC * E * P | 1.337e-04 | (-0.0009, 0.001) | 5.063e-04 | 8665 | 0.264 | 0.792 | <0.001 | (<0.001, 0.001) |
| AQ | 6.089e-04 | (-0.0003, 0.002) | 4.795e-03 | 79.82 | 1.270 | 0.208 | 0.001 | (<0.001, 0.003) |
| AQ * E | 1.035e-04 | (-0.0009, 0.001) | 5.108e-04 | 70.33 | 0.203 | 0.840 | <0.001 | (<0.001, 0.001) |
| AQ * P | -3.860e-04 | (-0.001, 0.0003) | 3.294e-04 | 82.37 | -1.172 | 0.245 | <0.001 | (<0.001, 0.001) |
| AQ * E * P | 3.541e-04 | (-0.0004, 0.001) | 3.690e-04 | 8642 | 0.960 | 0.327 | <0.001 | (<0.001, 0.001) |

**Random Effects**

| Group | Effect | Variance | SD | 95% CI | Corr. I. | 95% CI | Corr. I. | 95% CI |
|---|---|---|---|---|---|---|---|---|
| Subject | Intercept | 5.944e-04 | 0.024 | (0.020, 0.029) | | | | |
| | Emotion | 5.663e-04 | 0.024 | (0.018, 0.029) | 0.30 | (0.021, 0.561) | | |
| | Presentation | 1.093e-04 | 0.010 | (0.005, 0.015) | 0.50 | (0.131, 0.859) | 0.12 | (-0.370, 0.280) |
| Type | Intercept | 9.809e-05 | 0.010 | (0.000, 0.031) | | | | |
| Residual | | 0.010 | 0.102 | (0.101, 0.104) | | | | |

Formula: CS ~ 1 + emotional_condition * presentation_condition * IRIEC + emotional_condition * presentation_condition * AQ + (1 + emotional_condition + presentation_condition | subject) + (1 | Type). Number of observations: 8,862. Number of subjects: 84. Abbreviations: See Table 1 footnotes.

Simple-effect contrast analysis was performed comparing the positive and negative conditions at mean-centered IRIEC levels from -3 SD to 3 SD in steps of 1 SD, using the Šidák method to correct for multiple comparisons. Individuals with a middle-to-high trait emotional empathy (IRIEC levels of the mean minus SD and above) tended to relax CS more when observing positive stimuli than when observing negative stimuli (Table 7, Fig 4C); individuals with low trait empathic concern tended not to differ in terms of CS responses when observing stimuli of different emotional conditions. The IRIEC-emotional condition interaction was not significant in the absence of model diagnostics. However, the robust estimation showed the same significant result pattern as model diagnostics (S7 Table). Regarding data from the 50 participants of the behavioral study, robust estimation also showed a significant interaction between the

**Table 7. Simple effect contrasts of IRIEC-emotional condition interaction in corrugator responses.**

| IRIEC | Mean Positive | Mean Negative | Diff. | SE | df | t-value | Pr(>\|t\|) |
|---|---|---|---|---|---|---|---|
| -3SD | -0.00696 | -0.01476 | 0.00780 | 0.01541 | 73.8 | 0.506 | 0.6143 |
| -2SD | -0.01241 | -0.00950 | -0.00291 | 0.01083 | 72.5 | -0.269 | 0.7889 |
| -SD | -0.01786 | -0.00424 | -0.01361 | 0.00668 | 69.6 | -2.039 | 0.0453* |
| 0 | -0.02331 | 0.00102 | -0.02432 | 0.00442 | 67.3 | -5.503 | < .0001* |
| +SD | -0.02875 | 0.00627 | -0.03503 | 0.00648 | 74.4 | -5.404 | < .0001* |
| +2SD | -0.03420 | 0.01153 | -0.04573 | 0.01058 | 76.5 | -4.321 | < .0001* |
| +3SD | -0.03965 | 0.01679 | -0.05644 | 0.01515 | 76.7 | -3.725 | 0.0004* |

Abbreviations: df: Satterthwaite approximations to the degrees of freedom; Other abbreviations: see Table 1 footnotes.

emotional and presentation conditions, but not between IRIEC and the emotional conditions (S8 Table).

## Discussion

We investigated whether the design of live interaction modulated the effects of trait emotional empathy and autistic traits on spontaneous facial mimicry and emotional contagion. We expanded the sample size and used a more complex statistical model to test two- and three-way interactions between individual traits, emotional conditions, and presentation conditions. We observed significant interactions between emotional and presentation conditions in terms of valence and arousal ratings and the ZM and CS responses. Specifically, we observed differences between positive-live and positive-video conditions, but not always between negative-live and negative-video conditions. Live stimuli enhanced subjective emotion perceptions and spontaneous facial mimicry in young female participants. These results are consistent with previously reported findings using the same paradigm [13, 14].

Initially, we expected to find that during live interaction, the modulatory effects of trait emotional empathy or autistic traits on spontaneous mimicry and emotional contagion would be enhanced (in the form of three-way interactions among trait, presentation conditions, and emotional conditions, the four research questions). However, we did not find such a three-way interaction as supporting evidence. In addition, the effect sizes (partial R-squared) of three-way interactions appeared small (partial $r^2$ around or smaller than 0.02) [81], especially for ZM and CS responses (partial $r^2 < 0.001$, with the upper confidence interval at 0.001) and the confidence interval of effect sizes covered 0. Given that 61 participants are sufficient to accurately estimate small effect sizes, reaching the point of stability with an 80% level of confidence [79], and the narrow confidence interval of effect sizes we observed, we could conclude that live interactions are very unlikely to modulate the relationship between autistic traits/trait empathy and emotion contagion/spontaneous facial mimicry.

### Presentation conditions did not modulate the relationship between the trait emotional empathy and emotional contagion/facial mimicry

Irrespective of the presentation condition, we found an IRIEC × emotional condition interaction in valence and arousal ratings and CS responses (Tables 1, 3, and 6, Fig 4). However, such relationships were not modulated by the presentation conditions. Simple-effect contrast analysis along the IRIEC range implied enhanced emotional contagion among participants with higher trait empathic concern (Tables 2 and 4). This is consistent with the findings of Dimberg et al., whereby a high-emotional-empathy group rated happy and angry expressions as significantly happier and angrier, respectively, compared with a low-emotional-empathy group [7]. However, an association of higher trait empathic concern with lower subjective experiential arousal when viewing negative stimuli does not necessarily imply that emotional contagion is involved. One possible reason is that we presented the positive and negative stimuli in a pseudorandomized/alternating sequence. The significant simple effects of emotional conditions (positive-live vs. negative-live and positive-video vs. negative-video) on subjective arousal showed that negative stimuli, in contrast to positive stimuli, were less arousing or more calming, irrespective of presentation condition. The negative correlation between trait empathic concern and subjective arousal is thus possibly a consequence of enhanced subjective-arousal contrasts in individuals with high trait empathic concern participating in a study involving an event-related design, such as the present study. Including a neutral condition might clarify the nature of the arousal response to the negative stimuli. Other issues regarding the negative stimuli are discussed below.

Higher trait empathic concern was associated with stronger CS mimicry of smiling faces (Fig 4C). Our results are compatible with Sonnby-Borgström et al. [5, 6] and Dimberg et al. [7], who showed that high trait emotional empathy groups (identified using the Questionnaire Measure of Emotional Empathy [100]) exhibited more relaxed corrugator activity when viewing happy faces than when viewing angry faces (Table 7). A recent meta-analysis reported a weak but significant positive correlation between facial mimicry and trait emotional empathy ($r = 0.13$, p = 0.001) [42].

## Limitations and future directions

We did not replicate findings of autistic traits modulating CS mimicry, which would have been identified as two-way interactions involving AQ and emotional conditions of the CS responses. In Hermans et al. [51], 18 extremely high AQ participants and 16 extremely low AQ participants were selected from a larger sample of 366 individuals. The modulatory effects of traits on CS mimicry may be detectable by comparing extreme groups. Such effects may be absent concerning the autistic traits in an average population.

The negative expressions of the models (Fig 2) may have been interpreted as angry or sad, despite the instruction only to display anger. The facial expressions of East Asians tend to overlap among emotional categories [101], and the ambiguity of negative facial expressions probably reflects cultural variance when a study is not conducted in the West [102]. Irrespective of culture, in real-life situations, the associations between facial expressions and affective semantic labels become much less stereotypical and more context-sensitive [103]. Nevertheless, naïve Japanese participants rated the anger intensity of the negative stimuli significantly stronger than the corresponding sadness intensity. Future studies investigating facial-expression processing during live interactions should examine naturalistic scenarios that consider social contexts [104]; such designs could further clarify the roles and functions of facial expressions in communication [1].

The live performances were highly controlled to ensure we could confidently attribute the observed effects to the live interaction and avoid confounding factors (e.g., the extent to which facial expressions were engaging and spontaneous). We considered such controls necessary when directly comparing the live and prerecorded conditions, but the live interactions in the present study differed from real-world interactions. Notably, the present study did not completely overcome the issue of the "spectatorial gap" that motivated the "second-person social neuroscience" construct [55].

Rating trials were performed after passive viewing trials to avoid EMG artifacts and electrode detachment associated with head and body movements during ratings. Participants may have become adapted and desensitized to the facial stimuli when performing ratings. The sequential effect might have increased the barrier to detecting modulatory effects relevant to emotional contagion. Randomizing the testing sequence might help address this issue, i.e., rating trials, electrode attachment, then passive viewing trials.

Spontaneous facial mimicry is regulated by sociodemographic factors, including age, gender, and socioeconomic status [105, 106]. We recruited only women in their 20s to avoid any effects of age or gender on spontaneous facial mimicry, which limited the generalizability of our results. Gender differences in autism have been extensively researched [107]; our results may not be generalizable to men. The pattern of spontaneous facial mimicry appeared to differ based on the gender of the sender-perceiver dyads. For example, there is more smile mimicry in female perceiver/dyads than in male perceiver/dyads [77, 108]. Female perceivers showed stronger smile mimicry to male than female happy faces [109]. Other contextual factors of between-gender interactions, such as attractiveness, motivations to flirt, or extroverted traits,

would further modulate the pattern of mimicry and the patterns of trait-mimicry associations. We hypothesize that the underlying patterns and effect sizes for the three-way interactions to detect how the live effect modulated the trait-mimicry and trait-emotional contagion association would differ between different gender combinations of dyads. Hence, dyads other than female-female combinations should be investigated in separate studies. Further studies are also required to investigate the effects of other sociodemographic factors on spontaneous facial mimicry [110].

In conclusion, we replicated findings implying that higher trait emotional empathy is associated with greater emotional contagion and CS mimicry in both live and pre-recorded conditions, which suggested that previous findings regarding the relationship between emotional empathy and emotional contagion/spontaneous facial mimicry using videos and photos could be generalized to real-life interactions. Using real-life social contexts in psychological studies remained important [58, 65, 66] for examining previous findings and providing empirical evidence in studies of social interaction under a second-person approach [57].

## Supporting information

**S1 File. Zip file containing data tables and the R markdown file.**
(ZIP)

**S1 Table. Statistical summary of valence ratings with robust estimation fixed effects.**
(DOCX)

**S2 Table. Statistical summary of valence ratings of 50 participants with robust estimation.**
(DOCX)

**S3 Table. Statistical summary of arousal ratings with robust estimation fixed effects.**
(DOCX)

**S4 Table. Statistical summary of arousal ratings of 50 participants with robust estimation.**
(DOCX)

**S5 Table. Statistical summary of zygomaticus responses with robust estimation fixed effects.**
(DOCX)

**S6 Table. Statistical summary of zygomaticus responses of 50 participants with robust estimation.**
(DOCX)

**S7 Table. Statistical summary of corrugator responses with robust estimation fixed effects.**
(DOCX)

**S8 Table. Statistical summary of corrugator responses of 50 participants with robust estimation.**
(DOCX)

## Acknowledgments

The authors would like to thank Mami Fujikura, Rena Kato, Yuko Kuroda, Kazusa Minemoto, Mei Nakabayashi, and Masaru Usami (in alphabetical order of surname) for assistance with data collection.

## Author Contributions

**Conceptualization:** Chun-Ting Hsu, Wataru Sato, Sakiko Yoshikawa.

**Data curation:** Chun-Ting Hsu.

**Formal analysis:** Chun-Ting Hsu.

**Funding acquisition:** Wataru Sato, Sakiko Yoshikawa.

**Investigation:** Chun-Ting Hsu, Wataru Sato.

**Methodology:** Chun-Ting Hsu.

**Project administration:** Wataru Sato, Sakiko Yoshikawa.

**Supervision:** Wataru Sato, Sakiko Yoshikawa.

**Validation:** Wataru Sato.

**Visualization:** Chun-Ting Hsu.

**Writing – original draft:** Chun-Ting Hsu.

**Writing – review & editing:** Chun-Ting Hsu, Wataru Sato, Sakiko Yoshikawa.

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
