## [Decision Letter · Decision Letter 0]

3 Dec 2021

PONE-D-21-34896The Modulatory Effects of Empathic and Autistic Traits on Emotional and Facial Motor Responses are Enhanced in Live Social InteractionsPLOS ONE

Dear Dr. Hsu,

Thank you for submitting your manuscript to PLOS ONE. After careful consideration, we feel that it has merit but does not fully meet PLOS ONE’s publication criteria as it currently stands. Therefore, we invite you to submit a revised version of the manuscript that addresses the points raised during the review process.

We look forward to receiving your revised manuscript.

Kind regards,

Mariska E. Kret

Academic Editor

PLOS ONE

Journal Requirements:

5. We note that Figures 2 and 3 includes an image of a [patient / participant / in the study]. 

Reviewers' comments:

Reviewer's Responses to Questions

**Comments to the Author**

1. Is the manuscript technically sound, and do the data support the conclusions?

Reviewer #1: Partly

Reviewer #2: Partly

2. Has the statistical analysis been performed appropriately and rigorously? 

Reviewer #1: I Don't Know

Reviewer #2: Yes

3. Have the authors made all data underlying the findings in their manuscript fully available?

Reviewer #1: Yes

Reviewer #2: No

4. Is the manuscript presented in an intelligible fashion and written in standard English?

Reviewer #1: Yes

Reviewer #2: Yes

5. Review Comments to the Author

Reviewer #1: Thank you for this opportunity to review this manuscript, which describes a study that examined facial mimicry and subjective ratings of emotional valence and arousal towards pre-recorded and live facial expressions of happiness and anger. Moreover, individual differences in empathic and autistic traits were taken into account. Overall, the authors found that responses to emotional expressions were enhanced in live conditions, compared to pre-recorded conditions, especially for individuals with higher affective empathy. Whilst individuals with higher autistic traits showed stronger zygomaticus responses to pre-recorded conditions than to live conditions. This study used a paradigm that allows for testing in live situations, and provided valuable data about how the (spontaneous) responses towards different types of emotional stimuli may be modulated by different personal traits. However, I do have some concerns about the manuscript in its current form, which dampen my enthusiasm. I will elaborate in greater detail below.

Main concerns:

1. I am afraid the current Introduction and Discussion do not reflect a correct picture of autism. Autistic people were described as having deficits that lead to low empathy and low sensitivity to social-emotional stimuli. However, this is not true. Increasing studies have shown that autistic people have the motivation to make social contacts and need a sense of belonging; and that autistic individuals are as emotionally aroused – and sometimes over-aroused – as non-autistic individuals. It also has to be taken into account that very often empathy of autistic people is evaluated by non-autistic people in the literature. Autistic individuals may be different in how they express and react to emotions, but they do feel it. The authors are suggested to modify the language in this respect (e.g., deficits, lacking) and to review the literature also from a non-medical point of view, to make this study more connected to the lived experiences of autistic individuals, or individuals with high autistic traits.

2. The analyses were conducted with rather complex LME models. However, the model complexity was only assessed in terms of random effects, but on of fixed effects. I wonder if the authors have a reason for this decision, and if the authors did an a priori power analysis to know whether the observations are enough for running such complex models. Moreover, the models seem not well grounded by the research questions and hypotheses. The inclusion of valence and IRI-perspective taking came as a surprise and the authors did not have hypotheses for these analyses. Due to these doubts, I am uncertain to what extent the results can be interpreted in the current form.

3. The manuscript would benefit from a more in-depth discussion on the findings. Currently, the authors mainly described how the findings are in line with previous studies, and did not really give a holistic picture of the implications and contributions of this study. Also, the autistic traits were only found to have an effect on the ZM reactions, but not on subjectively ratings of emotional valence and arousal. This finding shows that the participants with high autistic traits in this study felt similarly emotional and aroused as those with low autistic traits, echoing the point made earlier in comment 1. This is a result not expected by the authors, and thus worth further discussions.

Other concerns:

4. Page 9, 2nd paragraph, last line: Better add ‘autonomic’ before ‘facial mimicry’ given that voluntary facial mimicry has been reported to be present in autistic individuals.

5. Page 10, line 3 “than suggested by previous data”: This could be confusing. On which basis were the previous studies compared?

6. Page 10, 2nd paragraph “we expected that individuals with higher emotional empathy … would exhibit higher levels of emotional contagion and greater spontaneous facial mimicry of live stimuli”: From this hypothesis, it is unclear what ‘emotion contagion’ is referred to. Is it referring to the subjective ratings, or denoting the latent layer of facial mimicry?

7. Figure 2: This figure presents only video-recorded stimuli; this should better be specify. If possible, adding examples of live stimuli could be helpful.

8. Figure 3: It appears that participants were presented ‘Video’ or ‘Real Person’ before each stimulus was shown to them. I believe this design requires better explanations: Wouldn’t such instructions already prime the participants’ responses? I wonder whether the authors expect this design to have an effect on the results.

9. Page 12 – autism spectrum quotient: the last sentence is already in Statistical Analyses and not needed in this section.

10. Page 13 – paradigm and procedures: Throughout the Method section, there are many different versions of trial numbers. After reading multiple times, I understood that sometimes only part of the task was referred to. Yet it still remains unclear to me how many trials were presented for participants to rate valence and arousal, and whether these ratings were given right after each passive viewing trial, or in different blocks. It would be better if the authors could describe the procedures altogether in one section, rather than separating them in different parts as they are now.

11. Page 14 – statistical analyses “… and the difference in EMG (ZM and CS) between the neutral (0-1 s after stimulus onset) and maximal phases (2.5-3.5 s after stimulus onset) of the dynamic facial expression”: I understand that the baseline had already be deducted, and it comprised of the 3 seconds before the onset and 1 second after the onset, which is different from the description shown here.

12. Results – questionnaire scores: The total IRI score was reported, but not the IRI-Empathy Concern score, which should be the score reported here as it is part of the research question.

13. Page 23, line 4 “As expected, participants with higher trait emotional empathy (IRIEC) showed larger differences in subjective arousal between positive and negative facial expressions in the live compared to video conditions”: However, no hypothesis was made in regard to the difference in valence.

14. Page 23, 2nd paragraph: the sudden switch to cognitive empathy seems irrelevant to the topic discussed in this paragraph.

15. Page 24, lines 4-5 “watching eye effect and audience effect”: A brief explanation on these effects may help more readers understand the information presented here.

16. It should be noted and addressed that many effects found in this study were with rather small effect sizes, such as the interaction effect of AQ x Presentation Mode on ZM reactions. This again makes me concerned about the complexity of the models and the interpretation of the results.

Reviewer #2: In the current study, the authors aim to examine whether the “liveliness” of social encounters, i.e. live interactions vs. video presentations, have an effect on the modulation of the subjective emotional experience as well as facial mimicry by individual differences in empathic traits and autistic traits in a healthy female sample. To achieve this, participants were seated in front of a self-designed and previously validated live image relay system and were either presented with live emotional facial expressions (positive vs. negative) of a model seated on the other side, or with video-taped expressions of the same model. Replicating previous results, arousal ratings as well as facial mimicry (indicated by facial muscle (de-)activations) were higher/more pronounced in live interactions compared to video observations. Additionally, higher scores on the Empathic Concern subscale of the IRI were associated a stronger difference in corrugator supercilii between positive and negative expressions responses across presentation conditions, and with a stronger difference in arousal ratings as well as zygomaticus major responses between positive and negative expressions in the live vs. the video condition, which the authors interpreted as a higher emotional engagement with the other, especially in live interactions. Higher autistic traits, however, were associated with higher zygomaticus responses in the video compared to the live condition which was suggested to originate from the decreased sensitivity to social stimuli which is commonly associated with autism. The core strength of this study is its innovative setup which allows for a comparison between live and video-taped ‘interactions’ while all other experimental parameters are kept constant. It is a great example of how an incremental shift to a ‘second-person neuroscience’ can be made possible. Further, by measuring both subjective experiences and physiological responses to emotional expressions, different levels of analysis can be described and compared to achieve a more holistic view on modulations in emotion processing associated with different trait dimensions.

Since no page and line numbers were provided, I started numbering the pages, with the title page being page 1. All remarks can further be mapped by an indication of the paragraph in the respective section.

Major concerns:

Main general remarks: The authors claim that they are providing a more ecologically valid and naturalistic approach to study real-life interactions with their setup. The displayed expressions in the interaction were, however, highly posed and controlled rather than naturally occurring. The question therefore arises whether participants would have perceived a difference between live and videotaped expressions if no instructions were given, and, if not, what the additional value of setting up this ‘real’ live interaction was. While the setup can definitely be useful for research on social interactions, the authors should clarify why they decided for such a controlled design and in how far this decision limits the ecological validity compared to a real, uncontrolled social interaction with bidirectional information exchange. This limitation is briefly mentioned in the discussion but should be elaborated and also kept in mind when making claims about examining “live social interactions” in the current study.

Further, in the abstract, the introduction, the results and the discussion, the authors phrase their hypotheses/results in a way that a group comparison between individuals with low trait level and high trait levels (both traits) would be expected (e.g. abstract: “female individuals with low AQ scores showed […], with the opposite pattern seen in high AQ female individuals”). As the authors treat the trait dimensions as continuous variables in their analyses, the phrasing should be adjusted, e.g. “higher autistic trait levels were associated with …”.

Introduction: It would be beneficial to clarify the definition of constructs which the authors are using and to be consistent in their usage. For example, that “emotional empathy” would be “reflected in EMG activities” and correlate “with self-reported feelings” in the second paragraph (p.3) seems to suggest that the emotional experience would not be part of emotional empathy. Similarly, a clarification of the operationalization of emotional contagion in the last paragraph of the discussion (“individuals with higher emotional empathy […] would exhibit higher levels of emotional contagion”, p.4) would benefit to understand the author’s hypotheses. A more precise usage of terms should also be considered in the discussion.

Materials and methods: In the caption of Figure 2, it is described that the models were instructed to make “happy” and “angry” expressions whereas, in all other parts of the manuscript, the expressions are only referred to as positive/”smiling” or negative/”frowning”. Given the ambiguous examples in Figure 2 and the fact that “frowning”/activation of the corrugator supercilii is also associated with sad facial expressions, it would be a valuable information to know whether the expressions were indeed interpreted as “happy” and “angry” (and not only rated according to their valence and arousal as described in the Live performance validation section). Given that anger and sadness can have different effects with regard to mimicry and emotional empathy, the interpretation of the expressions seems to be crucial for this study.

Results: It was quite striking to me that, when comparing estimated means for interactions including emotion and presentation (and potentially also a trait dimension), only comparisons within a conditions but never between presentation conditions (e.g. positive-live vs positive-video) were reported, even though these comparisons seemed to be of main interest to the study. By only comparing slope differences between emotions within presentation conditions, it remains unclear whether the effect might be driven by one emotional expression. This seems especially relevant for the EMG data if meaningful inferences about “mimicry” are aimed to be made as the authors actually do in the second paragraph of the discussion (IRIEC and facial mimicry of ZM). In the Zygomaticus major analysis, the statistics for the comparison between the AQ slopes in the video vs. the live condition are not reported. Here, the interpretation that “Individuals with high AQ scores had stronger ZM reactions to video compared to live stimuli, while individuals with low AQ scores had stronger ZM reactions to live compared to video stimuli” seems inadmissible since the slope in the video condition only seems to be steeper in the video conditions (stronger effect of autistic traits) but no group comparison has been made. The results should therefore also be discussed more carefully in the discussion.

Discussion: As there was no emotion-specific modulatory effect of autistic traits on ZM activity, it should be clarified that the current studies results are actually not directly supporting the cited literature on facial mimicry (not “compatible”). General differences in zygomaticus activity depending on the context should rather be the focus of this discussion. Further, as the slope of the video condition seems to be, at least numerically, more affected by autistic traits than the slope of the live condition, a pure interpretation in terms of less zygomaticus activity with higher AQ traits in a live context appears to be foreshortening.

Minor comments:

Introduction: The authors provide a good amount of literature supporting the role of facial mimicry in emotion processing and social behaviour in the first paragraph of the introduction (p.3). As the role of facial mimicry in emotion recognition is however still under debate (see recent meta-analysis by Holland and colleagues), the introduction would benefit from a more critical/less one-sided description. Further, as the authors only investigate the role of autistic traits and do not compare a clinical population to healthy controls, some information on the relationship between autistic trait levels and facial mimicry/empathy should be included.

Materials and methods: The motivation for some influential experimental design choices is not clear, namely:

- Pre-recorded videos/paradigm and procedure: Why did the participants have a 3min conversation before the task? Why was it necessary to include practice trials for the passive viewing task at all and a comparably large number of practice trials for the rating task? Why was there a break after 32 trials (and not 30) in the passive viewing task and why did participants only rate such a number of trials (16) in the rating task compared to the passive viewing task (60)?

- EMG pre-processing: Why is a baseline of 4s selected, incl. very different visual inputs, i.e. a fixation cross, the instructions and part of the stimulus?

- Statistical analyses of live interaction data: Why is was a difference between the neutral and maximal phase in the EMG signals calculated if a baseline correction already took place which also, among others, included the neutral phase?

Results: While the authors state that “Autism spectrum disorder (ASD) is characterized by low empathy” in the introduction (second paragraph on p.3) and repeat this again in the discussion, they could not find any significant correlations between the trait dimensions (Questionnaire scores section). How can these results be explained?

Discussion: There are some unclarities in the discussion of the results related to empathic traits (third paragraph of the discussion section, p.17). First, it could be clarified how the current study’s results relate to the cited literature on mimicry and cognitive vs. emotional empathy, as well as how the ratings data is in line with Dimberg et al.’s study (only briefly mentioned). Further, the explanation with the role of the amygdala in upper facial feature representation seems a bit far-fetched, also given that the other cited studies (Sonnby-Bergström et al., Dimberg et al.) could find effects (or at least tendencies in Sonnby-Bergström et al.) for the corrugator muscle. Moreover, a restructuring of this section, starting with a comparison to previous studies on autistic traits and mimicry and then the more general discussion of ASD and responses to social stimuli in different contexts would allow for a better readability. In the limitation section, the potential influence of only using female participants could be discussed and the meaning behind the possibility “to more vividly observe the effect of social top-down regulation” should be clarified. It is further not clearly described why future studies should include “interactive activities”. Last but not least, the authors close their discussion with a circular argument namely that live interactions are needed to investigate individual differences and that individual differences should be controlled for in live interactions (similar to their abstract). As their main goal is to promote a “second-person neuroscience” to describe individual differences and they also focus on that throughout their paper, a focus on the benefits of the live context might be more appropriate.

Additional comments:

• Paragraph 4 of the introduction, p.4: “The participants viewed videotaped and live-relay performances alternatively.” This sentence only repeated the information from the previous sentence and can be removed.

• Paragraph 3 of the introduction, p.4: “more emotional reactions and facial mimicry than suggested by previous data.” Which previous data is referred to and in how far was the study design different?

• Figure 1 caption: “to the to the prompter”

• Second paragraph of Paradigms and procedure on p.7: As participants rated valence and arousal separately, it is on really an “affective grid rating”

• Results/Statistical analyses: The inclusion of supplemental analyses without “outlier” removal is a nice addition, especially given the to-be-expected non-normal EMG response distribution. I was personally wondering why the authors decided to cope with this problem by simply removing high EMG values. It might be a nice addition to have the benefits and potential pitfalls discussed in a few sentences so that the results without removal can be contextualized in a better way. Further, it would be helpful for the reader to not only have the number of excluded trials mentioned in the text but also the number of remaining trials for each analysis.

• Third paragraph in discussion: The sentence “The level of spontaneous mimicry was related to the individual differences in empathy” seems like an unnecessary repetition.

• Discussion first paragraph: The overall summary could be a bit less broad and more to the point/specific to the study.

• Third paragraph of the discussion on page 18: many concepts like “we-mode”, watching eye effect,... are introduced without any explanation

• As there are some small language mistakes in the manuscript, independent editorial help before publishing should be considered

6. PLOS authors have the option to publish the peer review history of their article (what does this mean?). If published, this will include your full peer review and any attached files.

Reviewer #1: No

Reviewer #2: **Yes: **Milica Nikolić & Julia Folz

---

## [Author Response · Author response to Decision Letter 0]

17 Jan 2022

Reviewer #1: 

Thank you for this opportunity to review this manuscript, which describes a study that examined facial mimicry and subjective ratings of emotional valence and arousal towards pre-recorded and live facial expressions of happiness and anger. Moreover, individual differences in empathic and autistic traits were taken into account. Overall, the authors found that responses to emotional expressions were enhanced in live conditions, compared to pre-recorded conditions, especially for individuals with higher affective empathy. Whilst individuals with higher autistic traits showed stronger zygomaticus responses to pre-recorded conditions than to live conditions. This study used a paradigm that allows for testing in live situations, and provided valuable data about how the (spontaneous) responses towards different types of emotional stimuli may be modulated by different personal traits. However, I do have some concerns about the manuscript in its current form, which dampen my enthusiasm. I will elaborate in greater detail below.

Main concerns:

1. I am afraid the current Introduction and Discussion do not reflect a correct picture of autism. Autistic people were described as having deficits that lead to low empathy and low sensitivity to social-emotional stimuli. However, this is not true. Increasing studies have shown that autistic people have the motivation to make social contacts and need a sense of belonging; and that autistic individuals are as emotionally aroused – and sometimes over-aroused – as non-autistic individuals. It also has to be taken into account that very often empathy of autistic people is evaluated by non-autistic people in the literature. Autistic individuals may be different in how they express and react to emotions, but they do feel it. The authors are suggested to modify the language in this respect (e.g., deficits, lacking) and to review the literature also from a non-medical point of view, to make this study more connected to the lived experiences of autistic individuals, or individuals with high autistic traits.

Authors’ response: We are grateful to the reviewer for highlighting this issue. We have updated the Introduction (lines 59–81) thoroughly (particularly the paragraphs that discuss autistic traits and ASD; lines 499–557) for a more neutral discussion of early findings, and have included more contemporary theories concerning social cognition in individuals with ASD or highly autistic traits.

2. The analyses were conducted with rather complex LME models. However, the model complexity was only assessed in terms of random effects, but on of fixed effects. I wonder if the authors have a reason for this decision, and if the authors did an a priori power analysis to know whether the observations are enough for running such complex models. Moreover, the models seem not well grounded by the research questions and hypotheses. The inclusion of valence and IRI-perspective taking came as a surprise and the authors did not have hypotheses for these analyses. Due to these doubts, I am uncertain to what extent the results can be interpreted in the current form.

Authors’ response: We will address the reviewer’s concerns via several bullet points.

-The existing literature on the complexity of LME has focused on random effects in models; for n variance components, there will be a maximum of n(n+1)/2 model parameters, while fixed-effect parameters usually do not drastically increase the degree of freedom. There is also a debate as to whether the parsimonious [1] or keep-it-maximal [2] approach is best. We adopted the parsimonious approach for random effects, and the keep-it-maximal approach for fixed effects.

-We considered both valence and arousal ratings (as dependent variables) to be measures of emotional contagion. Note that excluding valence will reduce the number of models, but not model complexity.

-We originally included IRI perspective-taking (IRIPT) in light of Chartrand et al.’s report on the utility of IRIPT, but not IRIEC, for predicting the chameleon effect (spontaneous motor mimicry) [3]. They conducted a between-group comparison based on a median split of IRIPT and IRIEC. As pointed out by Reviewer 1 (comment 14), this appears inconsistent with later studies, which found that individual differences in emotional empathy were more associated with spontaneous facial mimicry. It is also the case that IRIPT yielded no significant results in our study. Therefore, we excluded IRIPT and its interaction terms from the models and updated the results. We have also added a note in the Method section, entitled “Statistical analysis of live interaction data,” to the effect that “Note that we initially included the perspective taking subscale of IRI and its interaction terms, but dropped these from the final models because neither the main nor interaction effects involving this subscale were significant” (lines 255–257). Note that after simplifying the models, IRIEC and AQ appeared to have three-way interaction effects on valence ratings.

-We did not perform a priori power analysis, due to a lack of previous reports of effect sizes for interaction effects in the context of the “live effect”. However, we have now included power estimates for significant results as a reference for future studies. The data table in the supplementary material may also be used for sample size estimation by future studies. As expected, a different sample size was required to achieve a power of 0.8 for each effect, and at the time of data acquisition, 50 was the realistic limit of the sample size that we could achieve.

-Regarding the complexity of the statistical models, no standard method of determining whether an LME model includes too many parameters exists. We have relied on “warnings” of singular convergence or convergence failure, i.e., that the number of parameters to be estimated was too large for the dataset or reflected overfitting. In our iterative model comparison process, we did not select models leading to singular convergence or convergence failure (https://rpubs.com/palday/lme4-singular-convergence).

3. The manuscript would benefit from a more in-depth discussion on the findings. Currently, the authors mainly described how the findings are in line with previous studies, and did not really give a holistic picture of the implications and contributions of this study. Also, the autistic traits were only found to have an effect on the ZM reactions, but not on subjectively ratings of emotional valence and arousal. This finding shows that the participants with high autistic traits in this study felt similarly emotional and aroused as those with low autistic traits, echoing the point made earlier in comment 1. This is a result not expected by the authors, and thus worth further discussions.

Authors’ response: We are grateful to the reviewer for highlighting this issue. In the revised manuscript, we have updated and extended the discussion, and highlight the implications of our results in the context of contemporary understanding of ASD and autistic traits. Indeed, as the reviewer has pointed out, our results were more compatible with recent findings that individuals with ASD or high AQs do not differ from those with low AQs with respect to emotional contagion and spontaneous facial mimicry. Our findings regarding the effect of AQ appeared to be more specific to the live effect. We now discuss this point in the discussion (lines 499–541).We have also added a paragraph addressing the potential drawback of exclusively examining the participants’ autistic traits, and not the dyadic differences (lines 542–557).

Other concerns:

4. Page 9, 2nd paragraph, last line: Better add ‘autonomic’ before ‘facial mimicry’ given that voluntary facial mimicry has been reported to be present in autistic individuals.

Authors’ response: We have followed the reviewer’s suggestion.

5. Page 10, line 3 “than suggested by previous data”: This could be confusing. On which basis were the previous studies compared?

Authors’ response: We agree with the reviewer that this expression was vague and have changed it to “than using prerecorded photos or videos.”

6. Page 10, 2nd paragraph “we expected that individuals with higher emotional empathy … would exhibit higher levels of emotional contagion and greater spontaneous facial mimicry of live stimuli”: From this hypothesis, it is unclear what ‘emotion contagion’ is referred to. Is it referring to the subjective ratings, or denoting the latent layer of facial mimicry?

Authors’ response: We used subjective ratings as measures of emotional contagion. This information has been added to the paragraph in question, which now reads “Based on previous findings, we expected dynamic live expressions to enhance the modulatory effects of empathic and autistic traits. Specifically, we expected the live conditions to enhance the correlations between emotional empathy, as measured using the empathic concern subscale of the IRI, and emotional contagion, measured by the subjective valence and arousal ratings, as well as the positive correlation between emotional empathy and the level of spontaneous facial mimicry” (line 99-104). In this manuscript, we treat emotional contagion and facial mimicry separately, since it remains unclear whether there is a causal relationship between facial mimicry and emotional contagion, despite their frequent co-occurrence [4].

7. Figure 2: This figure presents only video-recorded stimuli; this should better be specify. If possible, adding examples of live stimuli could be helpful.

Authors’ response: We have now included examples of both prerecorded and live stimuli in Figure 2.

8. Figure 3: It appears that participants were presented ‘Video’ or ‘Real Person’ before each stimulus was shown to them. I believe this design requires better explanations: Wouldn’t such instructions already prime the participants’ responses? I wonder whether the authors expect this design to have an effect on the results.

Authors’ response: The live effect that this study aimed to investigate is the effect of the knowledge that live images are being transmitted (audience effect, watching eye effect, or “we mode”). In an earlier publication [5], the validating rating study showed prerecorded and live performance clips to naïve participants without such instruction, and the ratings showed no “live effect”. This effect was anticipated.

9. Page 12 – autism spectrum quotient: the last sentence is already in Statistical Analyses and not needed in this section.

Authors’ response: We have followed the reviewer’s suggestion and removed the sentence.

10. Page 13 – paradigm and procedures: Throughout the Method section, there are many different versions of trial numbers. After reading multiple times, I understood that sometimes only part of the task was referred to. Yet it still remains unclear to me how many trials were presented for participants to rate valence and arousal, and whether these ratings were given right after each passive viewing trial, or in different blocks. It would be better if the authors could describe the procedures altogether in one section, rather than separating them in different parts as they are now.

Authors’ response: The procedure was conducted sequentially, as follows: informed consent ->3-minute conversation -> 8 practice passive viewing trials -> 60 passive viewing trials -> 8 practice rating trials -> 16 rating trials. We recorded EMG data only in the passive viewing trials, because participants made movements (keyboard presses) during rating trials. In rating trials, participants first viewed the dynamic facial expression, exactly as in the passive viewing trials, and then gave valence and arousal ratings (Fig. 1). We have added sentences to the first paragraph (lines 187–190) of the “paradigm and procedures” to clarify the overall procedure sequence.

11. Page 14 – statistical analyses “… and the difference in EMG (ZM and CS) between the neutral (0-1 s after stimulus onset) and maximal phases (2.5-3.5 s after stimulus onset) of the dynamic facial expression”: I understand that the baseline had already be deducted, and it comprised of the 3 seconds before the onset and 1 second after the onset, which is different from the description shown here.

Authors’ response: The EMG data preprocessing included a “baseline correction” step prior to rectification and log transformation. The baseline for this step was the mean of the EMG values from 3 seconds before to 1 second after stimulus onset (i.e., 2 seconds before and after the onset of the trial). All data points in the trial/epoch (as EEGLAB terms it) were subtracted from this value to correct for low-frequency drifts in continuous recordings, and were then rectified and log-transformed. The dependent variable in the statistical analysis was the difference between the neutral and maximal phases of dynamic facial expression after rectification and log transformation. No conflict emerged between the description of the EMG preprocessing and statistical analyses.

12. Results – questionnaire scores: The total IRI score was reported, but not the IRI-Empathy Concern score, which should be the score reported here as it is part of the research question.

Authors’ response: We have modified this paragraph and reported the descriptive statistics of the IRIEC scores, as suggested.

13. Page 23, line 4 “As expected, participants with higher trait emotional empathy (IRIEC) showed larger differences in subjective arousal between positive and negative facial expressions in the live compared to video conditions”: However, no hypothesis was made in regard to the difference in valence.

Authors’ response: We expected the level of emotional contagion in the live conditions to be positively correlated with the IRIEC. We used subjective ratings of both valence and arousal [6], to measure the level of emotional contagion. In the Results section of the first submission, an interaction effect of trait measures on arousal ratings, but not on valence ratings, was reported; this discrepancy could have been discussed in greater depth. However, in this revised version, we have removed the parameters for IRIPT and their interaction effects from the models, based on comment 3 of Reviewer 1, and a significant three-way interaction effect of IRIEC on valence ratings emerged, similar to that seen on arousal ratings. The results appear to be more coherent in terms of emotion contagion.

14. Page 23, 2nd paragraph: the sudden switch to cognitive empathy seems irrelevant to the topic discussed in this paragraph.

Authors’ response: The literature review concerning the relationship between empathy and spontaneous motor mimicry (not limited to facial expressions) revealed Chartrand et al.’s report of the relationship between IRIPT (but not IRIEC) and the chameleon effect, as early evidence [3]. However, although it did not seem relevant to the literature reported in the first version of the manuscript, most other recent studies reported that emotional empathy was more influential with respect to spontaneous facial mimicry, including the reports cited in the current version of the manuscript. Considering Reviewer 1’s concern about the model’s complexity (comment 2) and non-significance of IRIPT in our data, we have decided to remove IRIPT from the statistical model, and have also removed sentences relating to Chartrand et al.’s study.

15. Page 24, lines 4-5 “watching eye effect and audience effect”: A brief explanation on these effects may help more readers understand the information presented here.

Authors’ response: We have now included short descriptions of “we mode,” the “audience effect,” and the “watching eye effect” in the Discussion (lines 492–498).

16. It should be noted and addressed that many effects found in this study were with rather small effect sizes, such as the interaction effect of AQ x Presentation Mode on ZM reactions. This again makes me concerned about the complexity of the models and the interpretation of the results.

Authors’ response: We have added this point to the Limitations section (lines 589-591).

Reviewer #2

In the current study, the authors aim to examine whether the “liveliness” of social encounters, i.e. live interactions vs. video presentations, have an effect on the modulation of the subjective emotional experience as well as facial mimicry by individual differences in empathic traits and autistic traits in a healthy female sample. To achieve this, participants were seated in front of a self-designed and previously validated live image relay system and were either presented with live emotional facial expressions (positive vs. negative) of a model seated on the other side, or with video-taped expressions of the same model. Replicating previous results, arousal ratings as well as facial mimicry (indicated by facial muscle (de-)activations) were higher/more pronounced in live interactions compared to video observations. Additionally, higher scores on the Empathic Concern subscale of the IRI were associated a stronger difference in corrugator supercilii between positive and negative expressions responses across presentation conditions, and with a stronger difference in arousal ratings as well as zygomaticus major responses between positive and negative expressions in the live vs. the video condition, which the authors interpreted as a higher emotional engagement with the other, especially in live interactions. Higher autistic traits, however, were associated with higher zygomaticus responses in the video compared to the live condition which was suggested to originate from the decreased sensitivity to social stimuli which is commonly associated with autism. The core strength of this study is its innovative setup which allows for a comparison between live and video-taped ‘interactions’ while all other experimental parameters are kept constant. It is a great example of how an incremental shift to a ‘second-person neuroscience’ can be made possible. Further, by measuring both subjective experiences and physiological responses to emotional expressions, different levels of analysis can be described and compared to achieve a more holistic view on modulations in emotion processing associated with different trait dimensions.

Since no page and line numbers were provided, I started numbering the pages, with the title page being page 1. All remarks can further be mapped by an indication of the paragraph in the respective section.

Major concerns:

1. Main general remarks: The authors claim that they are providing a more ecologically valid and naturalistic approach to study real-life interactions with their setup. The displayed expressions in the interaction were, however, highly posed and controlled rather than naturally occurring. The question therefore arises whether participants would have perceived a difference between live and videotaped expressions if no instructions were given, and, if not, what the additional value of setting up this ‘real’ live interaction was. While the setup can definitely be useful for research on social interactions, the authors should clarify why they decided for such a controlled design and in how far this decision limits the ecological validity compared to a real, uncontrolled social interaction with bidirectional information exchange. This limitation is briefly mentioned in the discussion but should be elaborated and also kept in mind when making claims about examining “live social interactions” in the current study.

Authors’ response: In the 2020 paper, which used some of the data included in the current manuscript, the post-hoc valence and arousal ratings of naïve participants demonstrated that, without instructions, participants perceived no differences between live and videotaped expressions [5]. As Reviewer 2 described above, this allowed for more confident attribution of observed effects to the live effect (we-mode, audience effect, or watching-eye effect). As it was difficult for us to prepare videotaped and live facial expressions that were simultaneously natural and comparable, we decided to opt for the classical paradigm of spontaneous facial mimicry, and asked the models to make reproducible facial expressions, but worked on the facilities to enable the “live” conditions. We agree that real social interactions rarely occurred, such that the present study represented an incremental move toward live interactions. We have further elaborated on this issue in the Limitations section in the Discussion (lines 578–589), and hope to perform future studies with realistic bidirectional interactions using our setup.

2. Further, in the abstract, the introduction, the results and the discussion, the authors phrase their hypotheses/results in a way that a group comparison between individuals with low trait level and high trait levels (both traits) would be expected (e.g. abstract: “female individuals with low AQ scores showed […], with the opposite pattern seen in high AQ female individuals”). As the authors treat the trait dimensions as continuous variables in their analyses, the phrasing should be adjusted, e.g. “higher autistic trait levels were associated with …”.

Authors’ response: We agree with the reviewer that the description should match the analysis performed. Regarding the existing studies described in the Introduction and Discussion, most of them performed between-group comparisons (ASD vs. neurotypical or median/mean split of trait levels), so we retained the description of between-group comparisons. In describing our current results, we deemed it more appropriate to refer to the trait as a continuous variable. We have thoroughly checked and corrected the phrasing as necessary.

3. Introduction: It would be beneficial to clarify the definition of constructs which the authors are using and to be consistent in their usage. For example, that “emotional empathy” would be “reflected in EMG activities” and correlate “with self-reported feelings” in the second paragraph (p.3) seems to suggest that the emotional experience would not be part of emotional empathy. Similarly, a clarification of the operationalization of emotional contagion in the last paragraph of the discussion (“individuals with higher emotional empathy […] would exhibit higher levels of emotional contagion”, p.4) would benefit to understand the author’s hypotheses. A more precise usage of terms should also be considered in the discussion.

Authors’ response: We are grateful to the reviewer for highlighting this issue. The relationship between spontaneous facial mimicry, emotional empathy and emotion contagion is complicated and not without controversy. We have further elaborated our position about this issue in the second paragraph of the Introduction (lines 41–50). We have also added background information about empathy and emotional contagion to the third paragraph. Spontaneous mimicry, in the context of embodied cognition, is proposed as a mechanism underlying emotional empathy. Both spontaneous mimicry and empathy have been proposed as mechanisms of emotional contagion (i.e., the transfer of affective states between people), but whether the relationship is causal remains the subject of debate [4,7,8]. Emotional contagion has also been proposed as a component of empathy. The emotional contagion process itself is also highly complicated, involving multilevel and parallel redundant mechanisms [9,10]. After consideration, we think that it might be clearer to treat these three concepts as co-occurring yet independent processes, despite the possibility that one may be in a causal relationship with another. Specifically, we examined how individual differences in emotional empathy correlated with different patterns of spontaneous mimicry and emotional contagion (as reflected in valence and arousal ratings).

4. Materials and methods: In the caption of Figure 2, it is described that the models were instructed to make “happy” and “angry” expressions whereas, in all other parts of the manuscript, the expressions are only referred to as positive/”smiling” or negative/”frowning”. Given the ambiguous examples in Figure 2 and the fact that “frowning”/activation of the corrugator supercilii is also associated with sad facial expressions, it would be a valuable information to know whether the expressions were indeed interpreted as “happy” and “angry” (and not only rated according to their valence and arousal as described in the Live performance validation section). Given that anger and sadness can have different effects with regard to mimicry and emotional empathy, the interpretation of the expressions seems to be crucial for this study.

Authors’ response: We performed a post-hoc rating study of negative video clips (two per presentation condition—prerecorded or live—for each model) representing sadness and anger on a Likert scale of 1–5, with 28 naïve female participants (mean ± SD age = 27.64 ± 3.07 years; range: 20–30 years). Both negative stimuli, i.e., anger (mean ± SD = 3.36 ± 0.61) and sadness (mean ± SD = 3.01 ± 0.85), were rated highly but the paired t-test of participant-wise mean rating values indicated that participants generally experienced more anger than sadness as a result of the negative stimuli (t = 2.55, df = 27, p = 0.017). It is also true that participants detected a certain amount of sadness in our negative stimuli. Our EMG results showed no modulatory effect of the live effect on individual differences in the CS responses, suggesting that the level of perceived anger may have evoked more social top-down control over facial mimicry. The post-hoc rating results correspond well with findings that East Asian facial expressions are less stereotypical and exhibit greater overlap between categories [11]. We instructed the models to make their own angry expressions, and did not ask them to pose/reproduce Ekman’s anger expression [12]. We made this decision based on evidence presented in Ekman’s study in the 1970s demonstrating that Western angry faces were only recognized by around 67% of Japanese participants [13], and based on more recent investigations from our lab [14]. 

We have added the post-hoc rating results to the Methods section under the heading “Stimuli validations” (lines 224–233), as well as an additional paragraph to the Discussion (lines 558–577).

5. Results: It was quite striking to me that, when comparing estimated means for interactions including emotion and presentation (and potentially also a trait dimension), only comparisons within [emotion] conditions but never between presentation conditions (e.g. positive-live vs positive-video) were reported, even though these comparisons seemed to be of main interest to the study. By only comparing slope differences between emotions within presentation conditions, it remains unclear whether the effect might be driven by one emotional expression. This seems especially relevant for the EMG data if meaningful inferences about “mimicry” are aimed to be made as the authors actually do in the second paragraph of the discussion (IRIEC and facial mimicry of ZM). In the Zygomaticus major analysis, the statistics for the comparison between the AQ slopes in the video vs. the live condition are not reported. Here, the interpretation that “Individuals with high AQ scores had stronger ZM reactions to video compared to live stimuli, while individuals with low AQ scores had stronger ZM reactions to live compared to video stimuli” seems inadmissible since the slope in the video condition only seems to be steeper in the video conditions (stronger effect of autistic traits) but no group comparison has been made. The results should therefore also be discussed more carefully in the discussion.

Authors’ response: We are grateful to the reviewers for highlighting both of these issues. For the interaction effects involving both emotion and presentation conditions, we have added the simple effects of emotion conditions by presentation condition (positive-live vs. negative-live, positive video vs. negative-video) to the Results section. We have also modified the description of this AQ x presentation condition interaction effect; it is now described merely in terms of there being a more positive correlation between AQ and ZM responses in the video than live conditions.

6. Discussion: As there was no emotion-specific modulatory effect of autistic traits on ZM activity, it should be clarified that the current studies results are actually not directly supporting the cited literature on facial mimicry (not “compatible”). General differences in zygomaticus activity depending on the context should rather be the focus of this discussion. Further, as the slope of the video condition seems to be, at least numerically, more affected by autistic traits than the slope of the live condition, a pure interpretation in terms of less zygomaticus activity with higher AQ traits in a live context appears to be foreshortening.

Authors’ response: We agree with the reviewers. Considering the above comment together with comment 1 of Reviewer 1, we have modified the background information and discussion pertaining to autistic traits and ASD. We have also clearly stated in the Discussion that our results revealed no reduction effect of AQ on facial mimicry, and have related the AQ x presentation condition interaction effect on ZM responses to a reduction in social smiling in live conditions.

Minor comments:

7. Introduction: The authors provide a good amount of literature supporting the role of facial mimicry in emotion processing and social behaviour in the first paragraph of the introduction (p.3). As the role of facial mimicry in emotion recognition is however still under debate (see recent meta-analysis by Holland and colleagues), the introduction would benefit from a more critical/less one-sided description. Further, as the authors only investigate the role of autistic traits and do not compare a clinical population to healthy controls, some information on the relationship between autistic trait levels and facial mimicry/empathy should be included.

Authors’ response: 

-We agree that the evidence supports a correlational relationship among spontaneous facial mimicry, emotional cognition, and contagion, and that causality remains unclear. We have included this information in the Introduction (lines 41–50).

-We have also included a paragraph providing background information on the relationship between autistic traits and spontaneous facial mimicry in the Introduction (lines 73–81). To the best of our knowledge, the amount of literature pertaining to the effects of autistic traits in neurotypical participants is relatively less than that comparing ASD between neurotypical groups. We have also discussed possible reasons why, contrary to previous findings, we did not find a modulatory effect of AQ on spontaneous facial mimicry.

8. Materials and methods: The motivation for some influential experimental design choices is not clear, namely:

- Pre-recorded videos/paradigm and procedure: Why did the participants have a 3min conversation before the task? Why was it necessary to include practice trials for the passive viewing task at all and a comparably large number of practice trials for the rating task? Why was there a break after 32 trials (and not 30) in the passive viewing task and why did participants only rate such a number of trials (16) in the rating task compared to the passive viewing task (60)?

Authors’ response: 

-The 3-minute conversation demonstrated to the participants that the prompter system actually transmitted live images of the model, as described (i.e., confirmed that we were not deceiving them). We have added this information under the “Paradigm and procedures” part of the Methods section (lines 187–188). 

-The practice trials aimed to give participants an idea of what would happen, and to allow them to ask questions before we officially commenced data collection. In passive viewing practice trials, we sometimes had to reaffirm to participants that they were required simply to focus on the screen without performing any other task. In the rating practice trials, participants sometimes required further clarification regarding the meaning of valence and arousal, or how the Likert scale should be used. We agree that the passive viewing task was simpler than the rating task, and the number of practice trials was determined arbitrarily as we designed the experiment.

-The passive viewing task was interrupted after 32 trials to ensure a balanced number of trials (n = 8) per condition; we have included this information in the Method section (line 200). The pseudo-random presentation of conditions took this into consideration.

-The relative proportions of EMG trials (passive viewing) and rating trials confirmed our prior assumption about the effect size and signal-to-noise ratio of spontaneous facial mimicry and ratings. The EMG would have a smaller effect size and lower signal-to-noise ratio than ratings, thus requiring significantly more repetitions per participant. However, participants might experience a loss of attention or interest if the passive viewing period persists for too long. Ultimately, 60 EMG trials (15 repetitions) and 16 ratings trials (4 repetitions) allowed us to restrict the session length to within 1 hour, which was the amount of time we remunerated the participants for.

9. - EMG pre-processing: Why is a baseline of 4s selected, incl. very different visual inputs, i.e., a fixation cross, the instructions and part of the stimulus?

- Statistical analyses of live interaction data: Why was a difference between the neutral and maximal phase in the EMG signals calculated if a baseline correction already took place which also, among others, included the neutral phase?

Authors’ response: The “baseline correction” at the preprocessing stage aimed to address the issue of low-frequency drift over a long recording period. For the entire epoch/trial, we believe it should be around the level of the trial’s beginning (i.e., the onset of the instructions), so we subtracted the average value 2 seconds before and after that from all data points. The rectification and log transformation steps took place after the baseline correction, before we extracted the dependent variable. Different dependent variables were extracted for the different analyses (i.e., analyses of the various phases of dynamic facial expressions).

We are confident that the differences between the mean values of the neutral phase (1 s of data) and maximal phase (1 s of data) (which represents the experience of looking at the same face with different expressions) are adequate to make inferences regarding facial mimicry. Since the preprocessing baseline correction value would have differed slightly from the mean value for the neutral phase of dynamic facial expressions, we calculated the exact values for the epochs. Using only the mean value of the neutral phase for preprocessing baseline correction may have been sufficient for the present analysis. However, owing to the oscillating nature of the EMG data in terms of amplitude) and the issue that we aimed to address (low-frequency drift), we do not believe that the result would have differed significantly post-rectification and -log transformation if the preprocessing baseline had the same period as the psychological baseline. No current guidelines recommend preprocessing the data differently for each individual analysis.

10. Results: While the authors state that “Autism spectrum disorder (ASD) is characterized by low empathy” in the introduction (second paragraph on p.3) and repeat this again in the discussion, they could not find any significant correlations between the trait dimensions (Questionnaire scores section). How can these results be explained?

Authors’ response: As noted in our response to Reviewer 1’s comment 1, we have revised our description of ASD and autistic traits in the Introduction (lines 59–81) and Discussion (lines 499–557). Note that after IRIPT is removed from the statistical models, there appears to be an AQ × emotion × presentation condition interaction effect on valence ratings, which also does not support the idea that ASD is characterized by low empathy.

11. Discussion: There are some unclarities in the discussion of the results related to empathic traits (third paragraph of the discussion section, p.17). First, it could be clarified how the current study’s results relate to the cited literature on mimicry and cognitive vs. emotional empathy, as well as how the ratings data is in line with Dimberg et al.’s study (only briefly mentioned). Further, the explanation with the role of the amygdala in upper facial feature representation seems a bit far-fetched, also given that the other cited studies (Sonnby-Bergström et al., Dimberg et al.) could find effects (or at least tendencies in Sonnby-Bergström et al.) for the corrugator muscle. Moreover, a restructuring of this section, starting with a comparison to previous studies on autistic traits and mimicry and then the more general discussion of ASD and responses to social stimuli in different contexts would allow for a better readability. In the limitation section, the potential influence of only using female participants could be discussed and the meaning behind the possibility “to more vividly observe the effect of social top-down regulation” should be clarified. It is further not clearly described why future studies should include “interactive activities”. Last but not least, the authors close their discussion with a circular argument namely that live interactions are needed to investigate individual differences and that individual differences should be controlled for in live interactions (similar to their abstract). As their main goal is to promote a “second-person neuroscience” to describe individual differences and they also focus on that throughout their paper, a focus on the benefits of the live context might be more appropriate.

Authors’ response:

-We have included a description of how Dimberg et al.’s (2011) rating results related to our IRIEC × emotion × presentation condition interaction effects on valence and arousal ratings (lines 488–490).

-We agree that the role of the amygdala in upper facial expressions is far-fetched and have removed the text in question.

-To address previous comments, we have rewritten the Discussion section according to the autistic trait results. We now emphasize that we found no negative correlation between AQ and the level of mimicry in general, and then focus on the fact that the live effect had different effects on valence ratings and ZM responses across the AQ spectrum. 

-We have included the point that the distribution of autistic traits differs considerably between males and females, in relation to the study limitation that only female participants were included.

-We have modified the sentence in the Limitations section as follows: “Spontaneous facial mimicry is regulated by sociodemographic factors such as age, gender, and socioeconomic status. Further studies are required to investigate the top-down regulatory effects of sociodemographic factors on spontaneous facial mimicry” (lines 594–597).

-We agree that the final sentences in the Conclusion are somewhat underwhelming and seemingly circular, so have removed the last two sentences from the Conclusion to focus on the benefit of the second-person approach.

Additional comments:

12. Paragraph 4 of the introduction, p.4: “The participants viewed videotaped and live-relay performances alternatively.” This sentence only repeated the information from the previous sentence and can be removed.

Authors’ response: We have followed this suggestion and removed this sentence.

13. Paragraph 3 of the introduction, p.4: “more emotional reactions and facial mimicry than suggested by previous data.” Which previous data is referred to and in how far was the study design different?

Authors’ response: As per our response to comment 5 of Reviewer 1, we agree that this expression is vague. By “previous data”, we meant the results obtained using prerecorded stimuli and have changed the text to “than using prerecorded photos or videos”.

14. Figure 1 caption: “to the to the prompter”

Authors’ response: We are grateful for the reviewers for pointing out this error and have corrected it.

15. Second paragraph of Paradigms and procedure on p.7: As participants rated valence and arousal separately, it is on really an “affective grid rating”

Authors’ response: We agree with the reviewers that this is not an affective grid rating, because we separated the valence and arousal ratings. We now refer to the ratings as valence and arousal ratings.

16. Results/Statistical analyses: The inclusion of supplemental analyses without “outlier” removal is a nice addition, especially given the to-be-expected non-normal EMG response distribution. I was personally wondering why the authors decided to cope with this problem by simply removing high EMG values. It might be a nice addition to have the benefits and potential pitfalls discussed in a few sentences so that the results without removal can be contextualized in a better way. Further, it would be helpful for the reader to not only have the number of excluded trials mentioned in the text but also the number of remaining trials for each analysis.

Authors’ response: 

-We would like to emphasize that, in LME model analysis, the outliers/influential data points do not necessarily denote extreme values (on initial inspection). Such concerns arose from the a priori assumption of conditional normality for regression analysis, including random effects and residuals/error terms. The outliers/influential data points identified in our LME model analysis were based on residual variance. The normality assumption can be fulfilled even if the (joint) marginal distribution of the dependent variable is not normal.

-The question of whether influential data points should be removed is still subject to debate, and we have received contradictory suggestions on this issue. Some authors suggest that all data should be treated according to Bakker and Wicherts’ work [15], which only considered the independent-samples t test (which assumes a marginal normal data distribution). I It is also unclear whether we could make reliable inferences based on fixed effects if the normality assumption of the LME models was violated. Essentially, we followed Baayen and Milin’s [16] approach to the exclusion of outliers (which they termed “model criticism”) but identified such data points using residual diagnostics [17]. We are not in a position to suggest the optimal solution to this issue, and have presented results with and without model diagnostics to allow the reader to determine which fixed effects are most robust to assumption violations. Furthermore, we have provided a data table in the supplementary material so that the data can be reanalyzed in future in the event that concrete guidelines are proposed. We have provided more context regarding this issue in the Method section (lines 272–278).

-We have followed the reviewer’s suggestion and report the number of remaining data points and participants in the Results section for the final models.

17. Third paragraph in discussion: The sentence “The level of spontaneous mimicry was related to the individual differences in empathy” seems like an unnecessary repetition.

Authors’ response: We agree with the reviewer and have removed the sentence.

18. Discussion first paragraph: The overall summary could be a bit less broad and more to the point/specific to the study.

Authors’ response: The first paragraph of the discussion now describes only the fact that the finding in our previous report was maintained; we describe new findings involving individual differences in the following paragraphs.

19. Third paragraph of the discussion on page 18: many concepts like “we-mode”, watching eye effect,... are introduced without any explanation

Authors’ response: We have followed the reviewers’ suggestion and included short descriptions of these concepts (lines 493–498).

20. As there are some small language mistakes in the manuscript, independent editorial help before publishing should be considered

Authors’ response: We are grateful to the reviewers for highlighting this issue. We have sent the revised manuscript for proofreading in advance of resubmission.

Bibliography

1. Bates D, Kliegl R, Vasishth S, Baayen H. Parsimonious Mixed Models. arXiv:150604967 [stat]. 2018 [cited 27 Jul 2020]. Available: http://arxiv.org/abs/1506.04967

2. Barr DJ, Levy R, Scheepers C, Tily HJ. Random effects structure for confirmatory hypothesis testing: Keep it maximal. Journal of Memory and Language. 2013;68: 255–278. doi:10.1016/j.jml.2012.11.001

3. Chartrand TL, Bargh JA. The chameleon effect: the perception-behavior link and social interaction. J Pers Soc Psychol. 1999;76: 893–910. 

4. Olszanowski M, Wróbel M, Hess U. Mimicking and sharing emotions: a re-examination of the link between facial mimicry and emotional contagion. Cognition and Emotion. 2020;34: 367–376. doi:10.1080/02699931.2019.1611543

5. Hsu C-T, Sato W, Yoshikawa S. Enhanced emotional and motor responses to live versus videotaped dynamic facial expressions. Scientific Reports. 2020;10: 16825. doi:10.1038/s41598-020-73826-2

6. Russell JA, Mehrabian A. Evidence for a three-factor theory of emotions. Journal of Research in Personality. 1977;11: 273–294. doi:10.1016/0092-6566(77)90037-X

7. Hess U, Blairy S. Facial mimicry and emotional contagion to dynamic emotional facial expressions and their influence on decoding accuracy. Int J Psychophysiol. 2001;40: 129–141. doi:Doi 10.1016/S0167-8760(00)00161-6

8. van der Schalk J, Fischer A, Doosje B, Wigboldus D, Hawk S, Rotteveel M, et al. Convergent and divergent responses to emotional displays of ingroup and outgroup. Emotion. 2011;11: 286–298. doi:10.1037/a0022582

9. Hatfield E, Bensman L, Thornton PD, Rapson RL. New Perspectives on Emotional Contagion: A Review of Classic and Recent Research on Facial Mimicry and Contagion. Interpersona. 2014;8: 159–179. doi:10.5964/ijpr.v8i2.162

10. Elfenbein HA. The many faces of emotional contagion. Organizational Psychology Review. 2014;4: 326–362. doi:10.1177/2041386614542889

11. Jack RE, Garrod OGB, Yu H, Caldara R, Schyns PG. Facial expressions of emotion are not culturally universal. Proceedings of the National Academy of Sciences. 2012;109: 7241–7244. doi:10.1073/pnas.1200155109

12. Ekman P, Friesen WV. Pictures of Facial Affect. Palo Alto, CA: Consulting Psychologists Press; 1976. 

13. Ekman P, Friesen WV, O’Sullivan M, Chan A, Diacoyanni-Tarlatzis I. Universals and Cultural Differences in the Judgments of Facial Expressions of Emotion. Nebraska symposium on motivation. Lincoln, NE: University of Nebraska Press; 1972. pp. 207–283. 

14. Sato W, Hyniewska S, Minemoto K, Yoshikawa S. Facial Expressions of Basic Emotions in Japanese Laypeople. Front Psychol. 2019;10: 259. doi:10.3389/fpsyg.2019.00259

15. Bakker M, Wicherts JM. Outlier removal, sum scores, and the inflation of the type I error rate in independent samples t tests: The power of alternatives and recommendations. Psychological Methods. 2014;19: 409–427. doi:10.1037/met0000014

16. Baayen RH, Milin P. Analyzing reaction times. Int j psychol res. 2010;3: 12–28. doi:10.21500/20112084.807

17. Loy A, Hofmann H. Diagnostic tools for hierarchical linear models: Diagnostic tools for hierarchical linear models. WIREs Comp Stat. 2013;5: 48–61. doi:10.1002/wics.1238

---

## [Decision Letter · Decision Letter 1]

30 May 2022

PONE-D-21-34896R1The Modulatory Effects of Empathic and Autistic Traits on Emotional and Facial Motor Responses are Enhanced in Live Social Interactions

PLOS ONE

Dear Dr. Hsu,

Thank you for submitting your manuscript to PLOS ONE. As you will see, I have received two reviews from experts in the field. While they both appreciated the substantial edits you have made, we all believe that additional major revisions are warranted to meet PLOS ONE’s publication criteria.

While I invite you to carefully consider the comments from both reviewers, I would note that both were concerned with the interpretation of some of your findings. I agree that this is a significant concern and will need to be addressed before publication can be considered. Both reviewers provide several suggestions to this end. Additionally, it was noted that there are potential problems with your statistics. Please ensure that all necessary models are reported to justify your conclusions. Additionally, I agree that there is no need to report observed power. These calculations provide no additional information beyond p values and will need to be removed. Finally, some of the wording used was confusing or repetitive. Please ensure the readability of the manuscript throughout.

Given this, we invite you to submit a revised version of the manuscript that addresses the points raised during the review process. Please submit your revised manuscript by Jul 14 2022 11:59PM. If you will need more time than this to complete your revisions, please reply to this message or contact the journal office at plosone@plos.org. When you're ready to submit your revision, log on to https://www.editorialmanager.com/pone/ and select the 'Submissions Needing Revision' folder to locate your manuscript file.

Please include the following items when submitting your revised manuscript:A rebuttal letter that responds to each point raised by the academic editor and reviewer(s). You should upload this letter as a separate file labeled 'Response to Reviewers'.A marked-up copy of your manuscript that highlights changes made to the original version. You should upload this as a separate file labeled 'Revised Manuscript with Track Changes'.An unmarked version of your revised paper without tracked changes. You should upload this as a separate file labeled 'Manuscript'.

We look forward to receiving your revised manuscript.

Kind regards,

Eric J. Moody, Ph.D.

Academic Editor

PLOS ONE

Reviewers' comments:

Reviewer's Responses to Questions

**Comments to the Author**

1. If the authors have adequately addressed your comments raised in a previous round of review and you feel that this manuscript is now acceptable for publication, you may indicate that here to bypass the “Comments to the Author” section, enter your conflict of interest statement in the “Confidential to Editor” section, and submit your "Accept" recommendation.

Reviewer #1: (No Response)

Reviewer #2: (No Response)

2. Is the manuscript technically sound, and do the data support the conclusions?

Reviewer #1: Partly

Reviewer #2: Partly

3. Has the statistical analysis been performed appropriately and rigorously? 

Reviewer #1: Yes

Reviewer #2: No

4. Have the authors made all data underlying the findings in their manuscript fully available?

Reviewer #1: Yes

Reviewer #2: Yes

5. Is the manuscript presented in an intelligible fashion and written in standard English?

Reviewer #1: No

Reviewer #2: Yes

6. Review Comments to the Author

Reviewer #1: I greatly appreciate the extensive changes the authors made to the manuscript. The study design is now much clearer. However, I still have some lingering concerns regarding the revised manuscript:

Introduction

1. The key terms trait emotional empathy, spontaneous mimicry, and emotional contagion require clearer definitions and should be used consistently throughout the manuscript. First, although the authors referred to “emotional empathy”, what they measured was actually “empathic concerns” i.e., “warmth, compassion, and concern toward others” (line 161-162). This is not entirely the same as the affective/emotional empathy, which involves sharing others’ emotional states, thus emotional contagion; whilst empathic concerns require not only sharing others’ emotions, but also prosocial motivation and sympathy/compassion. Second, as a result of the first point, the hypotheses were not well grounded by the literature reviewed by the authors in Introduction, which was about emotional empathy. Third, line 44-46 “Although both spontaneous mimicry and empathy have been proposed as mechanisms facilitating emotional contagion, it has also been argued that spontaneous mimicry and emotional contagion are more likely to simply co-occur” is confusing and could also be misleading. I would suggest the authors to check the Hatfield and colleagues’ chapter for the role of emotional contagion, and also the Prochazkova & Kret (2017) study for mimicry and emotional contagion.

Hatﬁeld, E., Rapson, R. L., & Le, Y. C. L. (2011). Emotional contagion and empathy. The Social Neuroscience of Empathy, 19.

Prochazkova, E., & Kret, M. E. (2017). Connecting minds and sharing emotions through mimicry: A neurocognitive model of emotional contagion. Neuroscience & Biobehavioral Reviews, 80, 99-114.

2. The study aimed to examine the effects of a) empathic concerns and b) autistic traits on i) emotional contagion and ii) spontaneous facial mimicry. But not all the hypotheses for these relations were clearly outlined. For example, the hypothesis for the relation between autistic traits and emotional contagion is missing.

Methods

3. Some parts of the methodology remain unclear. For example, it was mentioned in line 153-155 that “Fifteen smiling and fifteen frowning clips were used for the passive viewing component; two per condition were used for the practice ratings, and four per condition were used for the actual ratings.” Only until reading later text did I realize that the 15+15 trials described here did not include the practice trials, and that the “condition” refers to the 2 (presentation) x 2 (valence) design.

4. Also, in line 195 “they were shown the instructions for either “Video” or “Real Person””, a previous comment on this was not addressed. The participants already knew beforehand whether the expression presented to them was pre-recorded or live. It’d be helpful if the authors can explain why it was designed like this and how this could affect the outcomes.

5. The rating scales for valence and arousal were shown in the figure, but were only very vaguely and confusingly described in the text that a scale of 1 to 9 was used and a question “please rate the emotion that you feel” was asked. Yet this question is rather broad and vague, and how the scale 1 to 9 should be interpreted is unclear. This also applies to line 226.

6. Regarding line 251-253 “the mean value difference in EMG (ZM and CS) between the neutral (0-1 s after stimulus onset) and maximal phases (2.5-3.5 s after stimulus onset) of the dynamic facial expression”. This treatment remains unclear to me. It was described earlier that baseline-corrected values would be use, while here the difference between neutral and maximal phases was said to be used. Taken together, the authors are suggested to carefully check throughout their Methods section for a clearer presentation of their design.

7. Reporting observed power is not appropriate, and also not meaningful, as it is basically a reflection of the p values. Instead, the authors are encouraged to provide an indication for how their effect sizes should be interpreted, which would be helpful for understanding the results.

Results

8. Overall, the results are not very easy to follow. Oftentimes, it was not clearly presented what were being compared, and how one condition differs from the other. Below are some examples.

9. Line 323-324 “During the live stimuli observations, a high IRIEC level was associated with a greater discrepancy in valence ratings for positive vs. negative stimuli”: Better to also describe what is the outcome in the video condition.

10. Line 336 “The AQ slope difference … was more negative”: What does “more negative” mean?

11. Line 398-399 “Although the AQ slopes for ZM reactions were not different from zero in neither live nor video conditions, the slope was significantly more positive in the video”: First, what does "more positive" mean? Second, which interaction effect was referred to?

12. The term “positive” and “negative” were being used for three purposes in the Results section: for describing valence, for indicating the direction of a slope, and for indicating the direction of a difference. Thus, the text is rather confusing. The authors are suggested to consider using different sentence structures to avoid using the same term for different contexts (e.g., say “higher A was related to higher B” instead of “A and B were positively related”).

13. Sometimes both fixed and random effects in the final model were listed, while sometimes only random effects were outlined. Consider listing all fixed and random effects when reporting final models, or simply removing such sentences and referring to the tables, for clarity.

Discussion

14. Discussion on the effect of empathic concerns: I am not sure I am convinced by the authors’ explanation. Results showed that participants with high EC were more sensitive to the differences between different emotional expressions (positive or negative), and a live condition enhanced this effect. This finding possibly shows that the subtle signals conveyed during live interactions and that knowing the interaction partner is real, is helpful for feeling and recognizing the emotion of the partner, especially for those with higher levels of empathic concerns, who are high in sympathy and prosocial motivation. However, note that the results about arousal rating is not entirely in line with the hypotheses (higher EC was related to lower arousal ratings when the expression was negative), and thus also requires further explanations.

15. Discussion on the effect of autistic traits: I am not sure that I understand the authors’ argument that a weaker emotion contagion is relate to reduced social attention. In people with higher AQ, positive expressions were rated more positive, and negative ones rated more negative, than people with lower AQ, and this effect was even stronger in video condition than in live condition. Possibly, hypersensitivity, which is often part of the autistic traits, may play a role here, and when in live condition, they can make more "balanced" judgement. Or, it could be that the video condition, i.e., a more controlled condition, helps people with high AQ to differentiate the expressions.

16. The effect of AQ on ZM reaction: Results showed that, first, among people with lower AQ, ZM reaction in live condition was stronger than in video condition; while among people with higher AQ, no differences were between conditions in ZM reaction. Second, in video condition, higher AQ was related to stronger ZM reaction, while this association was weaker in live condition. These suggest that people with lower AQ reacted with more smiles (not necessarily mimicry as they also did so for negative expressions) when they knew the partner is real than in pre-recorded condition. But for people with higher AQ, whether in the video condition or in the live condition, they smiled to the extent similar to people with lower AQ in the live condition. In other words, participants with higher AQ did not reduce smiling in live condition, but actually increased smiling in video condition, compared to people with lower AQ. Therefore, the authors’ interpretation does not seem to hold as participants with higher AQ did not reduce their social smiling in live condition. Also, given that there is no standard for what is adequate ZM reaction, no “reduction” can be assumed.

These two papers could be helpful in interpreting empathic responses in autistic individuals:

Rieffe, C., O’Connor, R. A. G., Bülow, A., Willems. D., Hull, L., Sedgewick, F., Stockmann, L., & Blijd-Hoogewys, E. (2021). Quantity and quality of empathic responding by autistic and non-autistic adolescent girls and boys. Autism, 25(1), 199-209.

O'Connor, R. A. G., Stockmann, L., & Rieffe, C. (2019). Spontaneous helping behavior of autistic and non-autistic (pre-)adolescents: A matter of motivation? Autism Research,12, 1796-1804.

Other issues related to autism

17. Line 60 “Facial expressions of individuals with ASD are considered less natural”: The autistic people express emotions in their "natural" way. It is not fair to compare the naturalness of facial expressions, and to use non-autistic people as the standard.

18. Line 62 “Spontaneous facial mimicry was also found to be reduced and delayed in the ASD population”: compared to whom?

19. Line 69 “individuals with ASD showed less social smiling and mimicry of positive facial expressions”: compared to whom?

20. Line 75 “Hermans et al. showed reduced spontaneous facial mimicry of the CS response to static angry faces in females with a high autism spectrum quotient (AQ)”: compared to whom?

Other issues related to clarity and readability

21. Line 34 “Dimberg was the first to demonstrate similar muscle activation patterns when seeing happy faces…”: Similar to what? Also this sentence is not easy to follow.

22. Line 40 “increased liking for the mimicker, and facilitated prosocial behavior”: the behavior of whom?

23. Line 49 “…and investigated the interaction effect of trait differences and live interactions…”: It will be clearer if empathy and autistic trait are mentioned here as part of the aim. Trait differences is a very broad term.

24. Line 63 “autonomic” should better be “spontaneous”, which is what was consistently used throughout the text.

25. Line 71-72 “Recent evidence indicated that there is no difference in emotional empathy or emotion contagion between the ASD and neurotypical population”: This sentence does not connect well to the rest of the paragraph. It is in-between the paragraphs on facial mimicry. Therefore, hard to follow and grasp what is the message here. Consider moving it to the beginning of the paragraph, or to the end of the next paragraph.

26. Line 101-104: This is an important sentence about the hypotheses, but unfortunately very hard to read.

27. Line 165 “i.e., those with and without autism”: Better change “autism” to “a diagnosis of autism” for clarity.

28. Line 247 “Statistical analyses of live interaction data”: why only live data is mentioned here, while both presentation modes are actually in the models?

29. The authors are suggest to ensure consistent use of terms throughout the manuscript. For example, currently “valence”, “emotion”, and “emotion categories” were used interchangeably.

30. The manuscript would benefit from English language editing.

Reviewer #2: Overall, it is evident that the authors acknowledged and implemented the feedback. There are, however, still some points that require further improvement.

Major concern:

Results: It is appreciated that the authors implemented the suggested comparison between presentation conditions (e.g. positive-live vs. positive-video) to their Results section. However, I was puzzled by the interpretations that were drawn from the examination of slope differences. First, even though some slope differences did not reach significance (e.g. IRIEC slope difference in live vs. video for both arousal ratings of positive and negative stimuli, lines 357 – 361), the authors still interpreted this as e.g. more/less arousing perception of the stimuli in the live condition. Secondly, they described slope differences to be greater in one condition vs. the other (e.g. IRIEC slope difference in positive vs. neutral in the live vs video condition, lines 353-357) even though no statistical test supporting this claim is reported, i.e. they were most probably only referring to numerical values or graphs. Thirdly, while slope differences might be informative to compare between presentation conditions, they do not yield any information on whether there is indeed a significant association between e.g. the IRIEC and arousal ratings in a specific condition. As the coding of the predictors is not specified in the model description (e.g. whether they are sum coded or which level acts as a reference level), it is impossible to retrieve this information from the model tables. As simplest solution, I would suggest to report whether the individual slopes for each combination of the factor levels, e.g. positive-live, are significantly different from 0 in the three-way interaction with each of the trait variables and, as a next step, to look at slope differences. Only the significant findings should then be further interpreted which should, if necessary, lead to adjustments in the Abstract, the Result section and the Discussion section. The outlined concerns also apply to the reporting of the results in the Supporting Information. Lastly, the addition of a post-hoc power analysis is highly criticized and I would recommend to exclude it (see: https://daniellakens.blogspot.com/2014/12/observed-power-and-what-to-do-if-your.html;
https://data.library.virginia.edu/post-hoc-power-calculations-are-not-useful/;
https://link.springer.com/article/10.1007/s12144-018-0018-1).

Minor concerns:

Introduction: The introduction benefits from the clarification of concepts and addition of research on autistic traits. The authors might reconsider starting the first paragraph in the introduction with the same words as the abstract.

Methods (Stimuli validation) + Discussion: It was great to see that the authors put a lot of effort in examining whether the negative/angry expressions could potentially be confused with sad expressions. As this was not one of the main goals of the study, I was surprised to find a quite long paragraph on it in the Discussion section. Despite the fact that this paragraph contains very interesting information, the discussion might become more concise ad focused if the issue of potential misinterpretations is only briefly touched upon, e.g. in the limitations.

Methods/Results: It would be great to have a measure of reliability of the AQ and the IRIEC scale (e.g. Cronbach’s alpha) in either the Methods section or the Results section (when the scores in the sample are reported), as commonly done in research with questionnaire scores.

Discussion: The readability of the Discussion section would benefit from integrating information regarding specific findings and related literature, which currently seems repetitive in some cases. More specifically, paragraph 2 and 3 in the Discussion section both contain information about all findings related to the IRI. For example, the first sentence of the third paragraph (lines 473 – 474) is a mere repetition of the second sentence of the second paragraph (lines 462 – 465) in the Discussion section. The same applies to the fourth and the third paragraph regarding autistic traits. For example, the correlation between autistic traits and ZM activity, independent of emotion condition, is referred to twice, once in the third paragraph (lines 500 – 502) and the fourth paragraph (lines 520 – 521).

Small comments:

- Line 30: it is unclear what the authors refer to with “these findings”. They could use our findings” instead.

- Lines 37-40: the phrasing of this sentence suggests a causal role of mimicry in all the listed processes which seems a strong claim

- Figure 4: It is unclear why the results for the AQ and the valence ratings are visualized first (A) whereas the results for the IRIEC and the valence ratings are reported first in the Results section. Further, the figure title in the caption should also include modulatory effects of autistic traits.

7. PLOS authors have the option to publish the peer review history of their article (what does this mean?). If published, this will include your full peer review and any attached files.

Reviewer #1: No

Reviewer #2: No

---

## [Author Response · Author response to Decision Letter 1]

14 Jul 2022

We thank the reviewers for their time and insightful suggestions. During the second revision, when checking the Cronbach’s alpha values of the IRI subscores and the AQ (Reviewer 2, comment 5), we noted that the script reading the IRI did not invert the scores of nine items, resulting in low or negative Cronbach’s alpha values. This error has been corrected. Furthermore, in response to comment 1 of Reviewer 1 on the construct of emotional empathy, we decided to better couple the psychological construct and psychometric measures. We employed the construct of emotional empathy proposed by Davis (Davis, 1980, 1983) and included both the IRI empathic concern and personal distress subscores in the statistical models to evaluate the effect of emotional empathy. Therefore, the results differ from those of the previous version.

Reviewer #1: I greatly appreciate the extensive changes the authors made to the manuscript. The study design is now much clearer. However, I still have some lingering concerns regarding the revised manuscript:

Introduction

1. The key terms trait emotional empathy, spontaneous mimicry, and emotional contagion require clearer definitions and should be used consistently throughout the manuscript. First, although the authors referred to “emotional empathy”, what they measured was actually “empathic concerns” i.e., “warmth, compassion, and concern toward others” (line 161-162). This is not entirely the same as the affective/emotional empathy, which involves sharing others’ emotional states, thus emotional contagion; whilst empathic concerns require not only sharing others’ emotions, but also prosocial motivation and sympathy/compassion. Second, as a result of the first point, the hypotheses were not well grounded by the literature reviewed by the authors in Introduction, which was about emotional empathy. Third, line 44-46 “Although both spontaneous mimicry and empathy have been proposed as mechanisms facilitating emotional contagion, it has also been argued that spontaneous mimicry and emotional contagion are more likely to simply co-occur” is confusing and could also be misleading. I would suggest the authors to check the Hatfield and colleagues’ chapter for the role of emotional contagion, and also the Prochazkova & Kret (2017) study for mimicry and emotional contagion.

Hatﬁeld, E., Rapson, R. L., & Le, Y. C. L. (2011). Emotional contagion and empathy. The Social Neuroscience of Empathy, 19.

Prochazkova, E., & Kret, M. E. (2017). Connecting minds and sharing emotions through mimicry: A neurocognitive model of emotional contagion. Neuroscience & Biobehavioral Reviews, 80, 99-114.

Authors’ response: 

We have reorganized the Introduction to describe spontaneous facial mimicry, emotional contagion, emotional/affective empathy and autistic traits more thoroughly, and their relationships in the theoretical proposals and empirical data. Hatfield, and Prochazkova and Kret, are cited. We agree that some descriptions in the previous version were over-simplified. We would emphasize that the definitions and constructs of emotional empathy and emotional contagion remain very different among researchers; it is not easy to summarize these topics.

Reviewer 1 distinguished empathic concern from emotional empathy; this view is shared by some empathy researchers. For example, Zaki categorized empathic concern as a separate dimension of motivation (Zaki, 2017). However, there seems to be no widely used trait measurement that separates traits of both emotional and motivational empathy and has been validly translated into Japanese. Returning to the construct of Davis; the cited author used two kinds of reactivity when inferring the ability to feel the emotions of others, thus both the empathic concern (which measures other-oriented empathic responses) and personal distress (which measures the self-oriented empathic responses) (Davis, 1980). We thought it is best to use both the EC and PD subscales to estimate trait emotional empathy in this revision. Thus, we now refer specifically to “trait empathic concern” and “trait personal distress” in the Results.

2. The study aimed to examine the effects of a) empathic concerns and b) autistic traits on i) emotional contagion and ii) spontaneous facial mimicry. But not all the hypotheses for these relations were clearly outlined. For example, the hypothesis for the relation between autistic traits and emotional contagion is missing.

Authors’ response: 

The last part of the Introduction has been revised. Specifically, as the literature consistently showed that autistic traits do not affect emotional contagion, we now state that we did not expect the autistic traits to modulate emotional contagion under either live or video conditions.

Methods

3. Some parts of the methodology remain unclear. For example, it was mentioned in line 153-155 that “Fifteen smiling and fifteen frowning clips were used for the passive viewing component; two per condition were used for the practice ratings, and four per condition were used for the actual ratings.” Only until reading later text did I realize that the 15+15 trials described here did not include the practice trials, and that the “condition” refers to the 2 (presentation) x 2 (valence) design.

Authors’ response: 

We have revised the Methods and we have moved the “Pre-recorded videos” section to after “Paradigm and procedures”. We have also revised the “Pre-recorded videos” section to include the information that two smiling and two frowning clips were employed under the positive-video and negative-video conditions of practice passive viewing trials. We hope this improves readability.

4. Also, in line 195 “they were shown the instructions for either “Video” or “Real Person””, a previous comment on this was not addressed. The participants already knew beforehand whether the expression presented to them was pre-recorded or live. It’d be helpful if the authors can explain why it was designed like this and how this could affect the outcomes.

Authors’ response: 

As we replied to the Reviewer in the last revision, the instruction of “Video” or “Real Person” ensured that the participants knew whether an imminent stimulus was real or pre-recorded. This allowed a very clear audience effect. An effect of “the knowledge of others’ presence or being watched” has been demonstrated in previous studies using a “simulated” live interaction design (Drimalla et al., 2020; Gregory et al., 2015; Jiang et al., 2017; Koike et al., 2019; Myllyneva & Hietanen, 2015). In the absence of such instruction, participants would need to de-ambiguate or infer the live vs. pre-recorded situations by relying on subtle cues such as microexpressions or environmental inconsistencies, which do not guarantee successful detection of live performances. In the absence of instruction, it would have been difficult to validate whether participants were really prepared for live interaction, and it would have been more difficult to argue that an effect (if any) was attributable to low-level processing rather than social cognition. Particularly, when we showed prerecorded and live performance clips to naïve participants lacking such instruction during the validating rating study of our previous publication using the same paradigm, the ratings exhibited no “live effect” at all. It will be interesting to investigate the effect of low-level bottom-up processing in subsequent studies and to compare the results to those of the present study.

We have added this information to the “Paradigm and procedures” section of the Methods part (lines 324-331).

5. The rating scales for valence and arousal were shown in the figure, but were only very vaguely and confusingly described in the text that a scale of 1 to 9 was used and a question “please rate the emotion that you feel” was asked. Yet this question is rather broad and vague, and how the scale 1 to 9 should be interpreted is unclear. This also applies to line 226.

Authors’ response: 

We agree that our description of the rating procedures was incomplete. Before the practice rating trials, participants read two slides introducing the concepts of pleasure-displeasure (valence) and arousal in reference to the Russel Affect Grid (Russell et al., 1989), and were asked to rate the emotions they experienced after seeing the facial expressions using the sequence of pleasure-displeasure (valence) and arousal. The complete details are now given in the “Paradigm and procedures” section of the Methods (lines 339-353).

We have also corrected the number of practice rating trials from eight to four (one per condition).

6. Regarding line 251-253 “the mean value difference in EMG (ZM and CS) between the neutral (0-1 s after stimulus onset) and maximal phases (2.5-3.5 s after stimulus onset) of the dynamic facial expression”. This treatment remains unclear to me. It was described earlier [PREPROCESSING] that baseline-corrected values would be use, while here the difference between neutral and maximal phases was said to be used. Taken together, the authors are suggested to carefully check throughout their Methods section for a clearer presentation of their design.

Authors’ response: 

The baseline correction made during preprocessing sought to deal with low-frequency drift, so that the amplitude of EMG signal oscillation could be correctly retrieved. This does not affect the definition of the dependent variable. After preprocessing baseline correction, the trial-wise EMG data (concatenated) of the two EMG channels appeared as follows:

(Please see the attached figure in the PDF of Reply to Reviewers) 

The data were then rectified (the absolute value was calculated for each point) to reflect the amplitude of EMG oscillation. This measure is highly skewed; we thus performed a natural-log transformation. Then, for the dependent variables, we calculated the (natural log-transformed) oscillation amplitude mean differences between the neutral (0-1 s after stimulus onset) and maximal (2.5-3.5 s after stimulus onset) phases.

We have revised the “EMG data pre-processing” section to include all of the abovementioned steps to improve readability.

7. Reporting observed power is not appropriate, and also not meaningful, as it is basically a reflection of the p values. Instead, the authors are encouraged to provide an indication for how their effect sizes should be interpreted, which would be helpful for understanding the results.

Authors’ response: 

We agree with the editor and reviewers; we have removed the information.

We have added an interpretation of the partial R-squared measure: “The effect sizes for partial R-squared values in regression models are considered small at 0.01, medium at 0.09, and large at 0.25.” to the “Statistical analysis” section of the Methods (lines 442-444). The reference is (Gignac & Szodorai, 2016) and https://imaging.mrc-cbu.cam.ac.uk/statswiki/FAQ/effectSize.

Results

8. Overall, the results are not very easy to follow. Oftentimes, it was not clearly presented what were being compared, and how one condition differs from the other. Below are some examples.

9. Line 323-324 “During the live stimuli observations, a high IRIEC level was associated with a greater discrepancy in valence ratings for positive vs. negative stimuli”: Better to also describe what is the outcome in the video condition.

10. Line 336 “The AQ slope difference … was more negative”: What does “more negative” mean?

11. Line 398-399 “Although the AQ slopes for ZM reactions were not different from zero in neither live nor video conditions, the slope was significantly more positive in the video”: First, what does "more positive" mean? Second, which interaction effect was referred to?

Authors’ response: 

The comments indicate the difficulty in delineating the nature of a two-way interaction involving one continuous variable and one categorical independent variable, and a three-way interaction involving one continuous variable and two categorical independent variables. There is no definite guideline to do this. When dealing with three-way interactions, our strategy featured visualization and evaluation using separate, two-way simple slope difference testing by categorical variables (Dawson, 2014). We simply sought to reveal the nature of the interaction.

https://biologyforfun.wordpress.com/2017/10/03/three-way-interactions-in-r/

However, the coefficients themselves are “tricky” (difficult to interpret), and we agree with the editor and reviewers that more test statistics are required to render the descriptions appropriate. 

As we found no further significant three-way interaction, we needed only to clarify the two-way interactions involving one continuous and one categorical independent variable. In the revised manuscript, we visualized the interactions and reported simple slope analyses by the categorical variables (as before). In addition, we used the “pick-a-point” approach (Hayes & Matthes, 2009) and performed simple effect contrast analysis of estimated value differences between two levels of the categorical variables from the mean minus 3 SD to the mean plus 3 SD in steps of 1 SD of the continuous variable (note that the traits were already mean-centered before entry into the statistical model). This enabled us to better describe at which level of trait distribution we observe (what type of) difference in measurements of emotional contagion and facial muscular responses at two levels of the categorical variable. Indeed, this approach allowed us to correct some previous descriptions and interpretations. We hope this also improves the interpretability of the results. This information has been added to the “Statistical analysis” section of the Methods (lines 445-454).

https://stats.oarc.ucla.edu/r/seminars/interactions-r/#s4c

12. The term “positive” and “negative” were being used for three purposes in the Results section: for describing valence, for indicating the direction of a slope, and for indicating the direction of a difference. Thus, the text is rather confusing. The authors are suggested to consider using different sentence structures to avoid using the same term for different contexts (e.g., say “higher A was related to higher B” instead of “A and B were positively related”).

Authors’ response: 

We have generally followed this suggestion in the revised manuscript.

13. Sometimes both fixed and random effects in the final model were listed, while sometimes only random effects were outlined. Consider listing all fixed and random effects when reporting final models, or simply removing such sentences and referring to the tables, for clarity.

Authors’ response: 

All final models featured the same fixed effects but differed in terms of the random effects. We have listed only the complete fixed effects of the valence results. We did not mention (in the later Results sections) that other models featured the same fixed effects. We have now added this information.

Discussion

14. Discussion on the effect of empathic concerns: I am not sure I am convinced by the authors’ explanation. Results showed that participants with high EC were more sensitive to the differences between different emotional expressions (positive or negative), and a live condition enhanced this effect. This finding possibly shows that the subtle signals conveyed during live interactions and that knowing the interaction partner is real, is helpful for feeling and recognizing the emotion of the partner, especially for those with higher levels of empathic concerns, who are high in sympathy and prosocial motivation. However, note that the results about arousal rating is not entirely in line with the hypotheses (higher EC was related to lower arousal ratings when the expression was negative), and thus also requires further explanations.

Authors’ response: 

After correcting the errors in the IRI readings, we found that the live effect did not modulate the interaction between the trait and emotion conditions. In a sense, we can no longer discuss the issue. However, the interpretation of the Reviewer is attractive had the effect been observed.

The corrected results still revealed a robust two-way interaction between the IRIEC and emotion conditions in terms of arousal ratings. Indeed, a higher trait empathic concern was associated with the perception of negative expressions as less arousing. We considered that negative stimuli were inherently less emotionally contagious in terms of arousal, as revealed by the significant main effect of emotion conditions on the arousal ratings. The event-related design that in a pseudorandomized manner presented participants with positive and negative expressions during all sessions. We think that this enhanced the “contrast” in the arousal contagious ability between emotion conditions; the negative condition was more neutral (or even calming) in terms of arousal contagion, especially for individuals with high trait empathic concern. Addition of a further baseline condition (true neutral facial expressions) would aid exploration of this (non-intuitive) finding in future. We have added this information to the Discussion (line 751-760).

15. Discussion on the effect of autistic traits: I am not sure that I understand the authors’ argument that a weaker emotion contagion is relate to reduced social attention. In people with higher AQ, positive expressions were rated more positive, and negative ones rated more negative, than people with lower AQ, and this effect was even stronger in video condition than in live condition. Possibly, hypersensitivity, which is often part of the autistic traits, may play a role here, and when in live condition, they can make more "balanced" judgement. Or, it could be that the video condition, i.e., a more controlled condition, helps people with high AQ to differentiate the expressions.

16. The effect of AQ on ZM reaction: Results showed that, first, among people with lower AQ, ZM reaction in live condition was stronger than in video condition; while among people with higher AQ, no differences were between conditions in ZM reaction. Second, in video condition, higher AQ was related to stronger ZM reaction, while this association was weaker in live condition. These suggest that people with lower AQ reacted with more smiles (not necessarily mimicry as they also did so for negative expressions) when they knew the partner is real than in pre-recorded condition. But for people with higher AQ, whether in the video condition or in the live condition, they smiled to the extent similar to people with lower AQ in the live condition. In other words, participants with higher AQ did not reduce smiling in live condition, but actually increased smiling in video condition, compared to people with lower AQ. Therefore, the authors’ interpretation does not seem to hold as participants with higher AQ did not reduce their social smiling in live condition. Also, given that there is no standard for what is adequate ZM reaction, no “reduction” can be assumed.

These two papers could be helpful in interpreting empathic responses in autistic individuals:

Rieffe, C., O’Connor, R. A. G., Bülow, A., Willems. D., Hull, L., Sedgewick, F., Stockmann, L., & Blijd-Hoogewys, E. (2021). Quantity and quality of empathic responding by autistic and non-autistic adolescent girls and boys. Autism, 25(1), 199-209.

O'Connor, R. A. G., Stockmann, L., & Rieffe, C. (2019). Spontaneous helping behavior of autistic and non-autistic (pre-)adolescents: A matter of motivation? Autism Research,12, 1796-1804.

Authors’ response (comments 15 and 16): 

The issues of comments 15 and 16 are related to those of comments 8-11 of Reviewer 1, thus the interpretation of two-way interactions involving one continuous variable and one categorical variable. The previous text contained certain misinterpretations, particularly in terms of two two-way interactions involving the AQ. In the current revision, additional simple-effect contrast analyses showed that individuals with high autistic traits did not differ in terms of subjective experiential valence or ZM responses between the live and video conditions. Rather, individuals low in autistic traits exhibited lower subjective valence and weaker zygomaticus responses (irrespective of the emotion condition) under the video condition compared to the live condition. We interpreted the results based on previous findings of the audience effect (in those without ASD diagnosis) on social smiling, and used the facial feedback hypothesis to link this back to the results of subjective valence. Alternatively, it may be that individuals low in autistic traits may pay less attention to social stimuli when they know they are not dealing with a real person, but we do not state this because no previous study offered this interpretation and we thus lack the required evidence.

Other issues related to autism

17. Line 60 “Facial expressions of individuals with ASD are considered less natural”: The autistic people express emotions in their "natural" way. It is not fair to compare the naturalness of facial expressions, and to use non-autistic people as the standard.

Authors’ response: 

We did not investigate the facial expressions of individuals with ASD diagnosis. Indeed, this information is irrelevant; we have removed it.

18. Line 62 “Spontaneous facial mimicry was also found to be reduced and delayed in the ASD population”: compared to whom?

19. Line 69 “individuals with ASD showed less social smiling and mimicry of positive facial expressions”: compared to whom?

20. Line 75 “Hermans et al. showed reduced spontaneous facial mimicry of the CS response to static angry faces in females with a high autism spectrum quotient (AQ)”: compared to whom?

Authors’ response (comments 18 to 20): 

In cited studies, individuals with ASD diagnosis were compared to those without ASD diagnosis (“neurotypical populations” in the originals). The study cited in point 20 compared females of high AQ to females of low AQ. This information has been added for clarity.

Other issues related to clarity and readability

21. Line 34 “Dimberg was the first to demonstrate similar muscle activation patterns when seeing happy faces…”: Similar to what? Also this sentence is not easy to follow.

Authors’ response: 

The sentence was too long. We referred to “static grayscale pictures of happy or angry faces” in the end of that sentence. We have rephrased the sentence.

22. Line 40 “increased liking for the mimicker, and facilitated prosocial behavior”: the behavior of whom?

Authors’ response: 

We referred to the prosocial behaviors of, and social bonding between, the mimicker and mimickee. The text has been revised for clarity (lines 65-69).

23. Line 49 “…and investigated the interaction effect of trait differences and live interactions…”: It will be clearer if empathy and autistic trait are mentioned here as part of the aim. Trait differences is a very broad term.

Authors’ response: 

We have followed the suggestion (lines 108-109). Also, please note that the sentence is now followed by a subsection discussing trait emotional empathy and autistic traits.

24. Line 63 “autonomic” should better be “spontaneous”, which is what was consistently used throughout the text.

Authors’ response: 

We have followed the suggestion.

25. Line 71-72 “Recent evidence indicated that there is no difference in emotional empathy or emotion contagion between the ASD and neurotypical population”: This sentence does not connect well to the rest of the paragraph. It is in-between the paragraphs on facial mimicry. Therefore, hard to follow and grasp what is the message here. Consider moving it to the beginning of the paragraph, or to the end of the next paragraph.

Authors’ response: 

We have moved the information to the end of the paragraph, , thus after the description of the work of Press et al. (Press et al., 2010) (lines 181-184). The next paragraph discusses previous findings on the effects of the autistic traits in populations without ASD diagnosis.

26. Line 101-104: This is an important sentence about the hypotheses, but unfortunately very hard to read.

Authors’ response: 

We have divided the overtly long sentence into shorter sentences.

27. Line 165 “i.e., those with and without autism”: Better change “autism” to “a diagnosis of autism” for clarity.

Authors’ response: 

We have adopted this suggestion (“individual/population with/without a diagnosis of ASD”) throughout the revised manuscript.

28. Line 247 “Statistical analyses of live interaction data”: why only live data is mentioned here, while both presentation modes are actually in the models?

Authors’ response: 

We have changed the section title to “Statistical analyses” to avoid confusion.

29. The authors are suggest to ensure consistent use of terms throughout the manuscript. For example, currently “valence”, “emotion”, and “emotion categories” were used interchangeably.

Authors’ response: 

We have thoroughly examined this issue and made adjustments whenever appropriate. For example, we now use “emotion conditions” to describe the categorical variable of the study design.

30. The manuscript would benefit from English language editing.

Authors’ response: 

Both the previous and the current revised texts and replies to reviewers were sent for English proofreading. However, English-language editors vary in terms of style. We have mentioned the reviewer’s concern to our proofreading service.

The English in the current revised manuscript has been checked by at least two professional editors, both native speakers of English. For a certificate, please see:

http://www.textcheck.com/certificate/AkoSUv

Reviewer #2: Overall, it is evident that the authors acknowledged and implemented the feedback. There are, however, still some points that require further improvement.

Major concern:

1. Results: It is appreciated that the authors implemented the suggested comparison between presentation conditions (e.g. positive-live vs. positive-video) to their Results section. However, I was puzzled by the interpretations that were drawn from the examination of slope differences. First, even though some slope differences did not reach significance (e.g. IRIEC slope difference in live vs. video for both arousal ratings of positive and negative stimuli, lines 357 – 361), the authors still interpreted this as e.g. more/less arousing perception of the stimuli in the live condition.

Secondly, they described slope differences to be greater in one condition vs. the other (e.g. IRIEC slope difference in positive vs. neutral in the live vs video condition, lines 353-357) even though no statistical test supporting this claim is reported, i.e. they were most probably only referring to numerical values or graphs.

Thirdly, while slope differences might be informative to compare between presentation conditions, they do not yield any information on whether there is indeed a significant association between e.g. the IRIEC and arousal ratings in a specific condition. As the coding of the predictors is not specified in the model description (e.g. whether they are sum coded or which level acts as a reference level), it is impossible to retrieve this information from the model tables. As simplest solution, I would suggest to report whether the individual slopes for each combination of the factor levels, e.g. positive-live, are significantly different from 0 in the three-way interaction with each of the trait variables and, as a next step, to look at slope differences. Only the significant findings should then be further interpreted which should, if necessary, lead to adjustments in the Abstract, the Result section and the Discussion section. The outlined concerns also apply to the reporting of the results in the Supporting Information. 

Authors’ response: 

This comment again reflects the difficulty of interpretating interactions involving both continuous and categorical variables (see comments 8–11 of Reviewer 1). Please see our reply above. We agree that more statistical testing was required to properly describe the nature of the interactions. We now include visualizations, simple slope analyses per categorical level (Dawson, 2014), and the “pick-a-point” approach (Hayes & Matthes, 2009) to perform simple-effect contrast tests at mean-centered continuous variable levels from –3 SD to 3 SD in steps of 1 SD while comparing two levels of the categorical variables; we use the Šidák method to correct for multiple comparisons.

https://stats.oarc.ucla.edu/r/seminars/interactions-r/#s4c

We thank Reviewer 2 for pointing out that the reference levels of the categorical variables were not given in the previous manuscript. We have added this information to the “Statistical analysis” section of the Methods. The reference level for the emotion condition is “Negative” and that for the presentation condition is “Video” (lines 416-418).

2. Lastly, the addition of a post-hoc power analysis is highly criticized and I would recommend to exclude it (see: https://daniellakens.blogspot.com/2014/12/observed-power-and-what-to-do-if-your.html;
https://data.library.virginia.edu/post-hoc-power-calculations-are-not-useful/;
https://link.springer.com/article/10.1007/s12144-018-0018-1).

Authors’ response: 

We agree with the editor and Reviewers; we have removed this information.

Minor concerns:

3. Introduction: The introduction benefits from the clarification of concepts and addition of research on autistic traits. The authors might reconsider starting the first paragraph in the introduction with the same words as the abstract.

Authors’ response: 

We have restructured and expanded the Introduction to overview the objective of the present study first, and then more thoroughly summarize the constructs that we cover and the relationships among them. The Abstract has also been rewritten to reflect the corrected results (Reviewer 2 comment 5).

4. Methods (Stimuli validation) + Discussion: It was great to see that the authors put a lot of effort in examining whether the negative/angry expressions could potentially be confused with sad expressions. As this was not one of the main goals of the study, I was surprised to find a quite long paragraph on it in the Discussion section. Despite the fact that this paragraph contains very interesting information, the discussion might become more concise ad focused if the issue of potential misinterpretations is only briefly touched upon, e.g. in the limitations.

Authors’ response: 

We have shortened the relevant paragraph in the Discussion and moved it to before the limitations. The paragraph explains that the issue of ambiguity in negative expressions is inherent in the culture wherein we conduct our work. It is important for readers to bear this in mind when interpreting our results.

5. Methods/Results: It would be great to have a measure of reliability of the AQ and the IRIEC scale (e.g. Cronbach’s alpha) in either the Methods section or the Results section (when the scores in the sample are reported), as commonly done in research with questionnaire scores.

Authors’ response: 

We thank the reviewers for pointing out the issue with Cronbach α. We checked the values for the IRI and subscores, and found unusually low or negative values, which prompted us to recheck the questionnaire data pipeline. We discovered that the script reading the IRI and the subscore data did not invert nine of the IRI questions. This issue has been corrected; the results now differ.

After correction of the erroneous IRI inversions, the standardized Cronbach α values were 0.764 for the IRIEC, 0.833 for the IRIPD, and 0.747 for the AQ. This information has been added to the “Questionnaire scores” section of the Results (line 457-460).

6. Discussion: The readability of the Discussion section would benefit from integrating information regarding specific findings and related literature, which currently seems repetitive in some cases. More specifically, paragraph 2 and 3 in the Discussion section both contain information about all findings related to the IRI. For example, the first sentence of the third paragraph (lines 473 – 474) is a mere repetition of the second sentence of the second paragraph (lines 462 – 465) in the Discussion section. The same applies to the fourth and the third paragraph regarding autistic traits. For example, the correlation between autistic traits and ZM activity, independent of emotion condition, is referred to twice, once in the third paragraph (lines 500 – 502) and the fourth paragraph (lines 520 – 521).

Authors’ response: 

The discussion has been extensively re-organized and revised to accommodate the corrected results (Reviewer 2, comment 5). Care has been taken to avoid repetition.

Small comments:

7. - Line 30: it is unclear what the authors refer to with “these findings”. They could use our findings” instead.

Authors’ response: 

We have followed the suggestion.

8.- Lines 37-40: the phrasing of this sentence suggests a causal role of mimicry in all the listed processes which seems a strong claim

Authors’ response: 

Also in response to comment 1 of Reviewer 1, we have revised and expanded the summaries of the effects of spontaneous mimicry and also present alternative theoretical proposals on how other social modulatory factors may affect spontaneous mimicry (line 50-60).

9. - Figure 4: It is unclear why the results for the AQ and the valence ratings are visualized first (A) whereas the results for the IRIEC and the valence ratings are reported first in the Results section. Further, the figure title in the caption should also include modulatory effects of autistic traits.

Authors’ response: 

The former Fig. 4 would be more attractive if, for both rows, the red-blue legend of the emotion conditions appeared on the right side. However, as the three-way interaction disappeared after correcting the IRI errors, the issue no longer exists.

Bibliography

Davis, M. H. (1980). A Multidimensional Approach to Individual Differences in Empathy. JSAS Catalog of Selected Documents in Psychology, 10, 85.

Davis, M. H. (1983). Measuring Individual-Differences in Empathy—Evidence for a Multidimensional Approach. Journal of Personality and Social Psychology, 44, 113–126. https://doi.org/10.1037/0022-3514.44.1.113

Dawson, J. F. (2014). Moderation in Management Research: What, Why, When, and How. Journal of Business and Psychology, 29(1), 1–19. https://doi.org/10.1007/s10869-013-9308-7

Drimalla, H., Scheffer, T., Landwehr, N., Baskow, I., Roepke, S., Behnia, B., & Dziobek, I. (2020). Towards the automatic detection of social biomarkers in autism spectrum disorder: Introducing the simulated interaction task (SIT). Npj Digital Medicine, 3(1), 25. https://doi.org/10.1038/s41746-020-0227-5

Gignac, G. E., & Szodorai, E. T. (2016). Effect size guidelines for individual differences researchers. Personality and Individual Differences, 102, 74–78. https://doi.org/10.1016/j.paid.2016.06.069

Gregory, N. J., Lόpez, B., Graham, G., Marshman, P., Bate, S., & Kargas, N. (2015). Reduced Gaze Following and Attention to Heads when Viewing a “Live” Social Scene. PLOS ONE, 10(4), e0121792. https://doi.org/10.1371/journal.pone.0121792

Hayes, A. F., & Matthes, J. (2009). Computational procedures for probing interactions in OLS and logistic regression: SPSS and SAS implementations. Behavior Research Methods, 41(3), 924–936. https://doi.org/10.3758/BRM.41.3.924

Jiang, J., Borowiak, K., Tudge, L., Otto, C., & von Kriegstein, K. (2017). Neural mechanisms of eye contact when listening to another person talking. Social Cognitive and Affective Neuroscience, 12(2), 319–328. https://doi.org/10.1093/scan/nsw127

Koike, T., Sumiya, M., Nakagawa, E., Okazaki, S., & Sadato, N. (2019). What Makes Eye Contact Special? Neural Substrates of On-Line Mutual Eye-Gaze: A Hyperscanning fMRI Study. Eneuro, 6(1), ENEURO.0284-18.2019. https://doi.org/10.1523/ENEURO.0284-18.2019

Myllyneva, A., & Hietanen, J. K. (2015). There is more to eye contact than meets the eye. Cognition, 134, 100–109. https://doi.org/10.1016/j.cognition.2014.09.011

Press, C., Richardson, D., & Bird, G. (2010). Intact imitation of emotional facial actions in autism spectrum conditions. Neuropsychologia, 48(11), 3291–3297. https://doi.org/10.1016/j.neuropsychologia.2010.07.012

Russell, J. A., Weiss, A., & Mendelsohn, G. A. (1989). Affect Grid: A Single-Item Scale of Pleasure and Arousal. Journal of Personality and Social Psychology, 57(3), 493–502. https://doi.org/10.1037/0022-3514.57.3.493

Zaki, J. (2017). Moving beyond Stereotypes of Empathy. Trends in Cognitive Sciences, 21(2), 59–60. https://doi.org/10.1016/j.tics.2016.12.004

---

## [Decision Letter · Decision Letter 2]

18 Oct 2022

PONE-D-21-34896R2An Investigation of the Modulatory Effects of Empathic and Autistic Traits on Emotional and Facial Motor Responses during Live Social InteractionsPLOS ONE

Dear Dr. Hsu,

Thank you for submitting your manuscript to PLOS ONE. I have received comments from the previous two reviewers. All of us appreciate the extensive edits you have made and recognize the importance of this work. You will also see that both reviewers, along with myself continue to have several concerns. 

As before, I invite you to carefully consider the comments from both reviewers. Additionally, I would add that I struggled with several aspects of your manuscript. First, the introduction covered several theoretical debates about separate, but related topics (e.g., facial mimicry, emotional contagion, facial feedback hypothesis, etc). However, it was difficult to follow the core argument being made, and the text does not clearly lead to the hypotheses of this manuscript. This section could be considerably condensed for space, readability and cohesion of the argument. Clarity in this argument could help make stronger hypotheses. For example, this work appears to be an extension of previous findings, however, there is little discussion about how this study relates to those findings. Indeed, the hypothesis that live-action stimuli will increase the enhancement of mimicry and emotional contagion by high trait empathy, are not well developed by the introduction. Also, the exploratory hypothesis regarding the role of autism traits leads to a null prediction. Given this, I wonder if it is worth including this exploratory aspect of the paper. This topic might be better served as part of a separate manuscript.

In addition to the reviewers' concerns regarding the results, I also have concerns about the simple effects contrasts using -3 to 3 SD as stratification levels. Given the overall sample size of this study, what are the cell sizes for such an analysis? If there is only one or two participants in some of the extreme cells, how can the estimates be considered stable? I also worry about the presentation of main effects before interactions, as noted by the reviewers.

Similarly, both reviewers, as well as myself, were perplexed by the reference to Audience Effects in the discussion. This was not covered in the introduction at all. If this is a key feature of the study, it need to be raised in the introduction. However, I again caution of the manuscript becoming too complex, or becoming too speculative. This is especially important given that the authors' primary hypotheses were not supported by the data. I know it is difficult to discuss null findings, and I do not subscribe to the belief null findings are unpublishable. However, I do believe caution must be used when interpreting the results, and authors should be very careful not to speculate too heavily.

Finally, I noticed several typos throughout the manuscript.

Given this, we invite you to submit a revised version of the manuscript that addresses the points raised during the review process. Please submit your revised manuscript by Dec 02 2022 11:59PM. If you will need more time than this to complete your revisions, please reply to this message or contact the journal office at plosone@plos.org. Please include the following items when submitting your revised manuscript:A rebuttal letter that responds to each point raised by the academic editor and reviewer(s). You should upload this letter as a separate file labeled 'Response to Reviewers'.A marked-up copy of your manuscript that highlights changes made to the original version. You should upload this as a separate file labeled 'Revised Manuscript with Track Changes'.An unmarked version of your revised paper without tracked changes. You should upload this as a separate file labeled 'Manuscript'.

We look forward to receiving your revised manuscript.

Kind regards,

Eric J. Moody, Ph.D.

Academic Editor

PLOS ONE

Reviewers' comments:

Reviewer's Responses to Questions

**Comments to the Author**

1. If the authors have adequately addressed your comments raised in a previous round of review and you feel that this manuscript is now acceptable for publication, you may indicate that here to bypass the “Comments to the Author” section, enter your conflict of interest statement in the “Confidential to Editor” section, and submit your "Accept" recommendation.

Reviewer #1: (No Response)

Reviewer #2: (No Response)

2. Is the manuscript technically sound, and do the data support the conclusions?

Reviewer #1: Yes

Reviewer #2: Partly

3. Has the statistical analysis been performed appropriately and rigorously? 

Reviewer #1: Yes

Reviewer #2: Yes

4. Have the authors made all data underlying the findings in their manuscript fully available?

Reviewer #1: Yes

Reviewer #2: Yes

5. Is the manuscript presented in an intelligible fashion and written in standard English?

Reviewer #1: (No Response)

Reviewer #2: Yes

6. Review Comments to the Author

Reviewer #1: I appreciate the authors' effort in revising the manuscript. In this revised version, the analyses and the results were presented with clarity, and I believe the results are interesting to the readers of PLOS ONE. Yet, given that a large proportion of the text has been rewritten, I do still have some concerns regarding the readability of this manuscript. I will give some examples below.

1. Many theories/studies were listed to describe the literature or interpret the results, yet oftentimes only until the last sentence did the authors mention which is the one they follow, yet still without explicitly explained reasons for the choice. Thus, the text became less concise and with contradictory views that are difficult to follow. There are indeed different views on how empathy and emotional contagion can be defined in the literature; it is thus of importance that the authors are clear about which view they take and the reason for their choice. The authors are suggested to focus on the theoretical framework of their study and be clear about how it supports the hypotheses and study design.

For example, in the Intro, only until the last sentence under "Spontaneous facial mimicry and emotional contagion" did the authors mention that they considered facial mimicry and emotion contagion as parallel processes (despite their co-occurrence often observed), without a clear reason for why this was adopted in this study. The authors may already mention the importance of the social context earlier in the text, as this appears to be the support for why the authors chose to see them as two processes rather than one. Similarly in Discussion, these two processes were often discussed in a mixed manner. For example, the "general propensity to emotional contagion" was used to explain the effect of trait personal distress on ZM mimicry, without clear reasoning.

Also, under "Trait emotional empathy and autistic traits" (Intro), task design was mentioned to play a role in the effect of autistic trait. This is an important point, yet unfortunately came too late and not explained clearly. The authors did not mention anything about the task design in the previous description of these studies, so it is unclear how the Press et al study is different from the McIntosh and Drimalla studies, in terms of the attentional requirement in their tasks. Instead, many other features about those studies were described, yet their relevance to the current studied topic remained unexplained.

Moreover, whether the task could induce a clear "audience effect" was a focus in the task design of this study, yet this was not mentioned as part of the argument in the Intro when the hypotheses were defined.

2. Sometimes the text deviates from the main topic, or the topic switched. For example, in the Intro, the authors used one paragraph to describe the IRI, which should better be described in Methods to avoid repetition. In the Intro, the authors are suggested to focus on explaining the constructs. In the last paragraph under "Trait emotional empathy and autistic traits", the focus is about autistic traits and facial mimicry, yet in the concluding sentence the topic switched to trait emotional empathy and emotional contagion and mimicry. Also for example in the Discussion, in a paragraph that is about autistic people or people with high AQ, sometimes the topic switched to non-autistic people in-between the text. The authors may e.g., discuss the patterns of autistic vs. non-autistic (or high vs low AQ) in separate paragraphs to help readers follow the arguments.

Adding sub-headings may help readers follow the different topics discussed in the Discussion.

3. There are several repetitions throughout the manuscript. For example, in Intro under "Trait emotional empathy and autistic traits", the distinction between facial mimicry and emotional contagion and the role of the social context repeated what has been mentioned in the previous section. Also, the summary of findings in the 1st paragraph of the Discussion is rather vague. It is suggested to directly tell what the effects are (rather than only saying that there're modularly effects towards different conditions) to avoid repetitions later on and thus be more concise. Another example is in the paragraph in Discussion starting with "We found no reduction in facial mimicry...", which repeated the previous paragraph. It may also be an idea to focus the discussion on the findings that did not meet the expectations, as the ones that met the expectations and were in line with previous studies were already explained in the Intro.

4. The bridge between sentences was sometimes unclear, or two sentences were bridged in a not very logical way. For example, in the 1st paragraph of the Intro "Trait emotional empathy enhances.... However....", the "However" here should better be changed to "Moreover". In the 1st paragraph under Spontaneous facial mimicry and emotional contagion, it was mentioned that "as most previous studies did not provoke antagonistic responses," which contradicts a previous sentence about CS responses towards angry faces. In Methods, it was mentioned that "because some people might consider that the models' negative expressions...", yet in the text above, it was actually unclear that the models were instructed to express anger, and thus this sentence came as a surprise.

In the 1st paragraph of the Discussion, it was mentioned that "we did not find any evidence... that a live interaction design affected the modularly effects...", which immediately contradicts the next sentence where the effects on ZM responses and subjective valence were described. Also in the Discussion, although one important result was that the participants with high and low AQ did not differ in ZM/CS responses, this finding was not mentioned anywhere, leading to difficulty in understanding how the present study contradicted with the Drimalla study.

These sentences should be reformulated for clarity and a better flow.

Other issues:

- Throughout the manuscript, the expression "individuals with ASD diagnosis compared to 'others'" was often used, which should better be revised. The authors are suggested to specify the control group used by the study cited (e.g., changed to "compared to those without an ASD diagnosis" if this was the case for the study cited) for better clarity and for a more inclusive language use.

- From Methods and Results, it appears that participant gave their subjective ratings after all the passive-viewing trials, and that passive-viewing and subjective ratings had different numbers of trials (60 vs 16). What's less clear to me is whether this means that the participants had to rate an expression that they had already seen for a second time. The effect of such a design should be discussed.

- A brief discussion on the use of model diagnostics and the exclusion of extreme data would further help readers understand and interpret the results.

Reviewer #2: It is obvious that the authors put a lot of effort in addressing the reviewers’ comments and the manuscript clearly improved. Nevertheless, I still have some concerns which should be addressed before publication:

General concerns:

1. I appreciate that the authors aimed to clarify the relationship between the diagnosis ASD and autistic trait levels. However, the phrasing “the autism spectrum is a continuum that covers the entire population” (line 185 and line 293) seems to suggest that “spectrum” would reflect a continuum whereas it actually reflects the variability (multiple dimensions) within the ASD diagnosis. In these lines, the usage of “spectrum” when referring to variations in IRIEC traits (line 747, “along the spectrum of the IRIEC”) seems rather inappropriate.

2. The authors already become more consistent in using specific terms for specific concepts. This is, however, not always the case for “emotion contagion” (e.g. line 94, line 107, line 747) and “emotional contagion” (e.g. line 25, line 71, line 771).

3. In multiple instances, the authors claim that there was no effect of autistic traits on emotional contagion or spontaneous mimicry (abstract: lines 31-32, discussion: line 738). I assume that they refer to the lack of a significant interaction between emotion condition and autistic traits (and not the lack of a main effect of autistic traits) here, but this should be clarified to the reader.

Abstract:

4. As valence itself is a neutral word, “higher subjective valence” (line 26) should be defined in its directionality, i.e. more positive.

5. I would recommend the authors to already clarify in the abstract that emotional contagion is measured via subjective experiential ratings, i.e. by mentioning it in parentheses in line 21.

6. Since especially the contrast in the presentation effect between e.g. individuals low in autistic traits vs. high in autistic traits is interesting, I personally would not say that live interaction designs are especially valuable for individuals low in autistic traits (line 34, similar trait empathy).

Introduction:

7. It is unclear why findings on modulated facial mimicry by autistic trait levels which are reported in the respective paragraph would “imply a positive correlation between trait emotional empathy and both emotional contagion and spontaneous facial mimicry” (lines 197-198). Further, in the next sentence, it is not very clear why and how the authors relate findings with regard to emotional contagion and facial mimicry to each other, given that they claim to look at them as separate constructs. This might, however, only be due to the way in which it is phrased.

Results:

8. I would recommend the authors to be careful when reporting significant main effects in the context of a significant interaction in which the main effect is involved (e.g. lines 614-615) since they might not be apparent in the absence of the interaction.

9. In line with Reviewer 1’s comment on the previous revision, the authors could be more precise in their phrasing. For example, they refer to “significantly more ZM responses” in the description of their results while not using count data but continuous variables (strength of activation) in their analyses.

Discussion:

10. As the results greatly differ between the model fits including all data vs. remaining data after model diagnostics, the robustness of effects should be discussed.

- Given the introduction and hypotheses, I would further appreciate if the following points would be touched upon in the introduction:

o Why were no modulatory effects of any trait dimension found in the model on the corrugator data? It is only very briefly mentioned that the corrugator is generally less mimicked.

o Even though it is clearly mentioned in the abstract, there is no potential explanation on why the presentation condition had no impact on the modulatory effect of empathy on contagion and facial mimicry. As this was one of the primary hypotheses, I would have appreciated a short speculation.

o While arguing that emotional contagion and facial mimicry should be regarded as separate constructs in the introduction, the authors relate them to each other again in the discussion of the findings on modulatory effects of autistic traits (lines 728-737). Given that the relationship between emotional contagion and mimicry was quite extensively reviewed in the introduction and their separation strongly favored, the authors might consider referring back to this discussion, given their new findings and interpretation.

11. While the “audience effect” (line 703 and following) is surely relevant in research on social interactions, the presence of others (third person perspective) is qualitatively different from interacting with another person (second person perspective) as it occurred in this study. Therefore, I would be careful to claim that stronger smiles are displayed in the live condition due to an “audience effect”.

12. In some parts, the authors might consider being more concise, i.e.

- “smiled equally” (line 726): as ZM activity is very close to 0, it could as well be said that they did not smile in both conditions. I would therefore try to phrase it more neutral.

- “rather calming”(line 756): were the ratings of the negative stimuli generally indeed <5 (on the calming side of the affect grid) or were they simply less arousing than the positive stimuli?

7. PLOS authors have the option to publish the peer review history of their article (what does this mean?). If published, this will include your full peer review and any attached files.

Reviewer #1: No

Reviewer #2: No

---

## [Author Response · Author response to Decision Letter 2]

15 Nov 2022

The Editor: I also have concerns about the simple effects contrasts using -3 to 3 SD as stratification levels. Given the overall sample size of this study, what are the cell sizes for such an analysis? If there is only one or two participants in some of the extreme cells, how can the estimates be considered stable? 

Author's response:

The simple effect contrasts were estimated using the emmeans package. The emmeans package uses a reference grid approach to estimate/predict marginal means based on the fitted model [1], and not directly on the data. Therefore, the actual cell size should not be a concern (https://cran.r-project.org/web/packages/emmeans/vignettes/basics.html). We have added this information to the Statistical analysis section (L414-417).

Reviewer #1: I appreciate the authors' effort in revising the manuscript. In this revised version, the analyses and the results were presented with clarity, and I believe the results are interesting to the readers of PLOS ONE. Yet, given that a large proportion of the text has been rewritten, I do still have some concerns regarding the readability of this manuscript. I will give some examples below.

1. Many theories/studies were listed to describe the literature or interpret the results, yet oftentimes only until the last sentence did the authors mention which is the one they follow, yet still without explicitly explained reasons for the choice. Thus, the text became less concise and with contradictory views that are difficult to follow. There are indeed different views on how empathy and emotional contagion can be defined in the literature; it is thus of importance that the authors are clear about which view they take and the reason for their choice. The authors are suggested to focus on the theoretical framework of their study and be clear about how it supports the hypotheses and study design.

For example, in the Intro, only until the last sentence under "Spontaneous facial mimicry and emotional contagion" did the authors mention that they considered facial mimicry and emotion contagion as parallel processes (despite their co-occurrence often observed), without a clear reason for why this was adopted in this study. The authors may already mention the importance of the social context earlier in the text, as this appears to be the support for why the authors chose to see them as two processes rather than one. Similarly in Discussion, these two processes were often discussed in a mixed manner. For example, the "general propensity to emotional contagion" was used to explain the effect of trait personal distress on ZM mimicry, without clear reasoning.

Also, under "Trait emotional empathy and autistic traits" (Intro), task design was mentioned to play a role in the effect of autistic trait. This is an important point, yet unfortunately came too late and not explained clearly. The authors did not mention anything about the task design in the previous description of these studies, so it is unclear how the Press et al study is different from the McIntosh and Drimalla studies, in terms of the attentional requirement in their tasks. Instead, many other features about those studies were described, yet their relevance to the current studied topic remained unexplained.

Moreover, whether the task could induce a clear "audience effect" was a focus in the task design of this study, yet this was not mentioned as part of the argument in the Intro when the hypotheses were defined.

Author's response:

Based on the requests of the editor and reviewers, we have shortened the Introduction to include only information directly relevant to the present study and reviewers’ concerns. Conflicting theoretical proposals regarding the mechanisms of emotional mimicry have been presented in a series of recent articles [2–5], and we cite and refer interested readers to these articles for further details. We have stated that spontaneous mimicry involves multilevel appraisal and response generation mechanisms that cannot be entirely accounted for by primitive emotional contagion, as noted by Scherer [5]. Therefore, we consider spontaneous facial mimicry and emotional contagion as separate processes that often co-occur. Despite considering the Social Regulator or Communication Accommodation perspectives, we did not negate the primitive emotional contagion account, according to which spontaneous mimicry is involved in emotional contagion. We have noted that this provides an indirect explanation, where no previous report has indicated that IRIPD modulates facial mimicry (L734-735, 744-745).

In the Introduction, we have revised and shortened the paragraph that reviews studies investigating ASD and spontaneous mimicry, and moved Press et al.’s explanation of inconsistent findings to an earlier part of the paragraph (L108-111)). The main intention was to show that a clear relationship between ASD diagnosis and levels of spontaneous mimicry has not been demonstrated empirically. Moreover, the task design has been proposed as an influential factor. The present study also involved passive observation of facial stimuli, which should have led to reduced facial mimicry in high-AQ participants, according to Press et al.’s proposal, yet no such effect was found in our results. Therefore, the proposed task effect may not be relevant to the present study, although it remains as a possible explanation for the inconsistency in the previous findings. We have further discussed the absence of reduction in facial mimicry in participants with high AQ scores in the Discussion (L703-713).

We have added information about the audience effect to the Introduction (L144-150).

2. Sometimes the text deviates from the main topic, or the topic switched. For example, in the Intro, the authors used one paragraph to describe the IRI, which should better be described in Methods to avoid repetition. In the Intro, the authors are suggested to focus on explaining the constructs. In the last paragraph under "Trait emotional empathy and autistic traits", the focus is about autistic traits and facial mimicry, yet in the concluding sentence the topic switched to trait emotional empathy and emotional contagion and mimicry. Also for example in the Discussion, in a paragraph that is about autistic people or people with high AQ, sometimes the topic switched to non-autistic people in-between the text. The authors may e.g., discuss the patterns of autistic vs. non-autistic (or high vs low AQ) in separate paragraphs to help readers follow the arguments.

Adding sub-headings may help readers follow the different topics discussed in the Discussion.

Author’s response:

As suggested, we have moved the description of IRI from the Introduction to the Methods section (L232-241).

We have removed the part of the concluding sentence in the Introduction about autistic traits and facial mimicry, which referred to emotional empathy.

We have moved the part discussing “the consistent pattern of AQ × presentation condition interaction in terms of subjective valence and the extent of smiling” to the first paragraph of that sub-section in the discussion (L659-671).

We have added sub-headings to the Discussion.

3. There are several repetitions throughout the manuscript. For example, in Intro under "Trait emotional empathy and autistic traits", the distinction between facial mimicry and emotional contagion and the role of the social context repeated what has been mentioned in the previous section. Also, the summary of findings in the 1st paragraph of the Discussion is rather vague. It is suggested to directly tell what the effects are (rather than only saying that there're modularly effects towards different conditions) to avoid repetitions later on and thus be more concise. Another example is in the paragraph in Discussion starting with "We found no reduction in facial mimicry...", which repeated the previous paragraph. It may also be an idea to focus the discussion on the findings that did not meet the expectations, as the ones that met the expectations and were in line with previous studies were already explained in the Intro.

Author's response:

As suggested, we have removed repetitious text and shortened the Introduction. specifically, we merged the “Spontaneous facial mimicry and emotional contagion” and “Trait emotional empathy and autistic traits” sub-sections. The merged sub-section started by introducing Davis’ construct of empathy, spontaneous facial mimicry, and how they are associated with emotional contagion.

We have revised the sentence in the first paragraph of the Discussion regarding the presentation condition × autistic trait interaction accordingly (L647-653).

We modified the Discussion to clarify the information regarding “participants with high AQ showed the same social smile in both presentation conditions” and “no reduction in facial mimicry among participants with higher AQ scores”. We specified that the social smile occurs regardless of emotional condition, but facial mimicry refers to consistent muscular (ZM/CS) responses to the models’ facial expressions according to the emotional condition. Therefore, we have discussed these two issues in two consecutive but separate sub-sections, and added sub-headings to avoid confusion. (L692-701, 703-713). 

4. The bridge between sentences was sometimes unclear, or two sentences were bridged in a not very logical way. For example, in the 1st paragraph of the Intro "Trait emotional empathy enhances.... However....", the "However" here should better be changed to "Moreover". In the 1st paragraph under Spontaneous facial mimicry and emotional contagion, it was mentioned that "as most previous studies did not provoke antagonistic responses," which contradicts a previous sentence about CS responses towards angry faces. In Methods, it was mentioned that "because some people might consider that the models' negative expressions...", yet in the text above, it was actually unclear that the models were instructed to express anger, and thus this sentence came as a surprise.

In the 1st paragraph of the Discussion, it was mentioned that "we did not find any evidence... that a live interaction design affected the modularly effects...", which immediately contradicts the next sentence where the effects on ZM responses and subjective valence were described. Also in the Discussion, although one important result was that the participants with high and low AQ did not differ in ZM/CS responses, this finding was not mentioned anywhere, leading to difficulty in understanding how the present study contradicted with the Drimalla study.

These sentences should be reformulated for clarity and a better flow.

Author's response:

We have changed the “However” to “Moreover” in the first paragraph according to the reviewer’s suggestion (L42).

The sentence “as most previous studies did not provoke antagonistic responses…” has been deleted to shorten the Introduction (see response point 1 to Reviewer 1’s point 3). However, participants had a (simulated) antagonistic conflict with the person in the video whom they later observed in Mauersberger et al.’s paradigm [6]. The referred antagonistic context did not involve only the presentation of a frowning picture of an irrelevant person to the participant.

We have clarified that the models were instructed to make happy and angry faces in the “Pre-recorded videos” section and referred the readers to Fig 2 (L320-322).

We have clarified our interpretation of the findings in the Discussion. We did not detect a three-way interaction, but identified a two-way AQ×presentation conditions interaction effect on the ZM and valence ratings. The two-way interaction was not considered indicative of modulatory effects of the presentation conditions on facial mimicry or emotional contagion, because this would require the modulation to be consistent with the emotional condition, which would only be the case in a three-way interaction involving the emotional condition (L649-653). We have discussed social smiles (for the two-way interactions, L659-701), and the lack of a modulatory effect of autistic traits on facial mimicry (L703-713).

We added in the sub-section “Social smile, reputation management, and autistic traits” that “in the simple slope analysis of the AQ × presentation condition interaction, the AQ slopes of the ZM reactions were no different from zero under either live or video conditions, suggesting that participants with high and low AQ did not differ in ZM responses,” which contrast with Drimalla et al. who reported fewer social smiles in participants with an ASD diagnosis (L672-677).

Other issues:

- Throughout the manuscript, the expression "individuals with ASD diagnosis compared to 'others'" was often used, which should better be revised. The authors are suggested to specify the control group used by the study cited (e.g., changed to "compared to those without an ASD diagnosis" if this was the case for the study cited) for better clarity and for a more inclusive language use.

Author’s response:

We followed this suggestion on the issue of inclusive language. 

- From Methods and Results, it appears that participant gave their subjective ratings after all the passive-viewing trials, and that passive-viewing and subjective ratings had different numbers of trials (60 vs 16). What's less clear to me is whether this means that the participants had to rate an expression that they had already seen for a second time. The effect of such a design should be discussed.

Author's response:

We added to the “Paradigm and procedures” subsection that the relative proportions of passive viewing (EMG) trials and rating trials align with our prior assumption about the effect size and signal-to-noise ratio of spontaneous facial mimicry and ratings. The EMG would have a smaller effect size and lower signal-to-noise ratio than ratings, thus requiring significantly more repetitions per participant. However, participants might experience a loss of attention or interest if the passive viewing period persists for too long. With 60 EMG and 16 rating trials, the total testing time per session was less than 1 hour (L270-276).

We also added to the “Paradigm and procedures” subsection the information that no rating was requested in the passive viewing trials to detect spontaneous, automatic facial mimicry without top-down emotional processing, as in previous studies [7,8], and to prevent motion-related EMG artifacts associated with head and hand movements (L290-292).

For the video conditions, the clips (more than 20 per emotional condition) used in the passive viewing trials (15 per emotional condition) and rating trials (4 per emotional condition) were not repeated. We added this information to the “Paradigm and procedures” subsection. The live conditions were always newly performed by the models (L329-331).

Because the models consistently performed the smiling and frowning, participants might have experienced that they viewed passively the model smiling or frowning 60 times and then asked for ratings for another 16 trials. The sequential effect of “passive viewing � ratings” might have caused participants to be more adapted and desensitized to the stimuli during rating trials. This might have increased the hurdle of detecting modulatory effects relevant to emotional contagion. We discussed this issue in the subsection “The sequential effect of procedures” as a limitation (L817-824).

- A brief discussion on the use of model diagnostics and the exclusion of extreme data would further help readers understand and interpret the results.

Author's response:

We have added a paragraph to the Limitation sub-section to discuss how decisions regarding model diagnostics affected the results (L826-839). As new methods for linear mixed model diagnostics are currently being developed, and concrete guidelines have not been formulated, we encourage readers to consider the reported results both with and without model diagnostics under this context, and to reanalyze the deposited data as new model diagnostic algorithms and model treatment methods emerge.

Reviewer #2: It is obvious that the authors put a lot of effort in addressing the reviewers’ comments and the manuscript clearly improved. Nevertheless, I still have some concerns which should be addressed before publication:

General concerns:

1. I appreciate that the authors aimed to clarify the relationship between the diagnosis ASD and autistic trait levels. However, the phrasing “the autism spectrum is a continuum that covers the entire population” (line 185 and line 293) seems to suggest that “spectrum” would reflect a continuum whereas it actually reflects the variability (multiple dimensions) within the ASD diagnosis. In these lines, the usage of “spectrum” when referring to variations in IRIEC traits (line 747, “along the spectrum of the IRIEC”) seems rather inappropriate.

Author's response:

As suggested, we have changed the word “spectrum” to “along the range of the IRIEC” (L717).

2. The authors already become more consistent in using specific terms for specific concepts. This is, however, not always the case for “emotion contagion” (e.g. line 94, line 107, line 747) and “emotional contagion” (e.g. line 25, line 71, line 771).

Author's response:

As suggested, we have revised the manuscript thoroughly to ensure that “emotional contagion” is used throughout.

3. In multiple instances, the authors claim that there was no effect of autistic traits on emotional contagion or spontaneous mimicry (abstract: lines 31-32, discussion: line 738). I assume that they refer to the lack of a significant interaction between emotion condition and autistic traits (and not the lack of a main effect of autistic traits) here, but this should be clarified to the reader.

Author's response:

As suggested, we have clarified this text by explicitly mentioning that there was “no two-way interaction between AQ and emotional conditions” (L33-34, 703-704).

Abstract:

4. As valence itself is a neutral word, “higher subjective valence” (line 26) should be defined in its directionality, i.e. more positive.

Author's response: 

We have implemented the reviewer’s suggestion (L27).

5. I would recommend the authors to already clarify in the abstract that emotional contagion is measured via subjective experiential ratings, i.e. by mentioning it in parentheses in line 21.

Author's response: 

As suggested, we have described in the Abstract that we used subjective experiential valence and arousal ratings to estimate emotional contagion, and used ZM and CS EMG measurements to estimate spontaneous facial mimicry (L21-22).

6. Since especially the contrast in the presentation effect between e.g. individuals low in autistic traits vs. high in autistic traits is interesting, I personally would not say that live interaction designs are especially valuable for individuals low in autistic traits (line 34, similar trait empathy).

Author's response: 

We have implemented the reviewer’s suggestion and changed the last sentence in the abstract to be more neutral: “Our findings imply that studies using live interactions could yield valuable insights when investigating social behaviors in the context of trait emotional empathy and autistic traits.” (L34-36)

Introduction:

7. It is unclear why findings on modulated facial mimicry by autistic trait levels which are reported in the respective paragraph would “imply a positive correlation between trait emotional empathy and both emotional contagion and spontaneous facial mimicry” (lines 197-198). Further, in the next sentence, it is not very clear why and how the authors relate findings with regard to emotional contagion and facial mimicry to each other, given that they claim to look at them as separate constructs. This might, however, only be due to the way in which it is phrased.

Author's response: 

We have removed the sentence mentioning the trait emotional empathy and focused on the effect of autistic traits. We also revised the following sentence, which provides a summary of the current understanding of ASD diagnosis-facial mimicry and ASD diagnosis-emotional contagion relationships (L130-133).

Results:

8. I would recommend the authors to be careful when reporting significant main effects in the context of a significant interaction in which the main effect is involved (e.g. lines 614-615) since they might not be apparent in the absence of the interaction.

Author's response: 

We thank the reviewer for reminding us of this, and have revised the Results section accordingly: significant main effects involved in an interaction effect are no longer reported. We have added this information to the “Statistical analysis” section in the Methods (L417-420).

9. In line with Reviewer 1’s comment on the previous revision, the authors could be more precise in their phrasing. For example, they refer to “significantly more ZM responses” in the description of their results while not using count data but continuous variables (strength of activation) in their analyses.

Author's response: 

As suggested, we now use “stronger/weaker” to describe muscular responses.

Discussion:

10. As the results greatly differ between the model fits including all data vs. remaining data after model diagnostics, the robustness of effects should be discussed.

Author's response: 

We have included a paragraph in the limitation sub-section discussing how decisions regarding model diagnostics have affected the results (L826-839). As new methods for linear mixed model diagnostics are currently being developed, and because no concrete guidelines have been formulated, we encourage readers to consider the reported results both with and without model diagnostics under this context, and to reanalyze the deposited data as new algorithms and treatment methods emerge.

- Given the introduction and hypotheses, I would further appreciate if the following points would be touched upon in the introduction:

o Why were no modulatory effects of any trait dimension found in the model on the corrugator data? It is only very briefly mentioned that the corrugator is generally less mimicked.

Author's response: 

Indeed, contrary to Hermans et al. [9], we did not observe an AQ × emotional condition interaction effect on the CS responses. One possible explanation for this is that Hermans et al. included 18 participants in their high AQ group, and 16 in their low AQ group, with extremely high and low AQ scores, respectively, among a sample of 366 participants. In the present study, we did not focus on the extremes of the AQ spectrum. Therefore, the effect would have been much smaller than that in Hermans et al. Similarly, Dimberg et al. selected the 72 highest and 72 lowest QMEE-ranked participants from a large sample (> 500 participants) for their analysis of trait emotional empathy and its interaction with emotional condition. In contrast, we did not select participants with extremely high or low trait emotional empathy, which might have been required for these interaction effects to be detected. Furthermore, such effects might not be linear in an average population. Using our data as a pilot study for future studies, simulation using simr showed no increase in power with a sample size of up to 200 for these two-way interactions.

We have added this information to the Limitations section of the Discussion (L805-815).

o Even though it is clearly mentioned in the abstract, there is no potential explanation on why the presentation condition had no impact on the modulatory effect of empathy on contagion and facial mimicry. As this was one of the primary hypotheses, I would have appreciated a short speculation.

Author's response: 

We simulated the required sample size to detect three-way interaction effects in the population using the R package simr for a sample size of up to 200 participants.

Regarding the valence ratings, a three-way interaction involving IRIPD would achieve 70% power with a sample size of 200, while that involving AQ would reach 60% power.

Regarding the arousal ratings, a three-way interaction involving IRIPD would achieve 80% power with a sample size of 150.

Regarding the ZM responses, a three-way interaction involving IRIEC would achieve 70% power with a sample size of 150, and the power would not increase with a sample size of 200. A three-way interaction involving AQ would achieve 70% power with a sample size of 180, and the power would not increase with a sample size of 200.

The simulation results suggested that our ability to make inferences regarding three-way interactions was limited. Also, any effect size was likely to be very small. A much larger sample size would be required to reliably detect such effects. We have added this information to the Limitations section of the Discussion (L792-804).

o While arguing that emotional contagion and facial mimicry should be regarded as separate constructs in the introduction, the authors relate them to each other again in the discussion of the findings on modulatory effects of autistic traits (lines 728-737). Given that the relationship between emotional contagion and mimicry was quite extensively reviewed in the introduction and their separation strongly favored, the authors might consider referring back to this discussion, given their new findings and interpretation.

Author's response: 

That section discusses the similar pattern of results of the valence ratings and ZM responses in the context of the two-way AQ × presentation condition interaction. We discussed the extent of the social smile, rather than facial mimicry, which involves emotional condition-specific responses (ZM in the positive condition and CS in the negative condition). To avoid confusion, we have added sub-headings for differentiation from the later discussion concerning the fact that AQ does not modulate facial mimicry.

In our review of the literature on the relationship between emotional contagion and emotional facial mimicry, we included theories on primitive emotional contagion and social regulator perspectives. We cite Scherer’s comments regarding the ongoing theoretical conflicts [5]: facial mimicry is subject to multilevel regulation, and emotional contagion alone cannot fully account for facial mimicry, as made clear in the Introduction of the revised manuscript (L85-87). We did not rule out a possible influence of primitive emotional contagion; specifically, the “facial feedback hypothesis”, which is supported by empirical data from a meta-analysis (L664-671).

11. While the “audience effect” (line 703 and following) is surely relevant in research on social interactions, the presence of others (third person perspective) is qualitatively different from interacting with another person (second person perspective) as it occurred in this study. Therefore, I would be careful to claim that stronger smiles are displayed in the live condition due to an “audience effect”.

Author's response: 

We agree that the present second-person perspective paradigm involves more than the mere presence of others, and should have led to greater social preparedness. We have added this information to the “Social smile, reputation management, and autistic traits” sub-section following the discussion of the function of social smiles (L688-691).

12. In some parts, the authors might consider being more concise, i.e.

- “smiled equally” (line 726): as ZM activity is very close to 0, it could as well be said that they did not smile in both conditions. I would therefore try to phrase it more neutral.

- “rather calming”(line 756): were the ratings of the negative stimuli generally indeed <5 (on the calming side of the affect grid) or were they simply less arousing than the positive stimuli?

Author's response: 

We have changed the first sentence (L699-701) to “participants with high AQ values showed comparable levels of smiles (or no smiles) when viewing video stimuli and real persons.”

We have also changed the second sentence (L726-727) to “…negative stimuli, in contrast to positive stimuli, were less arousing or more calming, irrespective of presentation condition.”

 

Reference

1. Searle SR, Speed FM, Milliken GA. Population Marginal Means in the Linear Model: An Alternative to Least Squares Means. Am Stat. 1980;34: 216–221. doi:10.1080/00031305.1980.10483031

2. Hess U, Fischer A. Emotional mimicry as social regulator: theoretical considerations. Cogn Emot. 2022; 1–9. doi:10.1080/02699931.2022.2103522

3. Kret ME, Akyüz R. Mimicry eases prediction and thereby smoothens social interactions. Cogn Emot. 2022;36: 794–798. doi:10.1080/02699931.2022.2110452

4. Bernhold QS, Giles H. Emotional mimicry: a communication accommodation approach. Cogn Emot. 2022;36: 799–804. doi:10.1080/02699931.2022.2109599

5. Scherer KR. The mystery of emotional mimicry: multiple functions and processing levels in expression imitation. Cogn Emot. 2022;36: 781–784. doi:10.1080/02699931.2022.2115614

6. Mauersberger H, Hess U. When smiling back helps and scowling back hurts: individual differences in emotional mimicry are associated with self-reported interaction quality during conflict interactions. Motiv Emot. 2019;43: 471–482. doi:10.1007/s11031-018-9743-x

7. Sato W, Yoshikawa S. Spontaneous facial mimicry in response to dynamic facial expressions. Cognition. 2007;104: 1–18. doi:10.1016/j.cognition.2006.05.001

8. Fujimura T, Sato W, Suzuki N. Facial expression arousal level modulates facial mimicry. International Journal of Psychophysiology. 2010;76: 88–92. doi:10.1016/j.ijpsycho.2010.02.008

9. Hermans EJ, van Wingen G, Bos PA, Putman P, van Honk J. Reduced spontaneous facial mimicry in women with autistic traits. Biol Psychol. 2009;80: 348–353. doi:10.1016/j.biopsycho.2008.12.002

---

## [Decision Letter · Decision Letter 3]

5 Apr 2023

PONE-D-21-34896R3An Investigation of the Modulatory Effects of Empathic and Autistic Traits on Emotional and Facial Motor Responses during Live Social InteractionsPLOS ONE

Dear Dr. Hsu,

Thank you for submitting your manuscript to PLOS ONE. I have received reviews from two experts in the field. We all agree that this is important research and the manuscript has been greatly improved since the last revision. As you will see from the reviewers' comments, there are several editorial revisions that will need to be made, as well as some concern regarding the sample. In particular, one reviewer was concerned about the overall sample size, potential limits to power, gender limitations, and that this manuscript reports on data that may have been partially published elsewhere. I have confirmed with the editorial office that your current sample conforms to the PLOS ONE policy to not publish previously reported data due to the inclusion of additional data. However, you will need to respond to the reviewers additional concerns. Specifically, this reviewer expresses some concern about the sample size and gender composition, and requests additional data be collected. Given that psychophysiological data can be quite time consuming and costly to collect, please respond to this concern, especially as it relates to your ability to collect additional data and the potential impact that this has on your ability to make inferences from your data.

We look forward to receiving your revised manuscript.

Kind regards,

Eric J. Moody, Ph.D.

Academic Editor

PLOS ONE

Journal Requirements:

Reviewers' comments:

Reviewer's Responses to Questions

**Comments to the Author**

1. If the authors have adequately addressed your comments raised in a previous round of review and you feel that this manuscript is now acceptable for publication, you may indicate that here to bypass the “Comments to the Author” section, enter your conflict of interest statement in the “Confidential to Editor” section, and submit your "Accept" recommendation.

Reviewer #3: (No Response)

Reviewer #4: All comments have been addressed

2. Is the manuscript technically sound, and do the data support the conclusions?

Reviewer #3: Partly

Reviewer #4: Yes

3. Has the statistical analysis been performed appropriately and rigorously? 

Reviewer #3: Yes

Reviewer #4: Yes

4. Have the authors made all data underlying the findings in their manuscript fully available?

Reviewer #3: Yes

Reviewer #4: Yes

5. Is the manuscript presented in an intelligible fashion and written in standard English?

Reviewer #3: Yes

Reviewer #4: Yes

6. Review Comments to the Author

Reviewer #3: I would like to thank the editor for the invitation to read the article “An Investigation of the Modulatory Effects of Empathic and Autistic Traits on Emotional and Facial Motor Responses during Live Social Interactions”.

The study assessed the amplitude of emotional contagion (subjective valence and arousal) and the electromyographic activities of facial muscles (both zygomaticus major and corrugator supercilia) to estimate spontaneous facial mimicry in response to emotional expressions displayed under live or video conditions. In addition, the authors considered the role of some individual differences (empathic concern, personal and autistic traits).

I found the study to address a relevant research question and to be well-conducted. This is also the third revision of the manuscript. In my opinion, the authors have invested a significant amount of time and effort into addressing the comments made by the previous reviewers, leading to an improvement of the manuscript.

However, my main concerns are about the sample size, which is rather low (n = 50) to examine the potential moderation of several variables on different outcomes, and the fact that this study integrates data that were already published and discussed before (half of the sample includes data from the previous dataset, that is, the first publication included 23 participants and the present study only added 27 participants). Furthermore, the sample only included women. It would have been more relevant if the authors had collected a much larger sample to also include men. This can be particularly relevant because the research question includes traits that are important to both women and men.

Therefore, instead of acknowledging these concerns as limitations to be addressed in the future, I would recommend the authors to address them in this study and collect more data to provide more robust findings. In my view, these concerns can easily be solved by collecting a much larger sample and preferably not including the previous participants. It will also allow examining the nonsignificant two- and three-way interactions that the authors acknowledged might have been due to the low sample size.

With a different sample, the authors will also be able to compare some of the current findings with the previous publication, which is also important for identifying patterns that may have been missed in the previous sample. To sum up, given the importance of the research questions and the limitations also acknowledged by the authors concerns, but that can be solved with more data, I would strongly recommend that the authors consider collecting more data to strengthen the study's conclusions.

In conclusion, I found the research questions relevant and the methodology sound, but it is crucial to consider further data collection to provide more robust conclusions. By doing so, the study would also be able to have a greater impact among the research community.

Reviewer #4: I really appreciate the authors’ revised version of the document. The reviewer comments have been well addressed and the contents are presented more clearly. There are still a few minor concerns especially regarding the readability of the paper:

Abstract:

1) L17: Similar to L42 „moreover” (instead of “however”) seems to be more appropriate here.

2) L24: “Individual differences included…”: “Individual differences measures” might be clearer.

3) L25: The term “live conditions” has not been introduced. In the sentences before it was described as “image relay system to present live performances”. The usage of consistent terms would increase the comprehensibility.

4) L33: The abbreviation “AQ” should be introduced before the first use.

5) L34ff: “…could yield valuable insights” is a bit vague. Can it be made more precise on the basis of the core results of the study? What is the main benefit/implication of the paper?

Introduction:

6) L83/84: What is meant by “this perspective”?

7) L174ff: Before deriving the main research questions (L182ff) there a many methodological details concerning the conducted study (e.g. details regarding the sample). It should be considered to move these methodological aspects to the method section and to focus in the last section of the introduction on the main research questions.

Materials and methods:

8) L231: To make even more clear that IRI is your measure for trait emotional empathy you could explicitly name it in the sub-headline, for example: Trait emotional empathy: IRI.

9) L296ff: Since the affect grid measures of valence and arousal are important variables in the analysis section (and regarding the main research questions), it could be considered to include them in the instruments section (e.g. after AQ) rather than in the paradigm and procedures section.

10) L350ff: Why did you use both paired samples t-Test and the Wilcoxon rank test? You could mention the main reason (data quality, scale level…) or omit one of the two analyses.

Discussion:

11) L636ff: The first sentence summarizes the main research question of the paper. The following sentence combines information concerning the analysis strategy and specific results (interaction emotional conditions x presentation) that has not been introduced in the sentence before. It would improve the readability of the section if the aspects were clearly separated: main research question, analysis strategy, report of the main results regarding different facets of the main research questions (in different sentences).

12) A general recommendation to improve the readability of the discussion section: Due to the complex research design, there a many different results (two-way-/three-way interactions for different dependent variables) that have to be reported. Furthermore, the various results should be clearly related to the questions/hypotheses formulated at the end of the introduction section (L182ff). It might be helpful to number each research question and then systematically refer to these questions/hypotheses in the discussion sections, for example: regarding research question 2 postulating effect b, a two-way interaction between … could be found… Overall, it should be made clear which research question is addressed by reporting results for specific (interaction) effects.

13) Significant interactions are reported at several points in the discussion but are not fully explained. For example: L641ff: interaction emotional condition x presentation, L641 does not explain the nature of the interaction; similar: L647ff, L654ff. When a significant interaction is reported, the main characteristic of the interaction should be named (e.g. difference in this condition of variable a, but not in this condition…).

14) L657ff: After the first section of the discussion it is not entirely clear why and according to which criteria sub-headings have been chosen in the following. There are many interactions, dependent variables and research questions. It is recommended to reconsider the structuring of the results summary and interpretation within the discussion section in order to further improve the readability (see also comment 12).

7. PLOS authors have the option to publish the peer review history of their article (what does this mean?). If published, this will include your full peer review and any attached files.

Reviewer #3: No

Reviewer #4: No

---

## [Author Response · Author response to Decision Letter 3]

17 Apr 2023

Reviewer #3: I would like to thank the editor for the invitation to read the article “An Investigation of the Modulatory Effects of Empathic and Autistic Traits on Emotional and Facial Motor Responses during Live Social Interactions”.

The study assessed the amplitude of emotional contagion (subjective valence and arousal) and the electromyographic activities of facial muscles (both zygomaticus major and corrugator supercilia) to estimate spontaneous facial mimicry in response to emotional expressions displayed under live or video conditions. In addition, the authors considered the role of some individual differences (empathic concern, personal and autistic traits).

I found the study to address a relevant research question and to be well-conducted. This is also the third revision of the manuscript. In my opinion, the authors have invested a significant amount of time and effort into addressing the comments made by the previous reviewers, leading to an improvement of the manuscript.

However, my main concerns are about the sample size, which is rather low (n = 50) to examine the potential moderation of several variables on different outcomes, and the fact that this study integrates data that were already published and discussed before (half of the sample includes data from the previous dataset, that is, the first publication included 23 participants and the present study only added 27 participants). Furthermore, the sample only included women. It would have been more relevant if the authors had collected a much larger sample to also include men. This can be particularly relevant because the research question includes traits that are important to both women and men.

Therefore, instead of acknowledging these concerns as limitations to be addressed in the future, I would recommend the authors to address them in this study and collect more data to provide more robust findings. In my view, these concerns can easily be solved by collecting a much larger sample and preferably not including the previous participants. It will also allow examining the nonsignificant two- and three-way interactions that the authors acknowledged might have been due to the low sample size.

With a different sample, the authors will also be able to compare some of the current findings with the previous publication, which is also important for identifying patterns that may have been missed in the previous sample. To sum up, given the importance of the research questions and the limitations also acknowledged by the authors concerns, but that can be solved with more data, I would strongly recommend that the authors consider collecting more data to strengthen the study's conclusions.

In conclusion, I found the research questions relevant and the methodology sound, but it is crucial to consider further data collection to provide more robust conclusions. By doing so, the study would also be able to have a greater impact among the research community.

Authors’ reply: We thank the reviewer for acknowledging our effort in this manuscript. 

We agree with the reviewer that, ideally, it would have been more conclusive to test up to 200 participants as suggested by the power estimation in the “Sample selection and sample size” subsection of the Limitation section. However, this isn't easy in terms of time and budget. Preparing for physiological (EMG) recordings takes time to ensure adequate data quality. For each testing session, both the model and the participant of the dyad need to be monetarily compensated. The budget issue was the main reason we stopped at a sample size of 50 for the current study.

Furthermore, previous studies suggested that dyads of sender-perceiver with different gender combinations might yield different facial mimicry patterns. It is known that females showed stronger smile mimicry than males. When involving between-gender dyads, other confounding factors, such as attractiveness, motivations to flirt, or extroverted traits, would further modulate facial mimicry's pattern and effect size. Therefore, the required sample size for different gender pairs (female sender-male perceiver, male sender-male perceiver, male sender-female perceiver) might be different (presumably even larger) to detect the three-way interaction effect of interest in our paradigm. The introduction of further possible confounding factors would also further increase the complexity of the statistical models. Therefore, we consider that the live effect in different dyadic gender combinations needs to be evaluated separately.

We have added the above information in the “Sample selection and sample size” (lines 828-830) and “Generalizability” (lines 871-876) subsections of the Limitation section.

Reviewer #4: I really appreciate the authors’ revised version of the document. The reviewer comments have been well addressed and the contents are presented more clearly. There are still a few minor concerns especially regarding the readability of the paper:

Abstract:

1) L17: Similar to L42 „moreover” (instead of “however”) seems to be more appropriate here.

2) L24: “Individual differences included…”: “Individual differences measures” might be clearer.

Authors’ reply: We followed the reviewer’s suggestions.

3) L25: The term “live conditions” has not been introduced. In the sentences before it was described as “image relay system to present live performances”. The usage of consistent terms would increase the comprehensibility.

Authors’ reply: For consistency, we changed the term “live conditions” to “live performances”.

4) L33: The abbreviation “AQ” should be introduced before the first use.

Authors’ reply: We changed “AQ” to the full term “autism spectrum quotient.”

5) L34ff: “…could yield valuable insights” is a bit vague. Can it be made more precise on the basis of the core results of the study? What is the main benefit/implication of the paper?

Authors’ reply: To be more specific, we changed “valuable insights” to “different patterns of trait–behavior relationships” based on our findings.

Introduction:

6) L83/84: What is meant by “this perspective”?

Authors’ reply: We now specify that “the social regulator perspective also proposed that….”

7) L174ff: Before deriving the main research questions (L182ff) there a many methodological details concerning the conducted study (e.g. details regarding the sample). It should be considered to move these methodological aspects to the method section and to focus in the last section of the introduction on the main research questions.

Authors’ reply: We deleted the sentence about the three-minute conversation before the actual experiment since this information is also described in the “Paradigm and procedures” section (lines 287-289). We moved the sentence explaining why we recruited only female participants to the beginning of the “Participant” section (lines 230-231)

Materials and methods:

8) L231: To make even more clear that IRI is your measure for trait emotional empathy you could explicitly name it in the sub-headline, for example: Trait emotional empathy: IRI.

Authors’ reply: We followed the reviewer’s suggestion. We also changed the following sub-headline to “Autistic traits: AQ.” 

9) L296ff: Since the affect grid measures of valence and arousal are important variables in the analysis section (and regarding the main research questions), it could be considered to include them in the instruments section (e.g. after AQ) rather than in the paradigm and procedures section.

Authors’ reply: We followed the reviewer’s suggestion and moved the information related to the Affect Grid to a subsection “Subjective experiential ratings: the Russel ‘affect grid’” after the subsection “Autistic traits: AQ.”

10) L350ff: Why did you use both paired samples t-Test and the Wilcoxon rank test? You could mention the main reason (data quality, scale level…) or omit one of the two analyses.

Authors’ reply: Because the distribution of the anger and sadness ratings did not violate the assumption of normality (anger ratings: Shapiro-Wilk W = 0.96, p = 0.39; sadness ratings: Shapiro-Wilk W = 0.96, p = 0.30), we updated this information (lines 366-367) and omitted results of the nonparametric Wilcoxon rank test.

Discussion:

11) L636ff: The first sentence summarizes the main research question of the paper. The following sentence combines information concerning the analysis strategy and specific results (interaction emotional conditions x presentation) that has not been introduced in the sentence before. It would improve the readability of the section if the aspects were clearly separated: main research question, analysis strategy, report of the main results regarding different facets of the main research questions (in different sentences).

Authors’ reply: The second sentence of the discussion was extremely long. We followed the suggestion and separated the information concerning the analysis strategy and results into separate sentences (lines 651-655).

12) A general recommendation to improve the readability of the discussion section: Due to the complex research design, there a many different results (two-way-/three-way interactions for different dependent variables) that have to be reported. Furthermore, the various results should be clearly related to the questions/hypotheses formulated at the end of the introduction section (L182ff). It might be helpful to number each research question and then systematically refer to these questions/hypotheses in the discussion sections, for example: regarding research question 2 postulating effect b, a two-way interaction between … could be found… Overall, it should be made clear which research question is addressed by reporting results for specific (interaction) effects.

Authors’ reply: We have reorganized the discussion until the section on Limitations. We first defined four research questions at the end of the introduction. In the first part of the discussion, we summarized the main results in relation to the research questions: none of the expected modulatory effects of live conditions were supported by our data. The following discussions were done in the sequence of the four research questions on relevant findings, e.g., although the presentation conditions did not modulate the relationship between autistic traits and emotional contagion/facial mimicry, presentation conditions modulated the relationship between autistic traits and smiling behavior.

13) Significant interactions are reported at several points in the discussion but are not fully explained. For example: L641ff: interaction emotional condition x presentation, L641 does not explain the nature of the interaction; similar: L647ff, L654ff. When a significant interaction is reported, the main characteristic of the interaction should be named (e.g. difference in this condition of variable a, but not in this condition…).

Authors’ reply: We added descriptions for the nature of interactions for the abovementioned points (lines 655-656, 665-668, 672-673).

14) L657ff: After the first section of the discussion it is not entirely clear why and according to which criteria sub-headings have been chosen in the following. There are many interactions, dependent variables and research questions. It is recommended to reconsider the structuring of the results summary and interpretation within the discussion section in order to further improve the readability (see also comment 12).

Authors’ reply: We have restructured the discussion to address concerns raised in comments 12 and 14. Please see our reply to comment 12 for details.

---

## [Decision Letter · Decision Letter 4]

22 May 2023

PONE-D-21-34896R4An Investigation of the Modulatory Effects of Empathic and Autistic Traits on Emotional and Facial Motor Responses during Live Social InteractionsPLOS ONE

Dear Dr. Hsu,

Thank you for your responses to the last round of reviews. I first want to I acknowledge that you have made several rounds of revisions that have vastly improved the paper, and spent considerable time with this paper under review. However, as you will see, there is ongoing concern related to your statistical power and the appropriateness of the tests you used. Reviewer 3 offers a couple of options that may prove satisfactory. I believe one option, to scale back some of the tests, can feasibly be done within this current submission. However, the other option may require more time, effort and money than can be accommodated in the current process. Therefore, I would like to leave the decision to you as to how you would like to proceed to you. If you would like to revise and resubmit, I will gladly entertain another round of revisions and reviewer 3 has indicated that she is willing to review a revised manuscript as well. However, if you prefer to pursue other options, whether that is additional data collection or another venue, that is acceptable as well. In that case, it is probably most appropriate for you to withdraw the submission so the current process can be concluded, which will allow you the latitude to pursue additional funding, or any other option you choose. Regardless of your decision, I appreciate your dedication to PLOS One and the review process. I know this has been a lot work for your team, however, I believe your research will make an important contribution and I look forward to knowing what your next steps will be. If you choose to resubmit, please submit your revised manuscript by Jul 06 2023 11:59PM. If you will need more time than this to complete your revisions, please reply to this message or contact the journal office at plosone@plos.org. Please include the following items when submitting your revised manuscript:A rebuttal letter that responds to each point raised by the academic editor and reviewer(s). You should upload this letter as a separate file labeled 'Response to Reviewers'.A marked-up copy of your manuscript that highlights changes made to the original version. You should upload this as a separate file labeled 'Revised Manuscript with Track Changes'.An unmarked version of your revised paper without tracked changes. You should upload this as a separate file labeled 'Manuscript'.

We look forward to receiving your revised manuscript.

Kind regards,

Eric J. Moody, Ph.D.

Academic Editor

PLOS ONE

Reviewers' comments:

Reviewer's Responses to Questions

**Comments to the Author**

1. If the authors have adequately addressed your comments raised in a previous round of review and you feel that this manuscript is now acceptable for publication, you may indicate that here to bypass the “Comments to the Author” section, enter your conflict of interest statement in the “Confidential to Editor” section, and submit your "Accept" recommendation.

Reviewer #3: All comments have been addressed

Reviewer #4: All comments have been addressed

2. Is the manuscript technically sound, and do the data support the conclusions?

Reviewer #3: Partly

Reviewer #4: Yes

3. Has the statistical analysis been performed appropriately and rigorously? 

Reviewer #3: No

Reviewer #4: Yes

4. Have the authors made all data underlying the findings in their manuscript fully available?

Reviewer #3: Yes

Reviewer #4: Yes

5. Is the manuscript presented in an intelligible fashion and written in standard English?

Reviewer #3: Yes

Reviewer #4: Yes

6. Review Comments to the Author

Reviewer #3: Dear Authors, thank you for your response regarding the concerns I raised in my previous review. I fully understand the challenges involved in data collection, particularly when it includes physiological measurements. However, I wish to emphasize that the main focus of our scientific mission should be the reliability of the conclusions we share with the community. These aspects are influenced by the statistical power of a given study, since these estimates serve as guides, indicating the necessary sample size to reliably detect an effect, if it indeed exists.

I appreciate your acknowledgment of this principle in the "Sample selection and size" subsection of your paper. However, I encourage you to contemplate the problems of not meeting these power estimates. Conducting statistical tests with insufficient power may increase the risk of Type II errors, leading to the possible overinterpretation of nonsignificant findings related to the failure in detecting effects. If your study's intention is to test moderation effects, a larger sample size is still required based on your power estimates to the robustness of these tests. Despite the constraints you face, I believe addressing this issue is of paramount importance.

The statement in your paper, “If an effect size exists, it is likely to be very small; thus, a much larger sample size is required to detect such an effect reliably...", while acknowledging the need for more data, may not send the most constructive message to the research community. Similarly, since the primary focus of your paper are not the difficulties in data collection, the sentence, "This could be difficult for many researchers in terms of time and budget. Preparing for physiological (EMG) recordings takes time to ensure adequate data quality. For each testing session, both the model and the participant of the dyad need to be monetarily compensated.", might not be relevant to the core message of your study.

Again, the emphasis should be on the reliability of the conclusions drawn from your study. Considering these constraints, I see two potential solutions:

Reframe your conclusions to align with the statistical power available from your current sample size of 50. This would involve narrowing the scope of your statistical tests (e.g. not addressing moderations).

Alternatively, seek additional funding or resources to support further data collection, thereby enhancing the study's power.

In closing, I commend the thoughtful work you have accomplished thus far, and I firmly believe that addressing these issues will increase the value of your contribution to the field. I look forward to reviewing your revised manuscript, if necessary.

Reviewer #4: Thank you very much for the opportunity to review this interesting and relevant paper again.

The authors did a very good job with the revision. From my point of view, all concerns were thoroughly and adequately addressed. The structure of several sections (theory, discussion) is much clearer, and the description of the interactions is much more precise now.

Of course, a larger sample size would provide more robust insights. Considering the high effort of collecting physiological data, I think the current sample size is justifiable. Limitations based on the sample size were sufficiently pointed out in the revised version of the document.

7. PLOS authors have the option to publish the peer review history of their article (what does this mean?). If published, this will include your full peer review and any attached files.

Reviewer #3: No

Reviewer #4: No

---

## [Author Response · Author response to Decision Letter 4]

30 Jun 2023

Reviewer #3: 

Point 1: Dear Authors, thank you for your response regarding the concerns I raised in my previous review. I fully understand the challenges involved in data collection, particularly when it includes physiological measurements. However, I wish to emphasize that the main focus of our scientific mission should be the reliability of the conclusions we share with the community. These aspects are influenced by the statistical power of a given study, since these estimates serve as guides, indicating the necessary sample size to reliably detect an effect, if it indeed exists.

I appreciate your acknowledgment of this principle in the "Sample selection and size" subsection of your paper. However, I encourage you to contemplate the problems of not meeting these power estimates. Conducting statistical tests with insufficient power may increase the risk of Type II errors, leading to the possible overinterpretation of nonsignificant findings related to the failure in detecting effects. If your study's intention is to test moderation effects, a larger sample size is still required based on your power estimates to the robustness of these tests. Despite the constraints you face, I believe addressing this issue is of paramount importance.

The statement in your paper, “If an effect size exists, it is likely to be very small; thus, a much larger sample size is required to detect such an effect reliably...", while acknowledging the need for more data, may not send the most constructive message to the research community. Similarly, since the primary focus of your paper are not the difficulties in data collection, the sentence, "This could be difficult for many researchers in terms of time and budget. Preparing for physiological (EMG) recordings takes time to ensure adequate data quality. For each testing session, both the model and the participant of the dyad need to be monetarily compensated.", might not be relevant to the core message of your study.

Again, the emphasis should be on the reliability of the conclusions drawn from your study. Considering these constraints, I see two potential solutions:

Reframe your conclusions to align with the statistical power available from your current sample size of 50. This would involve narrowing the scope of your statistical tests (e.g. not addressing moderations).

Alternatively, seek additional funding or resources to support further data collection, thereby enhancing the study's power.

In closing, I commend the thoughtful work you have accomplished thus far, and I firmly believe that addressing these issues will increase the value of your contribution to the field. I look forward to reviewing your revised manuscript, if necessary.

Authors’ reply to point 1: 

We considered the reviewer’s suggestions and addressed the issues as follows:

1. We included the rating and EMG data from another published fMRI study using the same design [1], thus making the sample size of the present study 94. We also included the random factor “Type” (behavioral vs. neuroimaging) when model comparisons suggested the necessity, as in the valence ratings.

2. An estimation of the required sample size for two-tailed t-tests of multiple regression coefficients with 12 predictors with a medium effect size, partial r2 = 0.09, f2 = 0.0989 [2,3], to achieve 80% of power using G*Power Version 3.1.9.6 suggested that a sample size of 82 was required. In addition, the sensitivity analysis for two-tailed t-tests of multiple regression coefficients with 12 predictors and a sample size 94 showed that the present study has 80% of power for a minimum detectable effect size of f2 = 0.0855, r2 = 0.0788.

3. Due to the inversed relationship between the observed power and the p-values, a very small or null effect would invariably result in a large p-value and a low observed power [4,5]. Thus, making inferences based on the effect size under such circumstances is more informative. While it still requires a large sample size to make accurate estimations of small effect sizes, the required sample size for precise estimation is smaller than the required sample size to achieve high power. For a small effect size of r2 = 0.01, only a sample size of 61 is required to achieve the point of stability with 80% confidence for the effect size estimation, while a sample size of 394 is required to observe such an effect with an alpha of 0.05 [6]. Therefore, the sample size of the present study is also more than sufficient to accurately estimate small effect sizes and make inferences based on that.

4. We reduced the number of fixed-effect predictors by excluding IRIPD and associated interaction effects, making the number of fixed-effect predictors 12. IRIPD was significantly correlated with IRIEC and AQ in the expanded sample and easily resulted in overfitting (singular fit) or failure to converge in model estimation.

5. We also simplified our approach to model diagnostics. We iteratively increased model complexity using the complete dataset, and ran robust estimations of the final model [7] to check the significant result patterns when underweighting influential data points. When the result patterns of the robust and non-robust estimations are identical, we report and discuss the result details of the non-robust estimation of the complete dataset, keeping as much data and participants in the analysis as possible to retain statistical power. In the case of CS responses, the result patterns of the robust and non-robust estimations were not identical; we performed model diagnostics as before to exclude influential data points, which resulted in the same result pattern as the robust estimation of the complete dataset. We report and discuss the non-robust result details with model diagnostics for CS responses. Results of robust estimation were included in the supplementary tables (S1-S4 Tables).

We observed two-way interactions between emotional and presentation conditions in all four measurements (valence and arousal ratings and ZM and CS responses). We also observed two-way interactions between IRIEC and emotional conditions in valence and arousal ratings, and CS responses. We still observed no three-way interactions among traits, emotional and presentation conditions. The effect size and confidence interval of effect size estimates suggested that the effects of three-way interactions are indeed small. As the results are more straightforward, the Results and Discussion sections were also much reduced, including paragraphs that the reviewer considered irrelevant. 

Reviewer #4: Thank you very much for the opportunity to review this interesting and relevant paper again.

The authors did a very good job with the revision. From my point of view, all concerns were thoroughly and adequately addressed. The structure of several sections (theory, discussion) is much clearer, and the description of the interactions is much more precise now.

Of course, a larger sample size would provide more robust insights. Considering the high effort of collecting physiological data, I think the current sample size is justifiable. Limitations based on the sample size were sufficiently pointed out in the revised version of the document.

Authors’ reply: We thank the reviewer for the comments.

Bibliography

1. Hsu C-T, Sato W, Kochiyama T, Nakai R, Asano K, Abe N, et al. Enhanced mirror neuron network activity and effective connectivity during live interaction among female subjects. NeuroImage. 2022;263: 1–19. doi:10.1016/j.neuroimage.2022.119655

2. Cohen J. Statistical Power Analysis for the Behavioral Sciences. 2nd ed. New York: Routledge; 1988. doi:10.4324/9780203771587

3. Gignac GE, Szodorai ET. Effect size guidelines for individual differences researchers. Pers Individ Differ. 2016;102: 74–78. doi:10.1016/j.paid.2016.06.069

4. Hoenig JM, Heisey DM. The Abuse of Power: The Pervasive Fallacy of Power Calculations for Data Analysis. The American Statistician. 2001;55: 19–24. doi:10.1198/000313001300339897

5. Greenland S. Nonsignificance Plus High Power Does Not Imply Support for the Null Over the Alternative. Annals of Epidemiology. 2012;22: 364–368. doi:10.1016/j.annepidem.2012.02.007

6. Lakens D, Evers ERK. Sailing From the Seas of Chaos Into the Corridor of Stability: Practical Recommendations to Increase the Informational Value of Studies. Perspect Psychol Sci. 2014;9: 278–292. doi:10.1177/1745691614528520

7. Koller M. robustlmm : An R Package for Robust Estimation of Linear Mixed-Effects Models. J Stat Soft. 2016;75. doi:10.18637/jss.v075.i06

---

## [Editor Report · Decision Letter 5]

2 Aug 2023

PONE-D-21-34896R5An Investigation of the Modulatory Effects of Empathic and Autistic Traits on Emotional and Facial Motor Responses during Live Social InteractionsPLOS ONE

Dear Dr. Hsu,

Thank you for submitting your manuscript to PLOS ONE. After careful consideration, we feel that it has merit but does not fully meet PLOS ONE’s publication criteria as it currently stands. Therefore, we invite you to submit a revised version of the manuscript that addresses the points raised during the review process.

We look forward to receiving your revised manuscript.

Kind regards,

Eric J. Moody, Ph.D.

Academic Editor

PLOS ONE

Journal Requirements:

**Additional Editor Comments:**

Thank you for your continued dedication to the peer review process. I appreciate the substantial revisions you have made. Unfortunately, Reviewer 3 is not available to review your most recent changes. However, I have carefully reviewed your revised manuscript, as well as all previous comments. Based on this, I see only two minor points that would need to be addressed before your manuscript meets the publication criteria for PLOS ONE.

First, I appreciated your careful attention to the comments about statistical power. While increasing your sample size by including data from a related study certainly addressed the reviewer's concern about power, I worry that it also adds an additional confound that may introduce problems if the experimental paradigms introduce unknown systematic differences. For instance, if participants are laying in a MRI magnet, this may alter their attention or response to stimuli in a number of ways. They may be more anxious overall, which could alter their attention to positive or negative stimuli. Or the simple act of laying down may impact the magnitude of muscle activation. Of course, there is no way to know for sure, which is the danger. That said, you may want to keep this larger sample to maintain power. In this case, you might consider ways to verify that they data collection procedures do not impact your models. This could be done by comparing the two data collection methods directly, conducting a sensitivity analysis, or including the data collection method as a factor in the model. However, I think it is also acceptable to just report on your original sample (N=50) as these tests appeared to be sufficiently powered to address your main findings. If you choose to report on your original sample, however, I think your simplified models (as noted in points 4 and 5 of your response letter) are sufficient to address any concerns regarding your analytic approach. That is, you will need to continue to use these refined models if you choose to go back to your original sample.

Second, and related to the issue of power, you added text regarding a power analysis. While this seems to be an effort to convince the reader that your statistical tests are sufficiently powered, I worry that it would function as a null power test for many readers. As you note, post hoc power studies are inherently flawed, and therefore, I don't think this language is needed. Please remove this language. 

In both of these cases, I think these are relatively minor changes, and I do not foresee needing to send this out for another review. 

---

## [Author Response · Author response to Decision Letter 5]

11 Aug 2023

1. First, I appreciated your careful attention to the comments about statistical power. While increasing your sample size by including data from a related study certainly addressed the reviewer’s concern about power, I worry that it also adds an additional confound that may introduce problems if the experimental paradigms introduce unknown systematic differences. For instance, if participants are laying in a MRI magnet, this may alter their attention or response to stimuli in a number of ways. They may be more anxious overall, which could alter their attention to positive or negative stimuli. Or the simple act of laying down may impact the magnitude of muscle activation. Of course, there is no way to know for sure, which is the danger. That said, you may want to keep this larger sample to maintain power. In this case, you might consider ways to verify that they data collection procedures do not impact your models. This could be done by comparing the two data collection methods directly, conducting a sensitivity analysis, or including the data collection method as a factor in the model. However, I think it is also acceptable to just report on your original sample (N=50) as these tests appeared to be sufficiently powered to address your main findings. If you choose to report on your original sample, however, I think your simplified models (as noted in points 4 and 5 of your response letter) are sufficient to address any concerns regarding your analytic approach. That is, you will need to continue to use these refined models if you choose to go back to your original sample.

Authors’ reply: 

We thank the editor for the suggestion. According to the recommendations, we made the following changes to our analysis:

1. We add the random intercept over the random factor “Type,” wherever this did not cause a singular fit. We already included the random intercept over “Type” in the valence rating analysis in the previous revision. We also include the random intercept over “Type” in the ZM and CS response analysis. Including random slopes over “Type,” or modeling the random factor as “subject nested in Type” also invariably caused singular fits, so only random intercept could be added. However, adding the random intercept over “Type” in the arousal rating analysis invariably caused singular fits, so the random factor Type could not be added for the arousal ratings (Line 550). Information about model comparisons between models, including and not including random intercept over “Type,” was also added to the Results section (Lines 490-492, 594-596, 622-624). The significant patterns were the same as in the last revision.

2. We also included robust data estimations from 50 participants in the behavioral study of the same models as with 94 participants in the supplementary tables S2, S4, S6, and S8. We see a few interactions significant with 94 participants, not significant with 50 participants (Emotion X Presentation in valence and arousal ratings; IRIEC X Emotion in CS responses), which we attributed to the effect of statistical power. No further discussion was added. We also added information about a sensitivity analysis for 50 participants in the Participant section of the Methods, showing that 50 is apt for detecting medium effect sizes (Lines 244-247).

2. Second, and related to the issue of power, you added text regarding a power analysis. While this seems to be an effort to convince the reader that your statistical tests are sufficiently powered, I worry that it would function as a null power test for many readers. As you note, post hoc power studies are inherently flawed, and therefore, I don't think this language is needed. Please remove this language. 

Authors’ reply: 

We aimed to justify the sample size we offered in this study by showing that 94 is apt for detecting a small-to-medium effect size, and the effect size estimation for a small effect size is reliable. As the editor suggested, sensitivity analyses are sufficient for this purpose, so we have removed other means of sample size estimation in the Participant section of the Methods (Lines 236-247).

---

## [Editor Report · Decision Letter 6]

16 Aug 2023

An Investigation of the Modulatory Effects of Empathic and Autistic Traits on Emotional and Facial Motor Responses during Live Social Interactions

PONE-D-21-34896R6

Dear Dr. Hsu,

We’re pleased to inform you that your manuscript has been judged scientifically suitable for publication and will be formally accepted for publication once it meets all outstanding technical requirements.

Kind regards,

Eric J. Moody, Ph.D.

Academic Editor

PLOS ONE
---

## [Editor Report · Acceptance letter]

21 Aug 2023

PONE-D-21-34896R6 

An Investigation of the Modulatory Effects of Empathic and Autistic Traits on Emotional and Facial Motor Responses during Live Social Interactions 

Dear Dr. Hsu:

I'm pleased to inform you that your manuscript has been deemed suitable for publication in PLOS ONE. Congratulations! Your manuscript is now with our production department. 

Kind regards, 

on behalf of

Dr. Eric J. Moody 

Academic Editor

PLOS ONE